# Understanding Behavior Cloning with Action Quantization

**Haoqun Cao** [1]  **Tengyang Xie** [1]

## Abstract

Behavior cloning is a fundamental paradigm in machine learning, enabling policy learning from expert demonstrations across robotics, autonomous driving, and generative models. Autoregressive models like transformer have proven remarkably effective, from large language models (LLMs) to vision-language-action systems (VLAs). However, applying autoregressive models to continuous control requires discretizing actions through quantization, a practice widely adopted yet poorly understood theoretically.

This paper provides theoretical foundations for this practice. We analyze how quantization error propagates along the horizon and interacts with statistical sample complexity. We show that behavior cloning with quantized actions and log-loss achieves optimal sample complexity, matching existing lower bounds, and incurs only polynomial horizon dependence on quantization error, provided the dynamics are stable and the policy satisfies a probabilistic smoothness condition. We further characterize when different quantization schemes satisfy or violate these requirements, and propose a model-based augmentation that provably improves the error bound without requiring policy smoothness. Finally, we establish fundamental limits that jointly capture the effects of quantization error and statistical complexity.

## 1. Introduction

Behavior cloning (BC), the approach of learning a policy from expert demonstrations via supervised learning (Pomerleau, 1988; Bain and Sammut, 1995), has emerged as a foundational paradigm across artificial intelligence. In robotics and autonomous driving, BC enables learning complex manipulation and navigation skills directly from human demonstrations (Black et al., 2024). In generative AI, next-token prediction with cross-entropy (the standard pretraining objective for large language models) can be viewed as behavior cloning from demonstration data.

A key architectural choice driving recent progress is the use of *autoregressive* (AR) models. AR transformers have proven remarkably effective both in language modeling and, more recently, in vision-language-action (VLA) models for robotics (Brohan et al., 2022; Chebotar et al., 2023; Kim et al., 2025). However, applying AR models to continuous control introduces a fundamental design decision: continuous action signals must be *quantized* (or *tokenized*) into discrete symbols. This involves mapping each continuous action vector to a finite token via a quantizer (e.g., per-dimension binning), after which the learner models a distribution over a finite action alphabet. Quantization can (i) reduce effective model complexity, (ii) improve coverage of the hypothesis class under finite-data constraints, and (iii) leverage state-of-the-art transformer architectures designed for discrete prediction (Driess et al., 2026). While action quantization has been widely explored in practice, from uniform binning (Brohan et al., 2022; Zitkovich et al., 2023) to learned vector quantization (Lee et al., 2024; Belkhale and Sadigh, 2024) and time-series compression (Pertsch et al., 2025), its theoretical underpinnings remain poorly understood.

This paper aims to provide theoretical foundations for understanding when and why action quantization works (or fails) in behavior cloning. Raw BC is already vulnerable to *distribution shift*: small deviations from the expert can drive the learner into states rarely covered by training data. Action quantization introduces an additional, inevitable mismatch; even with perfect fitting, the quantizer induces nonzero distortion that can compound over long horizons. We study how *quantization error* and *statistical estimation error* jointly propagate in finite-horizon MDPs. While prior work analyzes statistical limits of BC (e.g., Ross and Bagnell, 2010; Ross et al., 2011; Rajaraman et al., 2020; Foster et al., 2024) and, separately, optimal lossy quantization (e.g., Widrow et al., 1996; Ordentlich and Polyanskiy, 2025), there is little understanding of their *interaction*; and this paper aims to fill this gap.

---

[1]University of Wisconsin–Madison, Madison, WI, USA. Correspondence to: Haoqun Cao <hcao65@wisc.edu>, Tengyang Xie <tx@cs.wisc.edu>.

*Proceedings of the $43^{rd}$ International Conference on Machine Learning*, Seoul, South Korea. PMLR 306, 2026. Copyright 2026 by the author(s).

**Contributions.** We study behavior cloning with log-loss (Foster et al., 2024) under action quantization, and make the following contributions:

1. We establish an upper bound on the regret as a function of the sample size and the quantization error, assuming stable dynamics and smooth quantized policies (expert and learner).

2. We show that a general quantizer can have small *in-distribution* quantization error yet still violate the smoothness requirement, which can lead to large regret; in contrast, binning-based quantizers are better behaved in this respect.

3. We propose a model-based augmentation that bypasses the smoothness requirement on the quantized policy and yields improved horizon dependence for the quantization term.

4. We prove information-theoretic lower bounds that depend on both the sample size and the quantization error, and show that our upper bound generally matches these limits.

### 1.1. Related Work

**Sample Complexity and Fundamental Limits of Imitation Learning.** The sample complexity of behavior cloning with 0-1 loss is studied in Ross et al. (2011) and Rajaraman et al. (2020). Let the sample size be $n$ and the horizon be $H$. Essentially, a regret of $\frac{H^2}{n}$ is established for deterministic and $\tilde{O}(\frac{H^2}{n})$ up to logarithmic factors for stochastic experts, with a modified algorithm in tabular setting. In a recent work of Foster et al. (2024), they study behavior cloning with Log-loss in a function class $\Pi$, and through a more fine-grained analysis through Hellinger distance, they establish a upper bound of $\frac{H \log |\Pi|}{n}$ for deterministic experts and $H\sqrt{\frac{\log |\Pi|}{n}}$ for stochastic experts with realizability and finite class assumptions. In terms of lower bound, Rajaraman et al. (2020) showed in tabular setting that $\frac{H^2}{n}$ is tight. For stochastic experts, Foster et al. (2024) showed that $\sqrt{\frac{1}{n}}$ is inevitable if the we allow the expert to be suboptimal.

**Behavior Cloning in Continuous Control Problem.** Another line of research studies behavior cloning through a control-theoretic lens (Chi et al., 2023; Pfrommer et al., 2022; Block et al., 2023; Simchowitz et al., 2025; Zhang et al., 2026). They focus on deterministic control problems where both the expert policy and the dynamics are deterministic, and use stability conditions to bound the rollout deviation in the metric space. Essentially, they point out that in continuous action spaces, imitating a deterministic

expert under deterministic transitions is difficult, and cannot be done with 0–1 or log loss (Simchowitz et al., 2025). Our work also tries to address this issue by discretizing the continuous action space via quantization, while controlling the quantization error.

**Quantization in Generative Models.** Quantization methods are widely used in modern generative models, where data are first compressed into discrete representations and then learned through a generative model (Van Den Oord et al., 2017; Esser et al., 2021; Tian et al., 2024). In sequential decision-making settings such as robotic manipulation, there is also a growing body of work that adopts action quantization. Brohan et al. (2022); Zitkovich et al. (2023); Kim et al. (2025) use a binning quantizer that discretizes each action dimension into uniform bins. Other works adopt vector-quantized action representations (Lee et al., 2024; Belkhale and Sadigh, 2024; Mete et al., 2024): they train an encoder–decoder with a codebook as a vector quantizer under a suitable loss, and then model the distribution over the resulting discrete codes. Dadashi et al. (2022) study learning discrete action spaces from demonstrations for continuous control. More recently, Pertsch et al. (2025) perform action quantization via time-series compression. Despite strong empirical progress, theoretical guarantees for these widely used settings remain limited.

## 2. Preliminaries

### 2.1. Background

**Markov Decision Process.** We consider such a finite-horizon Markov decision processes. Formally, a Markov decision process $M = (\mathcal{X}, \mathcal{U}, T, r, H)$. Among them $\mathcal{X} \subset \mathbb{R}^{d_x}, \mathcal{U} \subset \mathbb{R}^{d_u}$ are continuous state and action spaces. The horizon is $H$. $T$ is probability transition functions $T = \{T_h\}_{h=0}^{H}$, where $T_h : \mathcal{X} \times \mathcal{U} \to \Delta(\mathcal{X}), h \geq 1$ and $T_0(\emptyset)$ is the initial state distribution. $r$ is a reward function $r = \{r_h\}, r_h : \mathcal{X} \times \mathcal{U} \to [0, 1]$. A (time-variant, markovian) policy is a sequence of conditional probability function $\pi = \{\pi_h : \mathcal{X} \to \Delta(\mathcal{U})\}$. In this paper, we assume there is an expert policy denoted with $\pi^*$. We denote the distribution of the whole trajectory $\tau = (x_{1:H+1}, u_{1:H})$ generated by deploying $\pi$ on dynamic $T$ as $\mathbb{P}^{\pi,T}$. We denote the accumulative reward under $\pi$ (and transition T) as $J(\pi) = \mathbb{E}_{\mathbb{P}^{\pi,T}}[\sum_{h=1}^{H} r(x_h, u_h)]$.

**Quantizer.** Let $q : \mathcal{U} \to \tilde{\mathcal{U}}$ be a (measurable) quantizer, where $\tilde{\mathcal{U}} \subset \mathcal{U}$ is a finite set of representative actions. For any policy $\pi$, define the pushforward (quantized) policy

$$(q\#\pi)_h(\tilde{u} \mid x) := \pi_h(q^{-1}(\tilde{u}) \mid x), \tilde{u} \in \tilde{\mathcal{U}}.$$

For any $(x, \tilde{u})$ such that $\pi_h^*(q^{-1}(\tilde{u}) \mid x) > 0$, we define the *expert-induced dequantization kernel* $\rho_h(\cdot \mid \tilde{u}, x) \in \Delta(\mathcal{U})$

as the conditional law

$$\rho_h(\cdot \mid \tilde{u}, x) := \mathrm{Law}\big(u_h \mid \tilde{u}_h = \tilde{u}, x_h = x\big),$$
$$u_h \sim \pi_h^*(\cdot \mid x_h), \tilde{u}_h = q(u_h).$$

Equivalently,

$$\rho_h(du \mid \tilde{u}, x) = \frac{\mathbf{1}\{u \in q^{-1}(\tilde{u})\}\pi_h^*(du|x)}{\pi_h^*(q^{-1}(\tilde{u})|x)}, \text{ if } \pi_h^*(q^{-1}(\tilde{u}) \mid x) > 0.$$

For $(x, \tilde{u})$ such that $\pi_h^*(q^{-1}(\tilde{u}) \mid x) = 0$, we extend $\rho_h(\cdot \mid \tilde{u}, x)$ by setting it to an arbitrary reference distribution $\nu_{\tilde{u}}$ supported on $q^{-1}(\tilde{u})$. As a result, the kernel $\rho_h(\cdot \mid \tilde{u}, x)$ is now specified for all $(x, \tilde{u}) \in \mathcal{X} \times \tilde{\mathcal{U}}$. Now, given any quantized policy $\hat{\pi}_h(\cdot \mid x) \in \Delta(\tilde{\mathcal{U}})$, we define the induced raw-action policy by mixing $\rho_h$:

$$(\rho \circ \hat{\pi})_h(du \mid x) := \sum_{\tilde{u} \in \tilde{\mathcal{U}}} \rho_h(du \mid \tilde{u}, x)\hat{\pi}_h(\tilde{u} \mid x). \quad (1)$$

Similarly, define the $\rho$-perturbed transition kernel by

$$(T \circ \rho)_h(\cdot \mid x, \tilde{u}) := \int_{\mathcal{U}} T_h(\cdot \mid x, u)\rho_h(du \mid \tilde{u}, x). \quad (2)$$

By construction, $\rho_h(\cdot \mid \tilde{u}, x)$ is supported on $q^{-1}(\tilde{u})$, hence $q\#(\rho \circ \hat{\pi}) = \hat{\pi}$. Moreover, since $\rho$ is the expert-induced kernel, we also have $\rho \circ (q\#\pi^*) = \pi^*$.

**Type of Quantizers.** Fix a metric $\|\cdot\|$ on $\mathcal{U}$. Later we assume the reward is Lipschitz and the dynamics are stable with respect to this metric (typically the Euclidean norm). We consider the following two quantizers: the binning-based one and a learning-based quantizer. Let $\varepsilon_q > 0$ be a small quantity that denotes the quantization error of the quantizer. For binning-based quantizer, we assume that it holds,

$$\|q(u) - u\| \leq \varepsilon_q, \forall u \in \mathcal{U}. \quad (3)$$

For a learning-based quantizer, we assume,

$$\varepsilon_q = \frac{1}{H}\mathbb{E}_{\mathbb{P}^{\pi^*,T}}\left[\sum_{h=1}^H \|u_h - q(u_h)\|\right]. \quad (4)$$

Notice that the binning-based quantizer also satisfies Eq. (4), so it is a special case of a learning-based quantizer. Later, we will express the quantization error in terms of $\varepsilon_q$.

**Notation.** For two nonnegative quantities $a$ and $b$, we write $a = O(b)$ or $a \lesssim b$ if there exists a universal constant $C > 0$ such that $a \leq Cb$. We write $a = \Omega(b)$ if $b = O(a)$, and $a = \Theta(b)$ if both $a = O(b)$ and $a = \Omega(b)$ hold. The constants hidden in $O(\cdot), \Omega(\cdot), \Theta(\cdot)$, and $\lesssim$ are independent of the problem-dependent parameters unless otherwise specified.

## 2.2. Behavior Cloning

We will consider doing behavior cloning in a user-specified policy class $\Pi \subset \{\pi_h : \mathcal{X} \to \Delta(\mathcal{U})\}_{h=1}^H$. In most of this paper, $\Pi$ will consist of *quantized* policies, in which case each $\pi_h$ is supported at $\tilde{\mathcal{U}}$. We will also use the same notation for a general policy class when the intended output space is clear from the context.

**Warm-up: log-loss BC with raw actions (Foster et al., 2024).** In the standard setting where raw actions are observed, log-loss BC solves

$$\hat{\pi} \in \operatorname*{argmax}_{\pi \in \Pi} \sum_{i=1}^n \sum_{h=1}^H \log \pi_h(u_h^i \mid x_h^i),$$
$$\{x_{1:H+1}^i, u_{1:H}^i\}_{i=1}^n \sim \mathbb{P}^{\pi^*,T}.$$

Since the trajectory density under $\mathbb{P}^{\pi,T}$ factorizes as $T_1(x_1)\prod_{h=1}^H \pi_h(u_h \mid x_h)T_h(x_{h+1} \mid x_h, u_h)$ and $T$ does not depend on $\pi$, the above objective is exactly the MLE over the family $\{\mathbb{P}^{\pi,T}\}_{\pi \in \Pi}$.

For rewards in $[0, 1]$, we have the standard coupling bound

$$J(\pi^*) - J(\hat{\pi}) \leq H \cdot \mathrm{TV}\left(\mathbb{P}^{\pi^*,T}, \mathbb{P}^{\hat{\pi},T}\right).$$

Moreover, the TV term can be controlled by the (squared) Hellinger distance $D_H^2$: in Appendix A we show that

$$\mathrm{TV}\left(\mathbb{P}^{\pi^*,T}, \mathbb{P}^{\hat{\pi},T}\right) \lesssim D_H^2\left(\mathbb{P}^{\pi^*,T}, \mathbb{P}^{\hat{\pi},T}\right), \pi^* \text{ deterministic,}$$

$$\mathrm{TV}\left(\mathbb{P}^{\pi^*,T}, \mathbb{P}^{\hat{\pi},T}\right) \lesssim \sqrt{D_H^2\left(\mathbb{P}^{\pi^*,T}, \mathbb{P}^{\hat{\pi},T}\right)}, \text{ in general.}$$

Combining these with standard MLE guarantees in Hellinger distance (e.g., Geer, 2000) yields regret rates $O(H \log |\Pi|/n)$ for deterministic experts and $O(H\sqrt{\log |\Pi|/n})$ for stochastic experts, matching Foster et al. (2024).

**Log-loss BC with action quantization.** Now suppose the learner only observes quantized actions $\tilde{u}_h^i := q(u_h^i)$ and learns a quantized policy $\hat{\pi}_h(\cdot \mid x) \in \Delta(\tilde{\mathcal{U}})$. Then the observed data distribution over $(x, \tilde{u})$ is $\mathbb{P}^{q\#\pi^*,(T\circ\rho)}$. Log-loss BC,

$$\hat{\pi} \in \operatorname*{argmax}_{\pi \in \Pi} \sum_{i=1}^n \sum_{h=1}^H \log \pi_h(\tilde{u}_h^i \mid x_h^i), \quad (5)$$

is the MLE over the family $\{\mathbb{P}^{\pi,(T\circ\rho)}\}_{\pi \in \Pi}$ and therefore (statistically) controls the discrepancy between $\mathbb{P}^{q\#\pi^*,(T\circ\rho)}$ and $\mathbb{P}^{\hat{\pi},(T\circ\rho)}$. However, deployment executes representative actions $\tilde{u} \in \tilde{\mathcal{U}} \subset \mathcal{U}$ in the original environment, generating rollouts from $\mathbb{P}^{\hat{\pi},T}$. In general, guarantee from optimizing Eq. (5) does not directly translate to control of the rollout

distribution $\mathbb{P}^{\hat{\pi},T}$ since $\mathbb{P}^{\hat{\pi},(T\circ\rho)} \neq \mathbb{P}^{\hat{\pi},T}$. Thus, beyond the usual statistical estimation error, one must account for an additional quantization-induced mismatch between the distribution being learned and the distribution being executed, which is an approximation effect of discretizing a continuous action space and is not eliminated by increasing $n$. Our goal is to characterize when this mismatch does not compound badly with $H$, so that log-loss BC remains learnable under quantization.

# 3. Regret Analysis Under Stable Dynamic and Smooth Policy

Intuitively, quantization causes the learner to take actions that deviate from the expert's actions in certain metric space. Such deviations are then propagated through the dynamics. Therefore, stability of the dynamics mitigates this error amplification. Similarly, if the expert policy is non-smooth, the quantized actions may be substantially suboptimal relative to the expert actions, which can also lead to large regret. In this section, we show that log-loss BC is learnable provided that the dynamics are stable and the expert policy is smooth.

## 3.1. Incremental Stability and Total Variation Continuity

Here we introduce the notions of stability and smoothness needed for our results. Our stability notion comes from a concept that has been extensively studied in control theory (e.g., Sontag et al., 1989; Lohmiller and Slotine, 1998; Angeli, 2002) and recently leveraged in provable imitation learning (e.g., Pfrommer et al., 2022; Block et al., 2023; Simchowitz et al., 2025; Zhang et al., 2026). We slightly modify this notion to make it applicable to stochastic dynamics and stochastic policies. To this end, we begin by representing the transition kernel via an explicit noise variable, and then introduce a coupling between two trajectory distributions induced by shared noise.

**Definition 1** (Noise representation of a transition kernel). *We say that $T_h$ admits a noise representation if there exist a measurable space $\Omega$, a probability measure $P_{W_h} \in \Delta(\Omega)$, and a measurable map $f_h : \mathcal{X} \times \mathcal{U} \times \Omega \to \mathcal{X}$ such that for all $(x, u) \in \mathcal{X} \times \mathcal{U}$,*

$$\text{if } \omega \sim P_{\omega,h}, \text{ then } f_h(x, u, \omega) \sim T_h(\cdot \mid x, u).$$

*In particular, the initial state distribution can be represented as $x_1 = f_0(\omega_0)$ with $\omega_0 \sim P_{\omega,0}$.*

We will make such an assumption on the underlying dynamic $f_h$ throughout the paper, which holds in general cases if $T_h$ is a density or is deterministic as we show in Appendix B.1.

**Assumption 1.** *For each $h \in \{0, \ldots, H\}$, the transition kernel $T_h$ admits a noise representation $(f_h, P_{W_h})$ such*

*that, for every $(x, u) \in \mathcal{X} \times \mathcal{U}$, the map $\omega \mapsto f_h(x, u, \omega)$ is injective $P_{W_h}$-almost surely (i.e., given $(x_h, u_h, x_{h+1})$, the corresponding noise $\omega_h$ is uniquely determined, $P_{W_h}$-a.s.).*

**Definition 2.** *(*Shared-noise coupling*) Fix a noise representation $(f_h, P_{W_h})_{h=0}^H$ satisfying the invertibility assumption. Given a trajectory $\tau = (x_{1:H+1}, u_{1:H})$, define $\omega_0(\tau)$ as the (a.s. unique) element such that $x_1 = f_0(\omega_0(\tau))$, and for each $h \in [H]$ define $\omega_h(\tau)$ as the (a.s. unique) element such that*

$$x_{h+1} = f_h(x_h, u_h, \omega_h(\tau)).$$

*Given two policies $\pi^0$ and $\pi^1$, a coupling $\mu$ of $\mathbb{P}^{\pi^0,T}$ and $\mathbb{P}^{\pi^1,T}$ is called a **shared-noise coupling** if, for $\mu$-a.e. $(\tau^0, \tau^1)$,*

$$\omega_h(\tau^0) = \omega_h(\tau^1), \forall h \in \{0, 1, \ldots, H\}.$$

With the above two definitions in place, we introduce a probabilistic notion of incremental input-to-state stability (P-IISS). We use a modulus $\gamma$ that maps any finite list of nonnegative scalars to a nonnegative scalar. We say $\gamma$ is of class $\mathcal{K}$ if it is continuous, coordinate-wise increasing, and satisfies $\gamma(0, \ldots, 0) = 0$. We will write $\gamma\left((r_t)_{t=1}^k\right)$ to represent $\gamma(r_1, \ldots, r_k)$. We will also use $\|\cdot\|$ for the state metric (usually it is the Euclidean norm).

**Definition 3.** *(*Probabilistic Incremental-Input-to-State-Stability*) Define events,*

$$\Phi_h = \{\|u_t^0 - u_t^1\| \le d, \forall t \in [h]\},$$
$$\Psi_h = \{\|x_{t+1}^0 - x_{t+1}^1\| \le \gamma\left((\|u_k^0 - u_k^1\|)_{k=1}^t\right), \forall t \in [h]\}.$$

*We call the trajectory distribution $\mathbb{P}^{\pi^0,T}$ $(\gamma, \delta)$-$d$-locally probabilistically incremental-input-to-state stable (P-IISS) if and only if there exists $\gamma \in \mathcal{K}$, for any other policy $\pi^1$ and the related trajectory distribution $\mathbb{P}^{\pi^1,T}$, it holds that for any **shared noise coupling** $\mu$, $\mu(\Phi_h \cap \Psi_h^c) \le \delta, \forall h \in [H]$.*

*In addition, we say $\mathbb{P}^{\pi^0,T}$ is $\gamma$-globally P-IISS if $\mu(\Psi_h^c) = 0, \forall h \in [H]$. Furthermore, if $\gamma$ in particular satisfies for $C > 0$ and $\eta \in (0, 1)$,*

$$\gamma\left((\|u_k - u_k'\|)_{k=1}^t\right) = \sum_{k=1}^t C\eta^{t-k}\|u_k - u_k'\|,$$

*then we call it probabilistically exponentially-incremental-input-to-state-stable (P-EIISS).*

Definition 3 captures the following intuition: the dynamics are *incrementally stable* only when the action mismatch stays within a *stable region* of radius $d$, and under the expert policy this region is entered (and maintained) with high probability. Here $d$ specifies the locality scale under which two trajectories can remain stable, while $\delta$ accounts for stochasticity in the dynamics: even if the action mismatch

is controlled so that the system stays in the stable regime, the injected noise may still push the state into an unstable region with small probability. In Appendix B.2, we compare P-IISS to existing IISS-type notions in the literature and we also provide an example showing that a locally contractive system perturbed by Gaussian noise, coupled with a gaussian expert policy satisfies P-IISS.

Next, we introduce a smoothness notion for policies. This notion is first brought up in Block et al. (2023).

**Definition 4.** (Relaxed Total Variation Continuity; RTVC) *Fix $\varepsilon' \geq 0$ and define the $\{0,1\}$-valued transport cost $c_{\varepsilon'}(u, u') := \mathbf{1}\{\|u - u'\| > \varepsilon'\}$. For two distributions $P, Q \in \Delta(\mathcal{U})$, define the induced OT distance*

$$W_{c_{\varepsilon'}}(P, Q) := \inf_{\mu \in \Gamma(P,Q)} \mathbb{E}_\mu\big[c_{\varepsilon'}(U, U')\big]$$
$$= \inf_{\mu \in \Gamma(P,Q)} \mu\left(\|U - U'\| > \varepsilon'\right),$$

*where $\Gamma(P, Q)$ is the set of couplings of $(P, Q)$. We say a policy $\pi$ is $\varepsilon'$-RTVC with modulus $\kappa : \mathbb{R}_{\geq 0} \to \mathbb{R}_{\geq 0}$ if for all $h$ and all $x, x' \in \mathcal{X}$,*

$$W_{c_{\varepsilon'}}\left(\pi_h(\cdot \mid x), \ \pi_h(\cdot \mid x')\right) \leq \kappa\left(\|x - x'\|\right).$$

*When $\varepsilon' = 0$, $W_{c_0}(P, Q) = \mathrm{TV}(P, Q)$, and we will call 0-RTVC as total-variation continuity (TVC).*

Later, we will require the expert quantized policy $q\#\pi^*$ to satisfy TVC or RTVC.

**Remark 1.** *We adopt the thresholded 0–1 transport cost $c_{\varepsilon'}$ to obtain a soft notion of continuity that ignores small action perturbations below $\varepsilon'$. A more natural alternative is the distance cost $c(u, u') := \|u - u'\|$, which leads to Wasserstein continuity. Wasserstein continuity is not sufficient to establish the desired bound. For example, for deterministic policy, Wasserstein continuity reduces to the familiar Lipschitz continuity(or modulus-of-continuity). In Appendix B.3, we show that Lipschitz policies still imitate Lipschitz expert with exponential compounding error.*

### 3.2. Upper Bound

In this section, we present our first result, combining the statistical and quantization error, provided that the expert trajectory is stable and policy is smooth. Let $\bar{\mathbb{P}}^{\pi, T, \tilde{\pi}}$ denote the law of the extended trajectory $\bar{\tau} = (x_{1:H+1}, u_{1:H}, \tilde{u}_{1:H})$ generated as follows. Sample $x_1 \sim T_0(\emptyset)$. For each $h \in [H]$, given $x_h$, draw a pair $(u_h, \tilde{u}_h)$ from a coupling kernel $L_h(\cdot, \cdot \mid x_h) \in \Delta(\mathcal{U} \times \mathcal{U})$ whose marginals are $u_h \mid x_h \sim \pi_h(\cdot \mid x_h)$ and $\tilde{u}_h \mid x_h \sim \tilde{\pi}_h(\cdot \mid x_h)$, and then evolve the state by sampling $x_{h+1} \sim T_h(\cdot \mid x_h, u_h)$. Notice that given $x_h$, the dependence between $u_h, \tilde{u}_h$ is governed by the chosen coupling $L_h$ (and will be specified later when needed).

The following proposition is our most general bound: it reduces the regret to a total variation distance between the expert and learner extended trajectory laws, together with several in-distribution terms evaluated under the expert.

**Theorem 1.** *Consider two pairs of policies $\pi^0, \tilde{\pi}^0$ and $\pi^1, \tilde{\pi}^1$ and the associated extended trajectory laws $\bar{\mathbb{P}}^{\pi^0, T, \tilde{\pi}^0}$ and $\bar{\mathbb{P}}^{\pi^1, T, \tilde{\pi}^1}$. Assume $\mathbb{P}^{\pi^0, T}$ is $(\gamma, \delta)$-d-locally P-IISS and $\tilde{\pi}^1$ is $\varepsilon'$-RTVC with modulus $\kappa$. Let $\pi^2 := \tilde{\pi}^1$. If the reward function is $L_r$-Lipschitz, then it holds that*

$$\left|J(\pi^0) - J(\pi^2)\right| \lesssim H^2\delta + H \cdot \mathrm{TV}(\bar{\mathbb{P}}^{\pi^0, T, \tilde{\pi}^0}, \bar{\mathbb{P}}^{\pi^1, T, \tilde{\pi}^1})$$
$$+ H \cdot \bar{\mathbb{P}}^{\pi^0, T, \tilde{\pi}^0}\left(\exists h \in [H], \|\tilde{u}_h - u_h^0\| > d - \varepsilon'\right)$$
$$+ H \cdot \mathbb{E}_{\bar{\mathbb{P}}^{\pi^0, T, \tilde{\pi}^0}}\left\{\sum_{h=1}^{H}\left[\kappa\big(\gamma\big((\|u_k^0 - \tilde{u}_k^0\| + \varepsilon')_{k=1}^{h-1}\big)\big)\right.\right.$$
$$\left.\left. + \frac{1}{H}L_r\left(\gamma\big((\|u_k^0 - \tilde{u}_k^0\| + \varepsilon')_{k=1}^{h-1}\big) + \|u_h^0 - \tilde{u}_h^0\| + \varepsilon'\right)\right]\right\}.$$

When applying this result, we substitute $(\pi^0, \tilde{\pi}^0) = (\pi^*, q\#\pi^*)$ and $(\pi^1, \tilde{\pi}^1) = (\rho \circ \hat{\pi}, \hat{\pi})$. The total variation term is then statistically controlled via log-loss, while the remaining terms under $\bar{\mathbb{P}}^{\pi^*, T, q\#\pi^*}$ depend only on the expert and the quantizer and are assumed to be given in our setting. Consequently, this bound does not suffer from exponential compounding (for an appropriate $\gamma$ and $\kappa$). Now we present the concrete bounds about stochastic and deterministic experts. To reduce some irrelevant terms and make the bound simple, we will have structural assumption on modulus $\gamma$ and $\kappa$ as well as the quantizer.

**Theorem 2.** *Suppose $q\#\pi^*$ is stochastic. Suppose every policy in $\Pi$ is TVC with modulus $\kappa$ being a linear function, and $q\#\pi^* \in \Pi$. Then suppose that $\mathbb{P}^{\pi^*, T}$ is globally P-EIISS, then w.p. at least $1 - \delta$,*

$$J(\pi^*) - J(\hat{\pi}) \lesssim H \cdot \sqrt{\frac{\log(|\Pi|\delta^{-1})}{n}} + H^2 \cdot \varepsilon_q.$$

**Theorem 3.** *Suppose $q\#\pi^*$ is deterministic. If $q$ is a binning quantizer as defined in Eq. (3), and every policy in $\Pi$ is $k\varepsilon_q$-RTVC with modulus $\kappa$ for a constant $k > 0$ and $q\#\pi^* \in \Pi$. Suppose $\mathbb{P}^{\pi^*, T}$ is $\gamma$−globally P-IISS, with $\gamma$ satisfying,*

$$\gamma((r_t)_{t=1}^h) = \gamma(\max_{1 \leq t \leq h} r_t), \forall h \in [H],$$

*where we slightly abuse the notation $\gamma$. Then w.p. at least $1 - \delta$,*

$$J(\pi^*) - J(\hat{\pi}) \lesssim H \cdot \frac{\log(|\Pi|\delta^{-1})}{n} + H^2\kappa\left(\gamma((k+1)\varepsilon_q\right)$$
$$+ H[(\gamma((k+1)\varepsilon_q) + (k+1)\varepsilon_q]$$

Those results are both direct corollaries of Theorem 1. We will show in Section 4 that the assumptions that we make

will hold reasonably. Notice that the first result is on stochastic quantized expert, the $\sqrt{\frac{1}{n}}$ rate matches the rate in Foster et al. (2024). For the deterministic quantized expert, we assume the policy class is RTVC and the quantizer is a binning quantizer. We will show in Section 4 that such assumption is necessary.

**Proof Sketch.** We sketch the proof of Theorem 1. The core observation is that $\tilde{u}_h^1$ (under $\bar{\mathbb{P}}^{\pi^1,T,\tilde{\pi}^1}$) and $u_h^2$ (under $\mathbb{P}^{\pi^2,T}$) are both sampled from the same RTVC policy $\tilde{\pi}_h^1$, but at two different states $x_h^1$ and $x_h^2$. Thus, by the $\varepsilon'$-RTVC property, we can choose a stepwise coupling such that, for each $h$ and conditioning on the past,

$$\Pr\left(\|\tilde{u}_h^1 - u_h^2\| > \varepsilon' \big| x_h^1, x_h^2, \text{past}\right) \le \gamma_{\text{RTVC}}\left(\|x_h^1 - x_h^2\|\right).$$

Define the "good history" event $A_h := \{\|\tilde{u}_t^1 - u_t^2\| \le \varepsilon', \ \forall t \le h-1\}$. Conditioning on $A_h$, stability (P-IISS) converts the state mismatch into past action mismatch, and using the triangle inequality $\|u_t^1 - u_t^2\| \le \|u_t^1 - \tilde{u}_t^1\| + \|\tilde{u}_t^1 - u_t^2\| \le \|u_t^1 - \tilde{u}_t^1\| + \varepsilon'$ on $A_h$, we obtain

$$\Pr\left(\|\tilde{u}_h^1 - u_h^2\| > \varepsilon' \mid A_h\right) \le \mathbb{E}\left[\gamma_{\text{RTVC}}\left(\|x_h^1 - x_h^2\|\right)\big|A_h\right]$$
$$\overset{\text{(P-IISS)}}{\lesssim} \mathbb{E}\left[\gamma_{\text{RTVC}}\left(\gamma\left((\|u_t^1 - \tilde{u}_t^1\| + \varepsilon')_{t=1}^{h-1}\right)\right)\Big|A_h\right] + \delta.$$

Iterating over $h$ yields an in-distribution control of the "wrong-event" probability, of the form

$$\Pr(A_H^c) \lesssim H\delta + \sum_{h=1}^{H}\mathbb{E}\left[\gamma_{\text{RTVC}}\left(\gamma\left((\|u_t^1 - \tilde{u}_t^1\| + \varepsilon')_{t=1}^{h-1}\right)\right)\right],$$

where the expectation is taken under the corresponding extended rollout law.

Two technical simplifications were made above. First, $\mathbb{P}^{\pi^1,T}$ need not be stable. Second, the expectations are initially under $\bar{\mathbb{P}}^{\pi^1,T,\tilde{\pi}^1}$ rather than the expert-side law $\bar{\mathbb{P}}^{\pi^0,T,\tilde{\pi}^0}$. Both are handled by inserting a change-of-measure step: we add a total variation term to transfer bounds from $\bar{\mathbb{P}}^{\pi^1,T,\tilde{\pi}^1}$ to $\bar{\mathbb{P}}^{\pi^0,T,\tilde{\pi}^0}$.

Additionally, P-IISS is stated under a shared-noise coupling. However, our change-of-measure step relies on a maximal coupling that attains the total variation distance, and it is not a priori clear that such a coupling can be realized as shared-noise. This issue is resolved by Lemma 21, which shows that a maximal (TV-attaining) coupling can indeed be implemented via shared noise, so the stability argument remains valid. Finally, notice that the Log-loss guarantee we showed in Section 2.2 does not directly bound the distance between $\mathbb{P}^{\pi^*,T,q\#\pi^*}$ and $\mathbb{P}^{(\rho\circ\hat{\pi}),T,\hat{\pi}}$ (as we substitute $(\pi^0, \tilde{\pi}^0) = (\pi^*, q\#\pi^*)$ and $(\pi^1, \tilde{\pi}^1) = (\rho\circ\hat{\pi}, \hat{\pi})$), however Lemma 14 shows that this distance is the same as the distance of the marginal distribution $\mathbb{P}^{q\#\pi^*,(T\circ\rho)}$ and $\mathbb{P}^{\hat{\pi},(T\circ\rho)}$.

**Practical Takeaways.** The main practical takeaway is that, when applying behavior cloning with quantization (tokenization), the stability of the system plays a crucial role. Since quantization inevitably incurs information loss, the learned policy must deviate from the expert trajectory to some extent. Without sufficiently strong open-loop stability, such deviations can accumulate and lead to unbounded error. For example, if the modulus $\gamma$ in our P-IISS definition is arbitrarily large, then the resulting regret bound can also become arbitrarily poor.

### 3.3. A Discussion on Regression-BC

As another direct corollary of our Theorem 1, we discuss the phenomenon highlighted in Simchowitz et al. (2025). They study policy learning via *regression behavioral cloning* (regression-BC), which learns a policy by minimizing a metric-based regression loss to expert actions, evaluated under the expert state distribution:

$$\hat{\pi} := \underset{\pi \in \Pi}{\text{argmin}} \sum_{h=1}^{H} \mathbb{E}_{\bar{\mathbb{P}}^{\pi^*,T,\hat{\pi}}}[\|u_h^* - \hat{u}_h\|],$$

where $\Pi$ is a policy class and $u_h^* \sim \pi_h^*(\cdot|x_h^*), \hat{u}_h \sim \hat{\pi}_h(\cdot|x_h^*)$. This setting is unrelated to quantization. Roughly speaking, Simchowitz et al. (2025) proves an algorithmic lower bound showing that, for certain *simple* policy classes, optimizing the above regression loss can still lead to *exponential* compounding of error in the resulting regret. We show that under stability, using a TVC learner may prevent such exponential compounding.

**Proposition 4.** *Assume $\mathbb{P}^{\pi^*,T}$ globally P-EIISS. If $\hat{\pi}$ is TVC with modulus $\kappa$ being linear , then*

$$J(\pi^*) - J(\hat{\pi}) \lesssim H \cdot \sum_{h=1}^{H} \mathbb{E}_{\bar{\mathbb{P}}^{\pi^*,T,\hat{\pi}}}[\|u_h^* - \hat{u}_h\|].$$

This proposition directly links regret to the regression training error, with only a linear-in-$H$ compounding factor. While this does not match the horizon-free guarantees in, e.g., Zhang et al. (2026), the gap is attributable to working with TVC (as we will clarify in the next section). Nevertheless, it already shows that, once the regression objective is optimized within a TVC policy class, the resulting regret is correspondingly controlled rather than exponentially amplified.

## 4. When Policies Violate Smoothness: Characterization and Model-Based Remedy

In the previous section, we require the learner's policy to be (relaxed) total variation continuous. Intuitively, once the learner is off expert distribution, such property will forces

the learner to behave as if it was still on expert trajectory, with a cost of the distance between off-distribution state and in distribution state. This corrects the actions taken on OOD states, but also introducing an suboptimal factor $H$ on the regret in terms of quantization error. Now, we will first point out when quantized expert policy may violate such assumption and the consequence, then talk about how to attain a better rate without requiring TVC policy.

### 4.1. When Are Policies Non-Smooth

**Stochastic Expert is reasonable to be TVC.** The upper bound we establish so far requires the whole policy class to be TVC or RTVC, and the quantized expert policy to be realizable. Thus the quantized expert policy should also be TVC or RTVC. In general, many stochastic policies are TVC (e.g. Gaussian policy, shown in Appendix B.3) and Block et al. (2023) introduces a way to augment any policy to a TVC one by probabilistic smoothing (shown in Appendix B.3). Given the raw expert policy is TVC, Lemma 15 guarantees that the quantized policy is still TVC for any measurable quantizer. Thus our Theorem 2 holds in most cases for stochastic expert given the required stability of the dynamic.

**Deterministic Expert almost cannot be TVC.** In contrast, TVC is essentially incompatible with deterministic expert policies. Indeed, if $\pi_h(\cdot \mid x) = \delta_{\pi_h(x)}$ is deterministic, then

$$\text{TV}\left(\delta_{\pi_h(x)}, \delta_{\pi_h(x')}\right) = \mathbf{1}\{\pi_h(x) \neq \pi_h(x')\}.$$

If $\pi_h$ is not locally constant, then for every $\delta > 0$ there exist $x, x'$ with $\|x - x'\| < \delta$ but $\pi_h(x) \neq \pi_h(x')$, hence the left-hand side equals 1 at arbitrarily small scales. Therefore any modulus $\gamma$ satisfying

$$\mathbf{1}\{\pi_h(x) \neq \pi_h(x')\} \leq \gamma(\|x - x'\|),$$

must obey $\gamma(r) = 1$ for all $r > 0$, which is a trivial modulus and result in trivial upper bound. Consequently, we should not expect deterministic policy to satisfy TVC.

**When Deterministic Expert can be RTVC.** Nevertheless, a deterministic quantized expert can still be RTVC with a nontrivial modulus. In what follows, we will first describe what we mean by nontrivial modulus, then we provide a necessary condition for RTVC. We will also show that a binning-quantizer will naturally satisfy RTVC.

**Proposition 5.** *Let us consider a $L$-Lipschitz deterministic policy $\pi$.*

*(i) Suppose that the quantized policy $q\#\pi$ satisfies $\varepsilon'$-RTVC with a nontrivial modulus $\kappa : \mathbb{R}_{\geq 0} \to \mathbb{R}_{\geq 0}$, in the sense that, there exists $\delta_0 > 0$ such that $\kappa(r) < 1$ for all $r \in (0, \delta_0]$.*

*Then the quantizer $q$ must satisfy,*

$$\forall \|x - x'\| \leq \delta_0, \quad \|q(\pi_h(x)) - q(\pi_h(x'))\| \leq \varepsilon'.$$

*(ii) Suppose the quantizer $q$ is a binning quantizer in the sense of Eq. (3). Let $\varepsilon' > 2\varepsilon_q$ and define*

$$\delta_0 := \frac{\varepsilon' - 2\varepsilon_q}{L} > 0.$$

*Then the quantized policy $q\#\pi$ is $\varepsilon'$-RTVC with the following nontrivial modulus:*

$$\kappa(r) = \mathbf{1}\{r > \delta_0\}.$$

The above proposition suggests that, for deterministic policies, the necessary condition (i) implies that a RTVC policy must have certain uniform structure, which cannot be preserved by the general learning-based quantizer applied to a deterministic policy, wheres Part (ii) shows that a binning quantizer does preserve it and, moreover, yields an explicit modulus. This can be directly combined with Theorem 3. Let us consider modulus $\gamma$ as in Theorem 3 and $k > 2$ be some constant. Then whenever the raw expert policy $\pi^*$ is $L$-Lipschitz with $L < \frac{(k-2)\varepsilon_q}{\gamma((k+1)\varepsilon_q)}$, Part (ii) implies that $q\#\pi^*$ is $k\varepsilon_q$-RTVC with modulus $\kappa(r) = \mathbf{1}\{r > (k - 2)\varepsilon_q/L\}$. So realizability in a RTVC policy class can hold . Moreover, $\kappa(\gamma((k+1)\varepsilon_q)) = 0$, so substituting this modulus into Theorem 3 eliminates the $H^2$ term in the upper bound. And this upper bound will be valid. Notice that doing this requires bounding the Lipschitz constant of raw expert policy by $\frac{(k-2)\varepsilon_q}{\gamma((k+1)\varepsilon_q)}$. This condition becomes less restrictive when the state deviation is less sensitive to the action deviation, that is, when $\gamma((k+1)\varepsilon_q)$ is smaller.

**The Price of Non-Smooth Policies.** Now we prove an algorithmic lower bound, showing that using the Log-loss BC with quantized action may incur much larger quantization error than $\varepsilon_q$, when the quantized expert fails to be RTVC. Here we assume access to infinitely many samples. Thus, the error we capture comes from deploying the quantized expert, not from finite-sample effects. The error term is expressed in terms of $\varepsilon_q$.

**Theorem 6.** *For deterministic dynamic, there exists $\mathbb{P}^{\pi^*, T}$ and a learning-based quantizer $q$ such that $\pi^*$ is deterministic and Lipschitz, $\mathbb{P}^{\pi^*, T}$ is globally P-EIISS, and the state distribution of $x_h^*$ (under raw expert distribution) has positive density. Meanwhile it holds that,*

$$\frac{1}{H} \mathbb{E}_{\mathbb{P}^{\pi^*, T}} \left[ \sum_{h=1}^{H} \|u_h^* - q(u_h^*)\| \right] = O(\varepsilon_q),$$

$$J(\pi^*) - J(q\#\pi^*) = H \cdot \Omega(1).$$

*For stochastic dynamic, there exists $\mathbb{P}^{\pi^*, T}$ and a learning-based quantizer q such that the noise distribution is Gaussian, $\pi^*$ is deterministic and Lipschitz, $\mathbb{P}^{\pi^*, T}$ is globally P-EIISS, and the state distribution of $x_h^*$ has positive density . Meanwhile it holds that,*

$$\frac{1}{H}\mathbb{E}_{\mathbb{P}^{\pi^*, T}}\left[\sum_{h=1}^{H}\|u_h^* - q(u_h^*)\|\right] = O(\varepsilon_q),$$

$$J(\pi^*) - J(q\#\pi^*) = H \cdot \Omega\left(\frac{1}{|\log \varepsilon_q|}\right).$$

For the above result, we show that even when the quantization error under the expert distribution is small, on the order of $O(\varepsilon_q)$, deploying the quantized policy can induce rollouts under a drastically different state distribution and lead to large regret (e.g., $\Omega(1) \gg O(\varepsilon_q)$ or $\Omega(1/|\log \varepsilon_q|) \gg O(\varepsilon_q)$). Moreover, this failure mode cannot be remedied by stability of the dynamics or by benignness of the unquantized policy. Finally, we also conduct a simple simulation in Appendix D.4 following the lower-bound construction, showing empirically that a non-smooth learned quantizer can suffer from significantly larger compounding errors than binning-based quantizers.

**Practical Takeaways.** When applying behavior cloning with action quantization (tokenization), the choice of quantizer is crucial. Our theory suggests that, although some learned quantizers may achieve low error on the expert distribution (or during training on raw expert data), they can incur much larger errors at deployment due to non-smoothness. In contrast, a smooth quantized policy (i.e., a TVC or RTVC policy) like using the binning-based quantizers can guarantee that the quantization error at deployment remains comparable to that on the expert distribution. We note that Lee et al. (2024) emphasize that a continuous correction head, which they refer to as an offset head, is essential when using a VQ-type quantizer. In our view, this mechanism helps correct the distribution shift induced by quantization, which is not naturally controlled by a VQ-type quantizer that does not satisfy smoothness.

### 4.2. A Model-Based Augmentation

When (R)TVC policies are inaccessible, we introduce a simple modification that bypasses this requirement. It also improves the horizon dependence of the quantization error, since we no longer need to pay an additional cost associated with correcting out-of-distribution (OOD) states via (R)TVC.

The key observation is that deploying the learned policy on the true state during inference can be risky and may lead to OOD rollouts. Instead, we execute the learned policy on a *simulated* state that is generated only from the training

---

**Algorithm 1** Log-loss BC with auxiliary rollout

**Require:** Dataset $\mathcal{D} = \{(x_{1:H}^{(i)}, u_{1:H}^{(i)})\}_{i=1}^{n}$; quantizer $q$; horizon $H$; policy class $\Pi$; transition class $\mathcal{T}$; real dynamics $\{T_h\}_{h=0}^{H}$

1: **Training (MLE with quantized actions)**

$$\hat{\pi}, \widehat{(T \circ \rho)} = \operatorname*{argmax}_{\pi, (T\circ\rho)\in\Pi\times\mathcal{T}} \sum_{i=1}^{n}\sum_{h=1}^{H}\Big[\log \pi_h\big(q(u_h^{(i)}) \mid x_h^{(i)}\big) + \log(T \circ \rho)_h\big(x_{h+1}^{(i)} \mid x_h^{(i)}, q(u_h^{(i)})\big)\Big]$$

2: **Inference**
3: Sample $x_1^L \sim T_0$, set $x_1^a = x_1^L$
4: **for** $h = 1$ to $H$ **do**
5: $\quad u_h^a \sim \hat{\pi}_h(\cdot \mid x_h^a)$
6: $\quad x_{h+1}^a \sim \widehat{(T \circ \rho)}_h(\cdot \mid x_h^a, u_h^a)$
7: **end for**
8: **for** $h = 1$ to $H$ **do**
9: $\quad u_h^L \leftarrow u_h^a$
10: $\quad$ Observe $x_{h+1}^L \sim T_h(\cdot \mid x_h^L, u_h^L)$ and (optionally) collect $r_h$
11: **end for**
12: **return** $\hat{\pi}, \widehat{(T \circ \rho)}$ and total return $\sum_{h=1}^{H} r_h$

---

data and the initial state (the initial state distribution can be also seen as part of expert distribution) and thus is intended to stay within (or close to) the expert distribution. Concretely, we additionally learn a transition model $\widehat{(T \circ \rho)}$ that predicts the next state given the current state $x_h$ and the quantized action $\tilde{u}_h$, trained by maximum likelihood. At inference time, starting from the observed initial state $x_1^L$, we roll out an auxiliary trajectory $\tau^a = \{(x_h^a, u_h^a)\}_{h=1}^{H}$ by sampling from the learned policy and the learned transition model, with $x_1^a = x_1^L$. We then execute the resulting action sequence $\{u_h^a\}_{h=1}^{H}$ in the real environment from the same initial state. In this way, we effectively "copy" quantized actions along an in-distribution trajectory, and the resulting quantization error can be controlled via the stability condition. Denote $\mathcal{M} \subset \{(T \circ \rho)_h : \tilde{\mathcal{U}} \times \mathcal{X} \to \Delta(\mathcal{X})\}_{h=1}^{H}$ as the transition model class, we present Algorithm 1.

Now we provide the regret guarantee of our algorithm.

**Theorem 7.** (Convergence-Theorem of Model-augmented Imitation Learning)

*Suppose that $\mathbb{P}^{\pi^*, T}$ is globally P-EIISS. Assume realizibility for both $(T \circ \rho) \in \mathcal{M}$ and $q\#\pi^* \in \Pi$. Denote the return of the model-augmented algorithm as $J(\mathrm{alg})$, we have with probability at least $1 - \delta$*

$$J(\pi^*) - J(\mathrm{alg}) \le H \cdot \Big[\sqrt{\frac{\log(|\Pi|\delta^{-1}) + \log(|\mathcal{M}|\delta^{-1})}{n}} + \varepsilon_q\Big].$$

Notice that, compared with Theorem 2 and Theorem 3, this bound has a milder dependence on the horizon in the quantization term. This is not for free, since we need both additional $\log |\mathcal{M}|$, reflecting the complexity of the transition model class. And we also assume realizability for the transition class. Admittedly, learning an $H$-step transition model can be challenging, and $|\mathcal{M}|$ may itself hide extra $H$-dependent factors. This issue may be mitigated by applying the model augmentation only over short sub-horizons (rather than the entire trajectory), in the spirit of action chunking (Zhao et al., 2023). We will leave it as the future work.

## 5. Lower Bound

In this section, we establish lower bounds for offline imitation learning under action quantization. We show that the overall error has two sources: finite-sample estimation error and intrinsic quantization error. These contributions are additive in the sense that eliminating one does not remove the other. We present separate lower bounds for stochastic and deterministic experts, which exhibit different sample-complexity behavior, while the quantization term attains the same rate in both cases. We restrict attention to non-anticipatory offline imitation learning algorithms: for each time step $h$, the learned component $\hat{\pi}_h$ is constructed using only the training data up to step $h$ (i.e., all trajectory prefixes $\{(x_t^i, u_t^i)\}_{t \leq h}$), and it does not use any data from future steps $t > h$.

**Theorem 8.** (Deterministic Expert) *For any non-anticipatory offline imitation learning algorithm on quantized data, there exists $(\mathbb{P}^{\pi^*, T}, q, r)$ such that $q \# \pi^*$ is deterministic and $\pi^*$ is deterministic and $L_\pi$-Lipschitz, $r$ is 1-Lipschitz, $\mathbb{P}^{\pi^*, T}$ is $\gamma$-globally P-IISS with $\gamma((r_k)_{k=1}^h) = r_1$, such that*

$$\sup_{u \in \mathcal{U}} \|u - q(u)\| \leq \varepsilon_q,$$

$$\mathbb{E}[J(\pi^*) - J(\hat{\pi})] \gtrsim H(\frac{1}{n} + \varepsilon_q).$$

**Theorem 9.** (Stochastic Expert) *If we allow $\pi^*$ to be sub-optimal, then for any non-anticipatory offline imitation learning algorithm on quantized data, there exists $(\mathbb{P}^{\pi^*, T}, q, r)$ such that $\mathbb{P}^{\pi^*, T}$ is $\gamma$-globally P-IISS with $\gamma((r_k)_{k=1}^h) = r_1$, $\pi^*$ is stochastic and state independent, and $r$ is 1-Lipschitz, such that,*

$$\sup_{u \in \mathcal{U}} \|u - q(u)\| \leq \varepsilon_q,$$

$$\mathbb{P}\left(J(\pi^*) - J(\hat{\pi}) \geq cH \cdot (\sqrt{\frac{1}{n}} + \varepsilon_q)\right) \geq \frac{1}{8}.$$

These two lower bounds are largely in line with intuition. In offline imitation learning with quantized data, one cannot in general hope to improve upon a rate of order $H \cdot (\text{error}_{\text{stat}} +$

$\text{error}_{\text{quantization}})$, since the learner must effectively recover information across all $H$ stages. It is also natural that the lower bound is additive in the statistical and quantization errors: even if one source of error were removed entirely, the other would still remain. Notice that in the stochastic expert case, the statistical term scales as $\frac{H}{\sqrt{n}}$, this matches the lower bound established in Foster et al. (2024). Comparing these lower bounds with our upper bounds, we see that the sample-size-dependent terms are matched by Theorem 2, Theorem 3, and Theorem 7 (for stochastic experts), while the quantization-dependent term is matched by Theorem 7.

## 6. Conclusion

In this work, we study behavior cloning in continuous space when actions are quantized. For a fixed quantizer, we establish a quadratic-in-horizon upper bound for log-loss behavior cloning on quantized actions, which captures both finite-sample error and quantization error, by introducing a stability notion for the dynamics and assuming the quantized policy class is probabilistically smooth. We discuss two widely used quantizers and characterize when they satisfy the required smoothness condition. Our results indicate that, under log-loss behavior cloning, binning-based quantizers are preferable to general learning-based quantizers, especially when cloning a deterministic policy. We then introduce a model-based augmentation that bypasses the smoothness requirement on the quantized policy. Finally, we derive information-theoretic lower bounds that, in general, match our upper bounds.

## Acknowledgements

We thank Ali Jadbabaie and Alexander Rakhlin for their feedback on earlier versions of this work. We acknowledge support of the DARPA AIQ Award.

## Impact Statement

This paper presents work whose goal is to advance the field of machine learning. There are many potential societal consequences of our work, none of which we feel must be specifically highlighted here.

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

# Appendix

## A. Supervised Learning Guarantee

**MLE Guarantee.** We will first state the standard result from MLE estimator (e.g. Geer (2000), Zhang (2006)). Consider a setting where we receive $\{z^i\}_{i=1}^n$ i.i.d. from $z \sim g^\star$, where $g^\star \in \Delta(\mathcal{Z})$. We have a class $\mathcal{G} \subseteq \Delta(\mathcal{Z})$ that may or may not contain $g^\star$. We analyze the following maximum likelihood estimator:

$$\hat{g} = \arg\max_{g \in \mathcal{G}} \sum_{i=1}^n \log\left(g(z^i)\right). \tag{6}$$

To provide sample complexity guarantees that support infinite classes, we appeal to the following notion of covering number (e.g., Wong and Shen (1995) which tailored to the log-loss.

**Definition 5** (Covering number). *For a class $\mathcal{G} \subseteq \Delta(\mathcal{Z})$, we say that a class $\mathcal{G}' \subset \Delta(\mathcal{Z})$ is an $\varepsilon$-cover if for all $g \in \mathcal{G}$, there exists $g' \in \mathcal{G}'$ such that for all $z \in \mathcal{Z}$, $\log\left(g(z)/g'(z)\right) \leq \varepsilon$. We denote the size of the smallest such cover by $N_{\log}(\mathcal{G}, \varepsilon)$.*

**Proposition 10.** *The maximum likelihood estimator in Eq. (6) has that with probability at least $1 - \delta$,*

$$D_H^2(\hat{g}, g^\star) \leq \inf_{\varepsilon > 0}\left\{\frac{6\log\left(2N_{\log}(\mathcal{G}, \varepsilon)/\delta\right)}{n} + 4\varepsilon\right\} + 2\inf_{g \in \mathcal{G}}\log\left(1 + D_{\chi^2}(g^\star\|g)\right).$$

*In particular, if $\mathcal{G}$ is finite, the maximum likelihood estimator satisfies*

$$D_H^2(\hat{g}, g^\star) \leq \frac{6\log\left(2|\mathcal{G}|/\delta\right)}{n} + 2\inf_{g \in \mathcal{G}}\log\left(1 + D_{\chi^2}(g^\star\|g)\right).$$

Note that the term $\inf_{g \in \mathcal{G}}\log\left(1 + D_{\chi^2}(g^\star\|g)\right)$ corresponds to misspecification error, and is zero if $g^\star \in \mathcal{G}$. The proof can be found in Foster et al. (2024).

Now back to our settings. Notice that throughout the paper, we consider the following two Log-loss estimators,

$$\hat{\pi} = \underset{\pi \in \Pi}{\mathrm{argmax}}\, L_{\log}(\pi) = \underset{\pi \in \Pi}{\mathrm{argmax}} \sum_{i=1}^n \sum_{h=1}^H \log(\pi_h(\tilde{u}_h^i|x_h^i))$$

$$\hat{\pi}, \widehat{T \circ \rho} = \underset{\pi, T \circ \rho}{\mathrm{argmax}}\, L_{\log}(\pi, T \circ \rho) = \underset{\pi \in \Pi}{\mathrm{argmax}} \sum_{i=1}^n \sum_{h=1}^H \left[\log(\pi_h(\tilde{u}_h^i|x_h^i)) + \log((T \circ \rho)_h(x_{h+1}^i|\tilde{u}_h^i, x_h^i))\right]. \tag{7}$$

Now we specialize Proposition 10 to the following result.

**Proposition 11.** *Suppose realizibility for both $q\#\pi^*$ and $(T \circ \rho)$, the estimator learned in Eq. (7) has that with probability at least $1 - \delta$,*

$$D_H^2\left(\mathbb{P}^{q\#\pi^*, (T \circ \rho)}, \mathbb{P}^{\hat{\pi}, (T \circ \rho)}\right) \lesssim \frac{\log(|\Pi|\delta^{-1})}{n},$$

$$D_H^2\left(\mathbb{P}^{q\#\pi^*, (T \circ \rho)}, \mathbb{P}^{\hat{\pi}, \widehat{T \circ \rho}}\right) \lesssim \frac{\log(|\Pi|\delta^{-1}) + \log(|\mathcal{M}|\delta^{-1})}{n}.$$

Next we will connect the hellinger distance with the total variation distance. For a fixed transition T, let us define,

$$\rho(\pi^*\|\pi) := \mathbb{E}_{x_{1:H}^*, u_{1:H}^*, \tilde{u}_{1:H} \sim \mathbb{P}^{\pi^*, T, \pi}}[\mathbb{I}(\exists h \in [H], \tilde{u}_h \neq u_h^*)].$$

**Lemma 12.** (Foster et al., 2024) *If $\pi^*$ is deterministic, then it holds,*

$$\rho(\pi^*\|\pi) \leq 4D_H^2(\mathbb{P}^{\pi^*, T}, \mathbb{P}^{\pi, T}).$$

**Lemma 13.** (Connection to a Trajectory-wise $L_\infty$ metric for deterministic $q\#\pi^*$) *If $\pi^*$ is deterministic, then*

$$\mathrm{TV}\left(\mathbb{P}^{\pi^*,T}, \mathbb{P}^{\pi,T}\right) = \rho(\pi^*\|\pi).$$

*Proof.* For notational simplicity, denote $P = \mathbb{P}^{\pi^*,T}, Q = \mathbb{P}^{\pi,T}$. Since $\pi^*$ is deterministic, we abuse $\pi_h^*$ to denote the expert action mapping. Suppose $\pi$ satisfies that,

$$\forall h \in [H], \pi_h(\{\pi_h^*(x)\}|x) > 0, \forall x \in \mathcal{X}. \tag{8}$$

(this means the policy $\pi$ has positive probability measure on the point that deterministic $\pi^*$ will choose). Then we can find a common dominating measure, and the two associated density are,

$$p = T_0(x_1) \prod_{h=1}^{H} \delta_{\pi_h^*(x_h)}(u_h) T_h(x_{h+1}|x_h, u_h)$$

$$q = T_0(x_1) \prod_{h=1}^{H} [\mathbf{1}(u_h = \pi_h^*(x_h))\pi_h(\{\pi_h^*(x)\}|x_h) + \pi_h(u_h|x_h)] T_h(x_{h+1}|x_h, u_h).$$

Then we have,

$$1 - \mathrm{TV}(P, Q) = \int p \wedge q \, d\tau = \int T_0(x_1) \prod_{h=1}^{H} T_h(x_{h+1}|x_h, u_h) \min\{\delta_{\pi_h^*(x_h)}(u_h), \pi_h(\{\pi_h^*(x)\}|x_h)\} d\tau$$

$$= \int_{\mathrm{supp}(P)} T_0(x_1) \prod_{h=1}^{H} T_h(x_{h+1}|x_h, u_h)\pi(\{\pi_h^*(x_h)\}|x_h)$$

$$= \mathbb{E}_P\left[\prod_{h=1}^{H} \pi_h(\{\pi_h^*(x_h)\}|x_h)\right]$$

$$= \mathbb{E}_{x_{1:H^*}, u_{1:H}^* \sim P}\left[\prod_{h=1}^{H} \mathbb{E}_{\tilde{u}_h \sim \pi_h(\cdot|x_h)} \mathbf{1}\{\tilde{u}_h = u_h^*\}\right]$$

$$= 1 - \mathbb{E}_{x_{1:H}^*, u_{1:H}^*, \tilde{u}_{1:H} \sim \mathbb{P}^{\pi^*,T,\hat{\pi}}}[\mathbb{I}(\exists h \in [H], \tilde{u}_h \neq u_h^*)].$$

where the second equation above is because other than this point $\pi_h^*(x)$, $\delta_{\pi_h^*(x_h)}(\cdot) = 0$. The third equation is because $\pi_h(\{\pi_h^*(x_h)\}|x_h) \in (0,1)$. The last two equation is by definition of expectation. If Eq. (8) does not hold, then both sides will be 1, and the result still holds. Thus we prove the result. $\square$

**Remark 2.** *Notice that at the beginning we assume that $\pi$ has positive measure on a point. Essentially, in continuous case, $\pi$ is usually a density, so $\rho(\pi^*\|\pi)$ will become 1, the log-loss bound depend on $\rho(\pi^*\|\pi)$ (e.g. Lemma D.2 in Foster et al. (2024)) will become the maximum $H$, which is a useless trivial upper bound. This also validates the belief that in continuous and deterministic expert case, imitation learning through log-loss will fail, as mentioned in Simchowitz et al. (2025).*

*However if we quantize the action space, then $\pi$ is supported on finite points, and it can have positive mass on a point. So the statistical learning is possible.*

Now we again define $\bar{\mathbb{P}}^{\pi^0,T,\pi^1}$ as the law of extended trajectory $\{x_{1:H}, u_{1:H}, \tilde{u}_{1:H}\}$ where $\{x_{1:H}, u_{1:H}\} \sim \mathbb{P}^{\pi^0,T}$ and $\tilde{u}_h \sim \pi^1(\cdot|x_h)$. Now consider two extended measure $\bar{\mathbb{P}}^{\pi^*,T,q\#\pi^*}$ and $\bar{\mathbb{P}}^{(\rho\circ\hat{\pi}),T,\hat{\pi}}$. Notice that if we take marginal of those two extended trajectory on (state, raw action) trajectory, then the marginals are $\mathbb{P}^{\pi^*,T}$ and $\mathbb{P}^{(\rho\circ\hat{\pi}),T}$. If we take marginals on (state, quantized action) trajectory, the marginals are $\mathbb{P}^{q\#\pi^*,(T\circ\rho)}, \mathbb{P}^{\hat{\pi},(T\circ\rho)}$. The next lemma shows that for TV distance and Hellinger distance, the distance is the same between those extended measures and projected measures.

**Lemma 14.** *For $\pi^*$ and $\hat{\pi}$, define the induced raw policy and transition as in Eq. (1) and Eq. (2), we have*

$$\mathrm{TV}(\mathbb{P}^{q\#\pi^*,(T\circ\rho)}, \mathbb{P}^{\hat{\pi},(T\circ\rho)}) = \mathrm{TV}(\mathbb{P}^{\pi^*,T}, \mathbb{P}^{\rho\circ\hat{\pi},T}) = \mathrm{TV}(\mathbb{P}^{\pi^*,T,q\#\pi^*}, \mathbb{P}^{(\rho\circ\hat{\pi}),T,\hat{\pi}}),$$

$$D_H^2(\mathbb{P}^{q\#\pi^*,(T\circ\rho)}, \mathbb{P}^{\hat{\pi},(T\circ\rho)}) = D_H^2(\mathbb{P}^{\pi^*,T}, \mathbb{P}^{\rho\circ\hat{\pi},T}) = D_H^2(\mathbb{P}^{\pi^*,T,q\#\pi^*}, \mathbb{P}^{(\rho\circ\hat{\pi}),T,\hat{\pi}}).$$

*Proof.* We denote the density of $\mathbb{P}^{\pi^*,T,q\#\pi^*}, \mathbb{P}^{(\rho\circ\hat\pi),T,\hat\pi}$ w.r.t. a common dominating measure as $p^{\pi^*,T,q\#\pi^*}$ and $p^{(\rho\circ\hat\pi),T,\hat\pi}$. Notice that we have,

$$p^{\pi^*,T,q\#\pi^*} = T_0(x_1) \prod_{h=1}^{H} q\#\pi_h^*(\tilde u_h|x_h)\rho_h(u_h|\tilde u_h, x_h)T_h(x_{h+1}|x_h, u_h)$$

$$= T_0(x_1) \prod_{h=1}^{H} \pi_h^*(u_h|x_h)\mathbb{I}(\tilde u_h = q(u_h))T_h(x_{h+1}|x_h, u_h)$$

$$p^{(\rho\circ\hat\pi),T,\hat\pi} = T_0(x_1) \prod_{h=1}^{H} \hat\pi_h(\tilde u_h|x_h)\rho_h(u_h|\tilde u_h, x_h)T_h(x_{h+1}|x_h, u_h)$$

$$= T_0(x_1) \prod_{h=1}^{H} (\rho\circ\hat\pi)_h(u_h|x_h)\mathbb{I}(\tilde u_h = q(u_h))T_h(x_{h+1}|x_h, u_h).$$

Notice that the associated likelihood ratio is,

$$\frac{p^{\pi^*,T,q\#\pi^*}}{p^{(\rho\circ\hat\pi),T,\hat\pi}} = \prod_{h=1}^{H} \frac{q\#\pi_h^*(\tilde u_h|x_h)}{\hat\pi_h(\tilde u_h|x_h)} = \prod_{h=1}^{H} \frac{\pi_h^*(u_h|x_h)}{(\rho\circ\hat\pi)_h(u_h|x_h)}.$$

We define two projection mapping $\phi_1(x_{1:H+1}, u_{1:H}, \tilde u_{1:H}) = (x_{1:H+1}, u_{1:H})$, $\phi_2(x_{1:H+1}, u_{1:H}, \tilde u_{1:H}) = (x_{1:H+1}, \tilde u_{1:H})$. Since the likelihood ratio is a function of only $\phi_1$ or $\phi_2$, by Lemma 15, we have proved that both TV distance are equal to the TV distance of the two extended measure (without projection), so does the Hellinger distance. $\square$

**Lemma 15.** *Consider two probability measure $P, Q$ on a measurable space $\mathcal{X}$. $\mathcal{Y}$ is another measurable space, $\phi$ is a measurable mapping $\phi : \mathcal{X} \to \mathcal{Y}$. Then we have,*

$$\mathrm{TV}(\phi\#P, \phi\#Q) \le \mathrm{TV}(P, Q), D_H^2(\phi\#P, \phi\#Q) \le D_H^2(P, Q).$$

*If $P \ll Q$, then the equality above holds whenever there exists measurable $g$, such that*

$$\frac{dP}{dQ}(x) = g(\phi(x)).$$

The proof can be found in Theorem 3.1 of Liese (2012).

## B. Discussion on P-IISS and RTVC

### B.1. Verification of Assumption 1

**Proposition 16** (Existence of an invertible noise representation)**.** *Assume $\mathcal{X} \subseteq \mathbb{R}^{d_\mathcal{X}}$ is a Borel set. Suppose that for every $(x, u) \in \mathcal{X} \times \mathcal{U}$, $T_h(\cdot \mid x, u)$ is either*

(i) *deterministic: $T_h(\cdot \mid x, u) = \delta_{\bar f_h(x,u)}$ for some measurable $\bar f_h : \mathcal{X} \times \mathcal{U} \to \mathcal{X}$, or*

(ii) *has a density: $T_h(\cdot \mid x, u)$ is absolutely continuous w.r.t. Lebesgue measure and the associated density is strictly positive a.e. w.r.t. Lebesgue measure..*

*Then $T_h$ admits a noise representation $(f_h, P_{W_h})$ satisfying the invertibility assumption as in Assumption 1 .*

*Proof.* If $T_h(\cdot \mid x, u) = \delta_{\bar f_h(x,u)}$, take $\Omega_h = \{0\}$, $P_{W_h} = \delta_0$, and define $f_h(x, u, 0) = \bar f_h(x, u)$. Then $\omega \sim P_{W_h}$ implies $f_h(x, u, \omega) = \bar f_h(x, u)$ a.s., hence $f_h(x, u, \omega) \sim \delta_{\bar f_h(x,u)} = T_h(\cdot \mid x, u)$. Injectivity holds trivially since $\Omega_h$ is a singleton.

Now we discuss the stochastic transition case. Assume $T_h(\cdot \mid x, u)$ is absolutely continuous w.r.t. Lebesgue measure on $\mathbb{R}^d$ with (jointly) measurable density $t_h(\cdot \mid x, u)$. Let $\Omega_h = [0, 1]^{d_\mathcal{X}}$ and $P_{W_h} = \mathrm{Unif}([0, 1]^{d_\mathcal{X}})$. For each $(x, u)$, write $x' = (x'_1, \ldots, x'_{d_\mathcal{X}})$ and define the conditional CDFs recursively by

$$F_1(z \mid x, u) := \int_{(-\infty, z]} t_{h,1}(s \mid x, u)\, ds, \quad F_k(z \mid x, u, x'_{1:k-1}) := \int_{(-\infty, z]} t_{h,k}(s \mid x, u, x'_{1:k-1})\, ds,$$

for $k = 2, \ldots, d_{\mathcal{X}}$, where $t_{h,1}$ is the first marginal density induced by $t_h$ and $t_{h,k}(\cdot \mid x, u, x'_{1:k-1})$ is the usual conditional density induced by $t_h$. Define the (generalized) inverse/quantile maps

$$Q_k(w \mid x, u, x'_{1:k-1}) := \inf\{z \in \mathbb{R} : F_k(z \mid x, u, x'_{1:k-1}) \geq w\}, w \in [0, 1].$$

Given $\omega = (w_1, \ldots, w_{d_{\mathcal{X}}}) \in [0, 1]^{d_{\mathcal{X}}}$, construct $x' = f_h(x, u, \omega)$ by

$$x'_1 = Q_1(w_1 \mid x, u), \qquad x'_k = Q_k(w_k \mid x, u, x'_{1:k-1}), \ \ k = 2, \ldots, d.$$

By the standard inverse-transform sampling argument, if $\omega \sim \mathrm{Unif}([0, 1]^{d_{\mathcal{X}}})$, the resulting $x' = f_h(x, u, \omega)$ satisfies $x' \sim T_h(\cdot \mid x, u)$.

For invertibility, the above quantile maps are a.s. genuine inverses (since the density is positive), and we can recover the noise by

$$w_1 = F_1(x'_1 \mid x, u), \qquad w_k = F_k(x'_k \mid x, u, x'_{1:k-1}), \ \ k = 2, \ldots, d.$$

Hence $\omega \mapsto f_h(x, u, \omega)$ is injective $P_{W_h}$-a.s.

The same construction applies to the initial distribution (take $h = 0$ and drop $(x, u)$), yielding $x_1 = f_0(\omega_0)$. $\qquad\square$

### B.2. Stability of the Dynamic

A function $\beta(x, t)$ is class $\mathcal{KL}$ if it is continuous, $\beta(\cdot, t)$ is continuous, strictly increasing and $\beta(0, t) = 0$ for each fixed $t$, and $\beta(x, \cdot)$ is decreasing for each fixed $x$. We begin by recalling the standard notion of IISS from the literature.

**Definition 6.** *(Definition 4.1 in (Angeli, 2002)) A system $x_{t+1} = f(x_t, u_t)$ is incrementally input-to-state stable(IISS) if there exists a $\mathcal{KL}$ function $\beta$ and $\gamma \in \mathcal{K}$ such that,*

$$\|x_t - x'_t\| \leq \beta(\|x_1 - x'_1\|, t) + \gamma\left((\|u_k - u'_k\|)_{1 \leq k \leq t-1}\right).$$

This notion is open-loop in the sense emphasized in Simchowitz et al. (2025) and Zhang et al. (2026): the stability bound is stated directly in terms of the difference between two input sequences. Building on this perspective, Simchowitz et al. (2025) and section 3 of Zhang et al. (2026) adopt Definition 6 and further require $\gamma$ to take an exponential form, as in our P-EIISS notion.

A related but distinct line of work considers a closed-loop version of incremental stability. In particular, Pfrommer et al. (2022), Block et al. (2023), and section 4 of Zhang et al. (2026) formulate stability at the closed-loop level; see, for example, Definition 3.1 of Pfrommer et al. (2022) for a precise statement.

The above notions are typically stated for noiseless dynamics and as global properties of the system. In particular, any noiseless globally IISS system (which corresponds to Definition 6 above) is trivially $\gamma$-globally P-IISS. Our notion in Definition 3 should be viewed as a stochastic and localized variant of the open-loop perspective. Specifically, it allows stochastic noise in the dynamics and stochastic expert policies, and it does not require global IISS. Instead, it is enough that the system satisfy an IISS-type property locally on a subset. We next present a sufficient condition that guarantees Definition 3.

**Proposition 17.** *Suppose $\mathcal{X} \subset \mathbb{R}^{d_{\mathcal{X}}}$ and $\mathcal{U} \subset \mathbb{R}^{d_{\mathcal{U}}}$. Consider a closed set $S \subset \mathcal{X}$. Consider the stochastic dynamics $x_{t+1} = f(x_t, u_t) + \omega_t$ with $\omega_t \sim \mathcal{N}(0, \sigma^2 I_n)$ and a Gaussian expert policy $\pi_t^*(x_t) = \mathcal{N}(\mu(x_t), \sigma_\pi^2 I)$. The system starts with $x_1 = 0$. Let us assume*

(A1) Contractive Dynamic. *There exist $\eta \in (0, 1)$ and $C > 0$ such that for all $x, x' \in S$ and all $u, u' \in \mathbb{R}^m$,*

$$\|f(x, u) - f(x', u')\| \leq \eta\|x - x'\| + C\|u - u'\|.$$

(A2) Mean policy Lipschitz. *The mean map $\mu$ is $L_\mu$-Lipschitz on $S$:*

$$\|\mu(x) - \mu(x')\| \leq L_\mu\|x - x'\|, \forall x, x' \in S.$$

(A3) Margin for the mean dynamics. *Define the deterministic nominal trajectory*

$$\bar{x}_{t+1} = f(\bar{x}_t, \mu(\bar{x}_t)), \bar{x}_1 = \mathbf{0}$$

*and assume there exists $\rho > 0$,*

$$\text{dist}(\bar{x}_t, S^c) \geq 2\rho, \forall t \in [H].$$

*Define a geometric sum and a closed-loop constant as,*

$$A_H(L) := \sum_{j=0}^{H-1} L^j, L_{\text{cl}} := \eta + CL_\mu.$$

*And define,*

- *Gain function:* $\gamma\big((a_k)_{k=1}^t\big) := \sum_{k=1}^t C\eta^{t-k} a_k, t = 1, \ldots, H:$

- *Input radius:* $d := \frac{\rho}{CA_H(\eta)}.$

- *Failure probability,*

$$\delta = H \exp\left(-\frac{1}{2}\left(\left(\tfrac{\rho}{2\sigma A_H(L_{\text{cl}})} - \sqrt{d_{\mathcal{X}}}\right)_+\right)^2\right) + H \exp\left(-\frac{1}{2}\left(\left(\tfrac{\rho}{2C\sigma_\pi A_H(L_{\text{cl}})} - \sqrt{d_{\mathcal{U}}}\right)_+\right)^2\right),$$

*where $(z)_+ := \max\{z, 0\}$.*

*Then $\mathbb{P}^{\pi^*, T}$ is $(\gamma, \delta)$-d-P-IISS.*

**Remark 3.** *Notice that we allow the dynamic to be only locally contractive and we also allow stochastic dynamic and expert policy here. This setting is not covered in previous line of work (Pfrommer et al. (2022), Block et al. (2023), Simchowitz et al. (2025), Zhang et al. (2026)). Essentially, the failure probability $\delta$ is of order $O\left(H\varepsilon_q\right)$ if $L_{cl} \in (0, 1)$ and $\sigma^2 = \sigma_\pi^2 \in O(\frac{1}{\log|\varepsilon_q|})$, and the stability distance $d \in O(1)$.*

*Proof.* Denote the expert action $u_t^* = \mu(x_t^*) + \zeta_t$ and the system noise as $\omega_t$. Consider any shared-noise coupling $\mu$. Denote G as event,

$$G = \{C \max_{t \leq H} \|\zeta_t\| + \max_{t \leq H} \|\omega_t\| \leq \frac{\rho}{A_H(L_{\text{cl}})}\}.$$

Suppose that $G$ holds under $\mu$, then we have, if $x_t^*, \bar{x}_t \in S$

$$\begin{aligned}
\|x_{t+1}^* - \bar{x}_{t+1}\| &= \|f(x_t^*, \mu(x_t^*) + \zeta_t) + \omega_t - f(\bar{x}_t, \mu(\bar{x}_t))\| \\
&\leq (\eta + CL_\mu)\|x_t^* - \bar{x}_t\| + C\|\zeta_t\| + \|\omega_t\| \\
&\leq (\eta + CL_\mu)\|x_t^* - \bar{x}_t\| + \frac{\rho}{A_H(L_{\text{cl}})}.
\end{aligned}$$

Notice that the initial value is $\|x_1^* - \bar{x}_1\| = 0$ and through iterating, we have,

$$\|x_t^* - \bar{x}_t\| \leq \frac{\rho}{A_H(L_{\text{cl}})} \sum_{k=0}^{t-1} L_{\text{cl}}^k \leq \rho, \forall h \in [H] \Rightarrow \text{dist}(x, S^c) > \rho.$$

Notice that for any other trajectory that shares the same noise $\{\hat{x}_{1:H}, \hat{u}_{1:H}\}$, let us define,

$$\Phi_H := \{\|u_t^* - \hat{u}_t\| \leq \frac{\rho}{CA_H(\eta)}, \forall t \in [H]\}.$$

When $G$ holds, if $\Phi_H$ holds, then we have, if $x_t^*, \hat{x}_t \in S$

$$\begin{aligned}
\|\hat{x}_{t+1} - x_{t+1}^*\| &= \|f(\hat{x}_t, \hat{u}_t) - f(x_t^*, u_t^*)\| \\
&\leq \eta\|\hat{x}_t - x_t^*\| + C\|\hat{u}_t - u_t^*\| \\
&\leq \eta\|\hat{x}_t - x_t^*\| + \frac{\rho}{A_h(\eta)}.
\end{aligned} \tag{9}$$

Since $\hat{x}_1 = x_1^* \in S$, by iteration, we have,

$$\|\hat{x}_t - x_t^*\| \leq \frac{\rho}{A_h(\eta)} \sum_{k=0}^{t-1} \eta^k \leq \rho, \Rightarrow \operatorname{dist}(\hat{x}_t, S^c) > 0, \hat{x}_t \in S, \forall t \in [H].$$

In the meantime, since when G and $\Phi_H$ holds, $x_t^*, \hat{x}_t \in S, \forall h \in [H]$, therefore events,

$$\Psi_H = \{\|\hat{x}_t - x_t^*\| \leq \sum_{k=1}^{t-1} C\eta^{t-k}\|u_k^* - \hat{u}_k\|, \forall t \in [H]\}$$

hold due to Eq. (9).

Thus we have proved that under $G, \Phi_H \Rightarrow \Psi_H$. Then we have,

$$\mu(\Phi_H \cap \Psi_H^c) \leq \mu(G^c) \leq \mu(\max_{t \leq H} \|\zeta_t\| > \frac{\rho}{2CA_H(L_{\mathrm{cl}})}) + \mu(\max_{t \leq H} \|\omega_t\| > \frac{\rho}{2A_H(L_{\mathrm{cl}})}).$$

For $g \sim \mathcal{N}(0, \tau^2 I_d)$, the standard bound

$$\mathbb{P}(\|g\| > t) \leq \exp\left(-\frac{1}{2}\left(\left(\frac{t}{\tau} - \sqrt{d}\right)_+\right)^2\right).$$

Since $\omega_t, \zeta_t$ are independent, using a union bound gives,

$$\mu(\Phi_H \cap \Psi_H^c) \leq H \exp\left(-\frac{1}{2}\left(\left(\frac{\rho}{2\sigma A_H(L_{\mathrm{cl}})} - \sqrt{d_{\mathcal{X}}}\right)_+\right)^2\right) + H \exp\left(-\frac{1}{2}\left(\left(\frac{\rho}{2C\sigma_\pi A_H(L_{\mathrm{cl}})} - \sqrt{d_{\mathcal{U}}}\right)_+\right)^2\right). \qquad \square$$

### B.3. Smoothness of the Policy

Next we talk about when is a policy being TVC. We will show two simple results. The first is that for an arbitrary policy, after Gaussian smoothing, it is TVC with a linear modulus function.

**Proposition 18.** *(Lemma 3.1 of (Block et al., 2023)) Let $\pi = (\pi_h)$ be any policy. Define the Gaussian-smoothed policy as $\pi_\sigma = (\pi_{\sigma,h})$ be distributed,*

$$\pi_{\sigma,h}(\cdot|x_h) = \int \pi_h(\cdot|\tilde{x}_h)\mathcal{N}(\tilde{x}_h; x_h, \sigma^2)d\tilde{x}_h.$$

*Then $\pi_\sigma$ is $\gamma_{\mathrm{TVC}}$-TVC with $\gamma_{\mathrm{TVC}}(u) = \frac{u}{2\sigma}$.*

Moreover, by Lemma 15, if $\pi$ is TVC, then $q\#\pi$ is still TVC for any measurable quantizer $q$.

**Gaussian policies are TVC with linear modulus.** Consider a Gaussian policy with mean $\mu_h(x)$ and variance $\sigma^2 I$. Notice that for $x, x' \in \mathcal{X}$, we have,

$$\mathrm{TV}\left(\pi_h(\cdot \mid x), \pi_h(\cdot \mid x')\right) = \mathrm{erf}\left(\frac{\|\mu_h(x) - \mu_h(x')\|}{2\sqrt{2}\sigma}\right),$$

where,

$$\mathrm{erf}(x) = \frac{2}{\sqrt{\pi}} \int_0^x e^{-t^2} dt \leq \frac{2}{\sqrt{\pi}} x.$$

Suppose $\mu_h(x)$ is $L_\mu$-Lipschitz, then we have,

$$\mathrm{TV}\left(\pi_h(\cdot \mid x), \pi_h(\cdot \mid x')\right) \leq \frac{L_\mu \|x - x'\|}{\sqrt{2}\sigma}.$$

Therefore Gaussian policy is TVC with a linear modulus.

Next we discuss another notion of smoothness for the policy.

**Definition 7** ($W_p$-continuity). *Fix $p \geq 1$. A stochastic policy $\pi(\cdot \mid x)$ on $\mathbb{R}^{d_\mathcal{U}}$ is $W_p$-continuous on $S$ if there exists modulus $\kappa : \mathbb{R}_{\geq 0} \to \mathbb{R}_{\geq 0}$ such that*

$$W_p\left(\pi_h(\cdot \mid x), \pi_h(\cdot \mid x')\right) \leq \kappa(\|x - x'\|), \forall x, x' \in \mathcal{X}.$$

*where $W_p$ denotes the $p$-Wasserstein distance.*

The fact is Wasserstein continuity does not ensure that the in-distribution regression error can be used to control the regret. To see this, simply notice that for $p = 2$, if the policy is deterministic and $\kappa(\cdot)$ is a linear function, then $W_p$ continuous policy are Lipschitz policies. However, Lipschitz learner's may still incur an exponential compounding error.

For example, let us consider the linear system $x_{t+1} = Ax_t + Bu_t$. Let the expert control policy be $\pi^*(x_t) = Kx_t + \delta$, and the learner be $\hat{\pi}(x_t) = Kx_t$ (heuristically, we can also think of it as the quantized expert). Then notice that we have,

$$\sum_{h=1}^{H} \mathbb{E}_{\mathbb{P}^{\pi^*,T}} \|\pi^*(x_h) - \hat{\pi}(x_h)\| = H\delta,$$

$$\sum_{h=1}^{H} \mathbb{E}_{x_t^* \sim \mathbb{P}^{\pi^*,T}, \hat{x}_t \sim \mathbb{P}^{\hat{\pi},T}} \|x_t^* - \hat{x}_t\| = \sum_{h=1}^{H-1} (A+BK)^{H-1-h} B\delta.$$

Essentially, both the expert and the learner is Lipschitz. Notice that if we require $\mathbb{P}^{\pi^*,T}$ to be IISS, a sufficient condition is $\rho(A) < 1$ (where $\rho$ means the spectral radius). However this does not control $(A + BK)$. Thus the regret can be exponentially larger than the training error.

## C. Proofs from Section 3

### C.1. Supporting Lemmas

Below are two lemmas about some standard coupling argument.

**Lemma 19.** (The coupling that reaches the infimum exists) *Let* $d, k \in \mathbb{N}$. *Let* $\Delta := \{(x, y) \in \mathbb{R}^d \times \mathbb{R}^d : x = y\}$. *For Borel probability measures* $P, Q$ *on* $\mathbb{R}^d$, *let* $\mathcal{P}(P, Q)$ *denote the set of all couplings of* $(P, Q)$ *on* $\mathbb{R}^d \times \mathbb{R}^d$.

*(i) For any* $P, Q$ *on* $\mathbb{R}^d$,

$$\sup_{\mu \in \mathcal{P}(P,Q)} \mu(\Delta) = 1 - d_{\mathrm{TV}}(P, Q),$$

*and the supremum is attained by some* $\mu^\star \in \mathcal{P}(P, Q)$ *(we will call it maximal coupling from now on).*

*(ii) Let* $R$ *be a Borel probability measure on* $\mathbb{R}^k$, *and let* $z \mapsto P_z$ *and* $z \mapsto Q_z$ *be Borel stochastic kernels on* $\mathbb{R}^d$. *Then for* $R$-*almost every* $z$ *there exists* $\mu_z^\star \in \mathcal{P}(P_z, Q_z)$ *with*

$$\mu_z^\star(\Delta) = 1 - d_{\mathrm{TV}}(P_z, Q_z).$$

*Moreover, one can choose* $z \mapsto \mu_z^\star$ *Borel-measurably, and the probability measure*

$$\pi^\star(dx, dy, dz) = \mu_z^\star(dx, dy)R(dz)$$

*has* $Z \sim R$ *and conditionals* $X \mid Z = z \sim P_z, Y \mid Z = z \sim Q_z$.

The proof can be found in Lemma C.1 and Corollary C.1 of Block et al. (2023).

**Lemma 20.** (Glueing Lemma) *Suppose that* $X, Y, Z$ *are random variables taking value in Polish spaces* $\mathcal{X}, \mathcal{Y}, \mathcal{Z}$. *Let* $\mu_1 \in \Delta(\mathcal{X} \times \mathcal{Y}), \mu_2 \in \Delta(\mathcal{Y} \times \mathcal{Z})$ *be couplings of* $(X, Y)$ *and* $(Y, Z)$ *respectively. Then there exists a coupling* $\mu \in \Delta(\mathcal{X} \times \mathcal{Y} \times \mathcal{Z})$ *on* $(X, Y, Z)$ *such that under* $\mu$, $(X, Y) \sim \mu_1$ *and* $(Y, Z) \sim \mu_2$.

The proof can be found in Lemma C.2 of Block et al. (2023).

The below lemma shows that the coupling that reaches the TV distance is also a shared-noise coupling, which will help us to transform stability of expert to learner.

**Lemma 21.** *If the dynamics* $x_{h+1} = f(x_h, u_h, \omega_h), h \in [H]$ *are invertible in* $\omega_h$ *(i.e. given* $((x_h, x_{h+1}, u_h)$one can uniquely recover* $\omega_h$), *then there exists a maximal coupling (i.e. the coupling that achieves the TV distance.) between* $\mathbb{P}^{\pi^0,T}$ *and* $\mathbb{P}^{\pi^1,T}$ *that is also a shared-noise coupling.*

*Proof.* For notational convenience, let us denote $P = \mathbb{P}^{\pi^0,T}, Q = \mathbb{P}^{\pi^1,T}$. Let us denote the common support of $P, Q$ as $\mathcal{T}$. Denote $\omega = \omega_{0:H}$, with support $\Omega$ (we abuse the notation). The distribution of $\omega$ denoted with $\nu$. First we prove that

$$\mathrm{TV}(P, Q) = \int \mathrm{TV}(P_{\boldsymbol{\omega}}, Q_{\omega}) d\nu(\omega),$$

where $P_\omega, Q_\omega$ are the conditional distribution of the trajectory given the noise.

Assume for this proof that $\Omega$ and $\mathcal{T}$ are countable (just to use sums; the general case replaces sums by integrals and goes through verbatim). Given a trajectory $\tau = (x_{1:H+1}, u_{1:H}) \in \mathcal{T}$, define $\hat\omega(\tau) = (\hat\omega_0(\tau), \ldots, \hat\omega_H(\tau)) \in \Omega$ to be the unique noise sequence such that

$$x_{h+1} = f\big(x_h, u_h, \hat\omega_h(\tau)\big), h = 0, 1, \ldots, H.$$

Then there is a measurable partition

$$\mathcal{T} = \biguplus_{\omega \in \Omega} \mathcal{T}_\omega, \qquad \mathcal{T}_\omega = \{\tau : \hat\omega(\tau) = \omega\},$$

where $\biguplus$ denotes the disjoint union (i.e., $\mathcal{T} = \bigcup_{\omega \in \Omega} \mathcal{T}_\omega$ and $\mathcal{T}_\omega \cap \mathcal{T}_{\omega'} = \varnothing$ for $\omega \neq \omega'$). And the conditional laws $P_\omega, Q_\omega$ satisfy $P_\omega(\tau) = Q_\omega(\tau) = 0$ for $\tau \notin \mathcal{T}_\omega$. The unconditional laws are mixtures

$$P(\tau) = \sum_{\omega \in \Omega} \nu(\omega) P_\omega(\tau), \qquad Q(\tau) = \sum_{\omega \in \Omega} \nu(\omega) Q_\omega(\tau).$$

On a countable space,

$$d_{\mathrm{TV}}(P, Q) = 1 - \sum_{\tau \in \mathcal{T}} \min\{P(\tau), Q(\tau)\}.$$

Using the block-disjoint support,

$$
\begin{aligned}
\sum_{\tau \in \mathcal{T}} \min\{P(\tau), Q(\tau)\} &= \sum_{\tau \in \mathcal{T}} \min\Big\{\sum_\omega \nu(\omega) P_\omega(\tau), \sum_\omega \nu(\omega) Q_\omega(\tau)\Big\} \\
&= \sum_{\tau \in \mathcal{T}} \min\big\{\nu(\hat\omega(\tau)) P_{\hat\omega(\tau)}(\tau), \nu(\hat\omega(\tau)) Q_{\hat\omega(\tau)}(\tau)\big\} \\
&= \sum_{\omega \in \Omega} \nu(\omega) \sum_{\tau \in \mathcal{T}_\omega} \min\{P_\omega(\tau), Q_\omega(\tau)\},
\end{aligned}
$$

where the second equation is because $P_\omega(\tau) = 0, \omega \neq \hat\omega(\tau)$, the third equation is because $\mathcal{T} = \biguplus_{\omega \in \Omega} \mathcal{T}_\omega$. Hence

$$
\begin{aligned}
d_{\mathrm{TV}}(P, Q) &= 1 - \sum_{\tau \in \mathcal{T}} \min\{P(\tau), Q(\tau)\} \\
&= 1 - \sum_\omega \nu(\omega) \sum_{\tau \in \mathcal{T}_\omega} \min\{P_\omega(\tau), Q_\omega(\tau)\} \\
&= \sum_\omega \nu(\omega) \Big(1 - \sum_{\tau \in \mathcal{T}_\omega} \min\{P_\omega(\tau), Q_\omega(\tau)\}\Big) \\
&= \sum_\omega \nu(\omega) d_{\mathrm{TV}}(P_\omega, Q_\omega) = \int d_{\mathrm{TV}}(P_\omega, Q_\omega) d\nu(\omega).
\end{aligned}
$$

Now that the desired equation is proved. We can seek to construct a **shared-noise** coupling that reaches the TV distance. To do so, we first construct a distribution kernel $L(\tau^0, \tau^1 | \boldsymbol{\omega})$ such that

$$L(\tau^0 \neq \tau^1 | \omega) = \mathrm{TV}(P_\omega, Q_\omega).$$

This is always guaranteed by Lemma 19. Then we construct $\mu$ as $\mu(\cdot) = \int L(\cdot | \omega) d\nu(\omega)$. Then, $\mu$ is clearly a **shared-noise** coupling, meanwhile it's also a maximal coupling since

$$\mu(\tau^0 \neq \tau^1) = \int L(\tau^0 \neq \tau^1 | \omega) d\nu(\omega) = \int \mathrm{TV}(P_\omega, Q_\omega) d\nu(\omega) = \mathrm{TV}(P, Q). \qquad \square$$

### C.2. Proof of Theorem 1

We first prove a upper bound on the **bad event**, then Theorem 1 can be established by law of Total Expectation.

**Lemma 22.** *Consider two pairs of policies $\pi^0, \tilde{\pi}^0$ and $\pi^1, \tilde{\pi}^1$. Then if $\mathbb{P}^{\pi^0, T}$ is $(\gamma, \delta) - d$-locally P-IISS, and $\tilde{\pi}^1$ is $\varepsilon'$-RTVC with modulus $\kappa$. Now denote $\pi^2 := \tilde{\pi}^1$. Then there exists a shared-noise coupling of $\mathbb{P}^{\pi^0, T, \tilde{\pi}^0}$ and $\mathbb{P}^{\pi^2, T}$ such that,*

$$
\mu \left( \{ \cup_{h=1}^H \| \tilde{u}_h^0 - u_h^2 \| > \varepsilon' \} \right) \leq \sum_{h=1}^H \mathbb{E}_{\bar{\mathbb{P}}^{\pi^0, T, \tilde{\pi}^0}} \left[ \kappa (\gamma (\| u_k^0 - \tilde{u}_k^0 \| + \varepsilon')_{k=1}^{h-1}) \right] + H\delta
$$
$$
+ \bar{\mathbb{P}}^{\pi^0, T, \tilde{\pi}^0} \left( \{ \exists h, \| \tilde{u}_h^0 - u_h^0 \| > d - \varepsilon' \} \right) + \mathrm{TV}(\bar{\mathbb{P}}^{\pi^0, T, \tilde{\pi}^0}, \bar{\mathbb{P}}^{\pi^1, T, \tilde{\pi}^1}). \tag{10}
$$

*Proof.* We construct a coupling $\bar{\mu}$ of $\bar{\mathbb{P}}^{\pi^0, T, \tilde{\pi}^0}, \bar{\mathbb{P}}^{\pi^1, T, \tilde{\pi}^1}, \mathbb{P}^{\pi^2, T}$ as follows. Let $\mu^{0,1}$ be the maximal coupling between $\bar{\mathbb{P}}^{\pi^0, T, \tilde{\pi}^0}, \bar{\mathbb{P}}^{\pi^1, T, \tilde{\pi}^1}$. By Lemma 21, we know $\mu^{0,1}$ can be a shared-noise coupling. Now we construct the coupling $\mu^{12}$ between $\mathbb{P}^{\pi^1, T, \tilde{\pi}^1}$ and $\mathbb{P}^{\pi^2, T}$: first we let $\mu^{1,2}$ to be a shared-noise coupling, then for any $(x_h^1, x_h^2) \in \mathcal{X}^2$, by Lemma 19, there exists a distribution $\Gamma_h(d\tilde{u}_h^2, du' | x_h^1, x_h^2)$ that attains the infimum of the RTVC probability (remind readers again that $\pi^2 = \tilde{\pi}^1$). Now conditioning on $x_h^1$, consider another coupling of $u_h^1, \tilde{u}_h^1$ conditioning on $x_h^1$ as $L_h(du_h^1, d\tilde{u}_h^1 | x_h^1)$, which is the conditional distribution of $u_h^1, \tilde{u}_h^1 \mid x_h^1$ in $\bar{\mathbb{P}}^{\pi^1, T, \tilde{\pi}^1}$. Then we use the glueing lemma to glue $\Gamma_h$ and $L_h$ to construct a joint distribution of $u_h^1, \tilde{u}_h^1, u_h^2$ given $x_h^1, x_h^2$ such that $\tilde{u}_h^1, u_h^2 | x_h^1, x_h^2 \sim \Gamma_h(\cdot | x_h^0, x_h^1)$ and $u_h^1, \tilde{u}_h^1 | x_h^1 \sim L_h(\cdot, \cdot | x_h^1)$. By doing this for every $h$, now we have specified the whole distribution of $\mu^{1,2}$(since the distribution is determined by the noise distribution and action distribution). Finally, we use gluing lemma again to glue $\mu^{0,1}$ and $\mu^{1,2}$ and get $\bar{\mu}$. Notice that under $\bar{\mu}$ all three trajectories share noise. Therefore the marginal of $\mathbb{P}^{\pi^0, T, \tilde{\pi}^0}$ and $\mathbb{P}^{\pi^2, T}$, denoted with $\mu$, is also a shared-noise coupling.

Let $\mathcal{F}_{\mathrm{state}_h} := \sigma(x_{1:h}^0, x_{1:h}^1, x_{1:h}^2, u_{1:h-1}^0, \tilde{u}_{1:h-1}^0, u_{1:h-1}^1, \tilde{u}_{1:h-1}^1, u_{1:h-1}^2)$, define $A_h = \{ \| \tilde{u}_h^1 - u_h^2 \| \leq \varepsilon' \}$. Then by the construction of $\Gamma_h$, we have,

$$
\bar{\mu} \left( A_h^c | \mathcal{F}_{\mathrm{state}_h} \right) \leq \kappa (\| x_h^1 - x_h^2 \|), \bar{\mu} \text{ a.s.}
$$

Then notice that,

$$
\bar{\mu} \left( \bigcup_{h=1}^H A_h^c \cap (\bigcap_{k=1}^H \| \tilde{u}_k^0 - u_k^0 \| \leq d - \varepsilon') \right) \leq \underbrace{\bar{\mu} \left( \bigcup_{h=1}^H A_h^c \cap (\bigcap_{k=1}^H \| \tilde{u}_k^0 - u_k^0 \| \leq d - \varepsilon') \cap \{ \bar{\tau}^0 = \bar{\tau}^1 \} \right)}_{(1)} + \underbrace{\mu^{0,1}(\bar{\tau}^0 \neq \bar{\tau}^1)}_{= \mathrm{TV}(\bar{\mathbb{P}}^{\pi^0, T, \tilde{\pi}^0}, \bar{\mathbb{P}}^{\pi^1, T, \tilde{\pi}^1})}
$$

$$
(1) \leq \bar{\mu} \left( \bigcup_{h=1}^H (\cap_{t=1}^{h-1} A_t \cap A_h^c) \cap (\bigcap_{k=1}^H \| \tilde{u}_k^0 - u_k^0 \| \leq d - \varepsilon') \cap \{ \bar{\tau}^0 = \bar{\tau}^1 \} \right)
$$

$$
\leq \sum_{h=1}^H \bar{\mu} \left( (\cap_{t=1}^{h-1} A_t \cap A_h^c) \cap (\bigcap_{k=1}^H \| \tilde{u}_k^0 - u_k^0 \| \leq d - \varepsilon') \cap \{ \bar{\tau}^0 = \bar{\tau}^1 \} \right)
$$

$$
\leq \sum_{h=1}^H [\bar{\mu} \left( (\cap_{t=1}^{h-1} A_t \cap A_h^c) \cap \Phi_{h-1} \cap \Psi_{h-1} \cap \{ \bar{\tau}^0 = \bar{\tau}^1 \} \right) + \bar{\mu} \left( (\cap_{t=1}^{h-1} A_t \cap A_h^c) \cap \Phi_{h-1} \cap \Psi_{h-1}^c) \cap \{ \bar{\tau}^0 = \bar{\tau}^1 \} \right]
$$

$$
\leq \sum_{h=1}^H \left[ \bar{\mu} \left( (\cap_{t=1}^{h-1} A_t \cap A_h^c) \cap \Phi_{h-1} \cap \Psi_{h-1} \cap \{ \bar{\tau}_{:x_h^0}^0 = \bar{\tau}_{:x_h^1}^1 \} \right) \right] + \sum_{h=1}^H \mu(\Phi_{h-1} \cap \Psi_{h-1}^c),
$$

where we abuse the notation to denote,

$$
\Phi_h = \{ \| u_t^0 - u_t^2 \| \leq d, \forall t \in [h] \}, \Psi_h = \{ \| x_{t+1}^0 - x_{t+1}^2 \| \leq \gamma (\| u_k^0 - u_k^2 \|)_{k=1}^t, \forall t \in [h] \}.
$$

Above, the second inequality is a standard decomposition of $\cap A_h^c$. The thrid inequality is because the events are disjoint. The fourth equation is by $\cap_{t=1}^{h-1} A_t \cap (\cap_{k=1}^{h-1} \{ \| \tilde{u}_k^0 - u_k^0 \| \leq d - \varepsilon' \}) \cap \{ \bar{\tau}^0 = \bar{\tau}^1 \}$ implies $\Phi_{h-1}$. Then the second part above is bounded by $H\delta$ (since the marginal of $(\tau^0, \tau^2)$ is a shared-noise coupling).

And the first term above is further bounded by

$$
\bar{\mu} \left( (\cap_{t=1}^{h-1} A_t \cap A_h^c) \cap \Phi_{h-1} \cap \Psi_{h-1} \cap \{ \bar{\tau}_{:x_h^0}^0 = \bar{\tau}_{:x_h^1}^1 \} \right)
$$

$$= \mathbb{E}_{\bar{\mu}} \left[ \mathbb{I}((\cap_{t=1}^{h-1} A_t) \cap \Phi_{h-1} \cap \Psi_{h-1} \cap \{\bar{\tau}^0_{:x^0_h} = \bar{\tau}^1_{:x^1_h}\}) \cdot \mathbb{E}[\mathbb{I}(A^c_h) \mid \mathcal{F}_{\text{state}_h}] \right]$$

$$\leq \mathbb{E}_{\bar{\mu}} \left[ \mathbb{I}((\cap_{t=1}^{h-1} A_t) \cap \Phi_{h-1} \cap \Psi_{h-1} \cap \{\bar{\tau}^0_{:x^0_h} = \bar{\tau}^1_{:x^1_h}\}) \cdot \kappa(\|x^1_h - x^2_h\|) \right]$$

$$\leq \mathbb{E}_{\bar{\mu}} [\mathbb{I}((\cap_{t=1}^{h-1} A_t) \cap \Phi_{h-1} \cap \Psi_{h-1} \cap \{\bar{\tau}^0_{:x^0_h} = \bar{\tau}^1_{:x^1_h}\}) \cdot \kappa(\gamma(\|u^1_k - u^2_k\|)_{k=1}^{h-1})]$$

$$\leq \mathbb{E}_{\bar{\mu}} \left[ \mathbb{I}(\{\bar{\tau}^0_{:x^0_h} = \bar{\tau}^1_{:x^1_h}\}) \kappa(\gamma(\|u^1_k - \tilde{u}^1_k\| + \varepsilon')_{k=0}^{h-1}) \right]$$

$$\leq \mathbb{E}_{\mu} \left[ \kappa(\gamma(\|u^0_k - \tilde{u}^0_k\| + \varepsilon')_{k=0}^{h-1}) \right].$$

Here the first equation is since $(\cap_{t=1}^{h-1} A_t) \cap \Phi_{h-1} \cap \Psi^c_{h-1} \cap \{\bar{\tau}^0_{:x^0_h} = \bar{\tau}^1_{:x^1_h}\}$ is $\mathcal{F}_{\text{state}_h}$-measurable. And the first inequality is due to the RTVC property. The second inequality is because $\Psi_{h-1}$ is true, so we use the stability condition given by $\Psi_{h-1}$. For the last but not least inequality, it is due to that $\cap_{t=1}^{h-1} A_t$ holds, and we use the triangle inequality. For the last inequality, we use the property that $\{\bar{\tau}^0_{:x^0_h} = \bar{\tau}^1_{:x^1_h}\}$ to replace trajectory 0 with trajectory 1. And notice that the final expectation is only taken on $\mu$ since it only involves trajectory 0 and 2. Finally, we plug them all in one inequality, and marginalize $\mu$ to $\bar{\mathbb{P}}^{\pi^0, T, \tilde{\pi}^0}$ if the corresponding term only involves the zero trajectory. We get,

$$\mu \left( \{\cup_{h=1}^H \|\tilde{u}^0_h - u^2_h\| > \varepsilon'\} \right) = \bar{\mu} \left( \{\cup_{h=1}^H \|\tilde{u}^0_h - u^2_h\| > \varepsilon'\} \right)$$

$$\leq \bar{\mu} \left( \{\cup_{h=1}^H \|\tilde{u}^0_h - u^2_h\| > \varepsilon'\} \cap (\cap_{k=1}^H \{\|\tilde{u}^0_k - u^0_k\| \leq d - \varepsilon'\}) \right) + \mu \left( \cup_{k=1}^H \{\|\tilde{u}^0_k - u^0_k\| > d - \varepsilon'\} \right)$$

$$\leq \mathbb{E}_{\bar{\mathbb{P}}^{\pi^0, T, \tilde{\pi}^0}} \left[ \sum_{h=1}^H \kappa(\gamma(\|u^0_k - \tilde{u}^0_k\| + \varepsilon')_{k=1}^{h-1}) \right] + H\delta + \text{TV}(\bar{\mathbb{P}}^{\pi^0, T, \tilde{\pi}^0}, \bar{\mathbb{P}}^{\pi^1, T, \tilde{\pi}^1})$$

$$+ \bar{\mathbb{P}}^{\pi^0, T, \tilde{\pi}^0} \left( \exists h \in H, \|\tilde{u}_h - u^0_h\| > d - \varepsilon' \right). \qquad \square$$

Now we restate Theorem 1 and then prove it with the lemma above.

**Theorem 1.** *Consider two pairs of policies $\pi^0, \tilde{\pi}^0$ and $\pi^1, \tilde{\pi}^1$ and the associated extended trajectory laws $\bar{\mathbb{P}}^{\pi^0, T, \tilde{\pi}^0}$ and $\bar{\mathbb{P}}^{\pi^1, T, \tilde{\pi}^1}$. Assume $\mathbb{P}^{\pi^0, T}$ is $(\gamma, \delta)$-d-locally P-IISS and $\tilde{\pi}^1$ is $\varepsilon'$-RTVC with modulus $\kappa$. Let $\pi^2 := \tilde{\pi}^1$. If the reward function is $L_r$-Lipschitz, then it holds that*

$$\left| J(\pi^0) - J(\pi^2) \right| \lesssim H^2 \delta + H \cdot \text{TV}(\bar{\mathbb{P}}^{\pi^0, T, \tilde{\pi}^0}, \bar{\mathbb{P}}^{\pi^1, T, \tilde{\pi}^1})$$

$$+ H \cdot \bar{\mathbb{P}}^{\pi^0, T, \tilde{\pi}^0} \left( \exists h \in [H], \|\tilde{u}_h - u^0_h\| > d - \varepsilon' \right)$$

$$+ H \cdot \mathbb{E}_{\bar{\mathbb{P}}^{\pi^0, T, \tilde{\pi}^0}} \left\{ \sum_{h=1}^H \left[ \kappa(\gamma((\|u^0_k - \tilde{u}^0_k\| + \varepsilon')_{k=1}^{h-1})) \right. \right.$$

$$\left. \left. + \frac{1}{H} L_r \left( \gamma((\|u^0_k - \tilde{u}^0_k\| + \varepsilon')_{k=1}^{h-1}) + \|u^0_h - \tilde{u}^0_h\| + \varepsilon' \right) \right] \right\}.$$

*Proof.* Let $\mu$ be the desired shared noise coupling between $\bar{\mathbb{P}}^{\pi^0, T, \tilde{\pi}^0}$ and $\mathbb{P}^{\pi^2, T}$ (existence guaranteed by Lemma 22). Again we abuse the notation to denote,

$$\Phi_h = \{\|u^0_t - u^2_t\| \leq d, \forall t \in [h]\}, \Psi_h = \{\|x^0_{t+1} - x^2_{t+1}\| \leq \gamma(\|u^0_k - u^2_k\|)_{k=1}^t, \forall t \in [h]\}.$$

Notice that ,

$$\mathbb{E}_{\mu} \left\{ [\sum_{h=1}^H r(x^0_h, u^0_h) - r(x^2_h, u^2_h)] \mathbb{I} \left( \underset{h=1}{\overset{H}{\cap}} \{\|\tilde{u}^0_h - u^2_h\| \leq \varepsilon'\} \cap \underset{h=1}{\overset{H}{\cap}} \{\|\tilde{u}^0_h - u^0_h\| \leq d - \varepsilon'\} \right) \right\}$$

$$\leq \mathbb{E}_{\mu} \left\{ [\sum_{h=1}^H r(x^0_h, u^0_h) - r(x^2_h, u^2_h)] \mathbb{I} \left( \underset{h=1}{\overset{H}{\cap}} \{\|\tilde{u}^0_h - u^2_h\| \leq \varepsilon'\} \cap \Phi_H \right) \right\}$$

$$\leq \mathbb{E}_{\mu} \left\{ [\sum_{h=1}^H r(x^0_h, u^0_h) - r(x^2_h, u^2_h)] \mathbb{I} \left( \underset{h=1}{\overset{H}{\cap}} \{\|\tilde{u}^0_h - u^2_h\| \leq \varepsilon'\} \cap \Phi_H \cap \Psi_H \right) \right\} + H \cdot \mu(\Phi_H \cap \Psi^c_H)$$

$$\leq \mathbb{E}_\mu \left\{ [\sum_{h=1}^H L_r(\|x_h^0 - x_h^2\| + \|u_h^0 - u_h^2\|)] \mathbb{I} \left( \mathop{\cap}_{h=1}^H \{\|\tilde{u}_h^0 - u_h^2\| \leq \varepsilon'\} \cap \Psi_H \right) \right\} + H\delta$$

$$\leq \mathbb{E}_{\bar{\mathbb{P}}^{\pi^0, T, \pi^1}} \left\{ \sum_{h=1}^H L_r(\gamma(\|u_t^0 - \tilde{u}_t^0\| + \varepsilon')_{t=1}^{h-1} + \|u_h^0 - \tilde{u}_h^0\| + \varepsilon') \right\} + H\delta,$$

where in the last inequality, we use the stability condition $\Psi_H$, and we upper bound each $\|u_t^0 - u_t^2\|$ by $\|u_t^0 - \tilde{u}_t^0\| + \varepsilon'$, since $\cap_{h=1}^H \{\|\tilde{u}_h^0 - u_h^2\| \leq \varepsilon'\}$ holds.

At the meantime, we have,

$$\mathbb{E}_\mu \left\{ [\sum_{h=1}^H r(x_h^0, u_h^0) - r(x_h^2, u_h^2)] \mathbb{I} \left( \mathop{\cup}_{h=1}^H \{\|\tilde{u}_h^0 - u_h^2\| > \varepsilon'\} \cup \mathop{\cup}_{h=1}^H \{\|\tilde{u}_h^0 - u_h^0\| > d - \varepsilon'\} \right) \right\}$$

$$\leq H \cdot \left[ \mu(\mathop{\cup}_{h=1}^H \{\|\tilde{u}_h^0 - u_h^2\| > \varepsilon'\}) + \bar{\mathbb{P}}^{\pi^0, T, \pi^1} \left( \{\exists h, \|\tilde{u}_h^0 - u_h^0\| > d - \varepsilon'\} \right) \right]$$

$$\leq H \cdot \mathbb{E}_{\bar{\mathbb{P}}^{\pi^0, T, \pi^1}} \left[ \sum_{h=1}^H \kappa(\gamma(\|u_k^0 - \tilde{u}_k^0\| + \varepsilon')_{k=1}^{h-1}) \right] + H\delta + H \cdot \text{TV}(\bar{\mathbb{P}}^{\pi^0, T, \tilde{\pi}^0}, \bar{\mathbb{P}}^{\pi^1, T, \tilde{\pi}^1})$$

$$+ H \cdot \bar{\mathbb{P}}^{\pi^0, T, \pi^1} \left( \exists h \in H, \|\tilde{u}_h - u_h^0\| > d - \varepsilon' \right). \qquad \square$$

### C.3. Proof of Theorem 2, Theorem 3 and Proposition 4

**Proof of Theorem 2 and Theorem 3.**

*Proof.* Those two bounds are direct result of Theorem 1 by plugging in the definition of quantization error into specific modulus $\kappa$ and $\gamma$, meanwhile the irrelevant terms will vanish because of global stability assumption. Finally, we apply Lemma 14,

$$\text{TV}(\mathbb{P}^{\pi^*, T, q\#\pi^*}, \mathbb{P}^{(\rho \circ \hat{\pi}), T, \hat{\pi}}) = \text{TV}(\mathbb{P}^{q\#\pi^*, (T \circ \rho)}, \mathbb{P}^{\hat{\pi}, (T \circ \rho)}).$$

And we use the statistical guarantee which gives us,

$$\text{TV}(\mathbb{P}^{q\#\pi^*, (T \circ \rho)}, \mathbb{P}^{\hat{\pi}, (T \circ \rho)}) \leq \frac{\log |\Pi| \delta^{-1}}{n}, \text{ if } q\#\pi^* \text{ deterministic,}$$

$$\text{TV}(\mathbb{P}^{q\#\pi^*, (T \circ \rho)}, \mathbb{P}^{\hat{\pi}, (T \circ \rho)}) \leq \sqrt{\frac{\log |\Pi| \delta^{-1}}{n}}, \text{ if } q\#\pi^* \text{ stochastic,}$$

Plug this in, we have proved the desired upper bound. $\qquad \square$

**Proof of Proposition 4.**

*Proof.* This is a direct corollary of Theorem 1, by using $\pi^1 = \pi^0 = \pi^*, \tilde{\pi}^1 = \tilde{\pi}^0 = \hat{\pi}$. $\qquad \square$

## D. Proofs from Section 4

### D.1. Proof of Proposition 5

*Proof.* **Proof of (i).** Notice that if the condition holds, then for $x, x'$ such that $\|x - x'\| < \delta_0$, we have,

$$W_{c_{\varepsilon'}} (q\#\pi_h(\cdot \mid x), q\#\pi_h(\cdot | x')) = \mathbf{1}\{\|q(\pi_h(x)) - q(\pi_h(x'))\| > \varepsilon'\} \leq \mathbf{1}(\|x - x'\| > \delta_0) = 0.$$

Therefore it must holds that,

$$\forall \|x - x'\| \leq \delta_0, \|q(\pi_h(x)) - q(\pi_h(x'))\| \leq \varepsilon'$$

**Proof of (ii).** Fix any $h \in [H]$ and any $x, x' \in \mathcal{X}$. By the triangle inequality,

$$\|q(\pi_h(x)) - q(\pi_h(x'))\| \leq \|q(\pi_h(x)) - \pi_h(x)\| + \|\pi_h(x) - \pi_h(x')\| + \|\pi_h(x') - q(\pi_h(x'))\|.$$

Since $q$ is a binning quantizer, it satisfies $\|q(u) - u\| \leq \varepsilon_q$ for all $u \in \mathcal{U}$, and since $\pi_h(\cdot)$ is $L$-Lipschitz, we have

$$\|q(\pi_h(x)) - q(\pi_h(x'))\| \leq 2\varepsilon_q + L\|x - x'\|.$$

If $\|x - x'\| \leq \delta_0$, then by the definition $\delta_0 = (\varepsilon' - 2\varepsilon_q)/L$,

$$\|q(\pi_h(x)) - q(\pi_h(x'))\| \leq 2\varepsilon_q + L\delta_0 = 2\varepsilon_q + (\varepsilon' - 2\varepsilon_q) = \varepsilon',$$

and hence

$$W_{c_{\varepsilon'}}\left(q\#\pi_h(\cdot \mid x), q\#\pi_h(\cdot \mid x')\right) = \mathbf{1}\left\{\|q(\pi_h(x)) - q(\pi_h(x'))\| > \varepsilon'\right\} = 0 \leq \kappa(\|x - x'\|).$$

If $\|x - x'\| > \delta_0$, then $\kappa(\|x - x'\|) = 1$ and trivially $W_{c_{\varepsilon'}}(\cdot, \cdot) \leq 1 = \kappa(\|x - x'\|)$. Combining the two cases, we conclude that $q\#\pi^*$ is $\varepsilon'$-RTVC with modulus $\kappa(r) = \mathbf{1}\{r > \delta_0\}$. $\qquad\square$

## D.2. Proof of Theorem 6

**Proof of Theorem 6, deterministic dynamic part.** We use a seemingly very benign scalar dynamic, policy and reward: let $f(x, u) = Ax + Bu$, where $A, B > 0$ and $\lambda := A + B < 1$. The deterministic expert policy is $\pi^*(x) = \arctan(x)$ with reward function $r(x, u) = 1 - |u - \arctan(x)|$. The initial distribution is $X_1 \sim \mathcal{N}(0, 1)$. In this whole proof we will denote this homogeneous Markov chain by $X_t$.

We specify three non-overlapping intervals, let $d > \frac{k}{2} > \frac{B}{1-\lambda}, Ak < 1$ (e.g. $A = 0.2, B = 0.3, k = 1, d = 0.6$ works),

$$I_P = \left[-\frac{k\varepsilon_q}{2}, \frac{k\varepsilon_q}{2}\right], \qquad I_{T_1} = [d\varepsilon_q, (d+1)\varepsilon_q], \qquad I_{T_2} = [Ad\varepsilon_q + B, \ A(d+1)\varepsilon_q + B].$$

**Claim 1.** *For any $t \geq 1$,*

$$\mathbb{E}_{\mathbb{P}^{\pi^*, T}}\left[\mathbb{I}(X_t \in I_{T_1})\right] \leq \frac{1}{\sqrt{2\pi}A^{t-1}} \exp\left(-\frac{d^2\varepsilon_q^2}{2\lambda^{2(t-1)}}\right) \cdot \varepsilon_q,$$

$$\mathbb{E}_{\mathbb{P}^{\pi^*, T}}\left[\mathbb{I}(X_t \in I_{T_2})\right] \leq \frac{1}{\sqrt{2\pi}A^{t-1}} \exp\left(-\frac{B^2}{2\lambda^{2(t-1)}}\right) \cdot A\varepsilon_q.$$

*Proof.* Under the expert policy $u_t = \pi^*(X_t) = \arctan(X_t)$, the closed-loop system is the deterministic recursion

$$X_{t+1} = f^{\pi^*}(X_t), f^{\pi^*}(x) := Ax + B\arctan(x),$$

with random initialization $X_1 \sim \mathcal{N}(0, 1)$. Hence for $t \geq 1$,

$$X_t = (f^{\pi^*})^{\circ(t-1)}(X_1).$$

Since $A, B > 0$, we have

$$(f^{\pi^*})'(x) = A + \frac{B}{1 + x^2} \geq A \forall x \in \mathbb{R},$$

so $f^{\pi^*}$ is strictly increasing and $C^1$, hence invertible with $C^1$ inverse. Therefore $X_t$ admits a density $p_t$, and the change-of-variables formula yields, for $t \geq 1$,

$$p_t(y) = \frac{p_1\left((f^{\pi^*})^{-(t-1)}(y)\right)}{\left|\left((f^{\pi^*})^{\circ(t-1)}\right)'\left((f^{\pi^*})^{-(t-1)}(y)\right)\right|},$$

where $p_1(z) = \frac{1}{\sqrt{2\pi}} e^{-z^2/2}$ is the standard normal density. By the chain rule, for $t \geq 1$,

$$\left( (f^{\pi^*})^{\circ(t-1)} \right)'(x) = \prod_{s=0}^{t-2} (f^{\pi^*})' \left( (f^{\pi^*})^{\circ s}(x) \right) \geq A^{t-1},$$

hence

$$p_t(y) \leq \frac{1}{A^{t-1}} p_1 \left( (f^{\pi^*})^{-(t-1)}(y) \right).$$

Next, using $|\arctan(x)| \leq |x|$, we obtain

$$\left| f^{\pi^*}(x) \right| \leq A|x| + B|\arctan(x)| \leq (A+B)|x| = \lambda|x|, \lambda := A + B < 1.$$

Let $x = (f^{\pi^*})^{-1}(y)$. Then $|y| = |f^{\pi^*}(x)| \leq \lambda|x|$, so $|(f^{\pi^*})^{-1}(y)| \geq |y|/\lambda$, and iterating gives, for $t \geq 1$,

$$\left| (f^{\pi^*})^{-(t-1)}(y) \right| \geq \frac{|y|}{\lambda^{t-1}}.$$

Since $p_1$ is decreasing in $|z|$, it follows that

$$p_1 \left( (f^{\pi^*})^{-(t-1)}(y) \right) \leq \frac{1}{\sqrt{2\pi}} \exp\left( -\frac{y^2}{2\lambda^{2(t-1)}} \right),$$

and therefore, for all $y \in \mathbb{R}$ and $t \geq 1$,

$$p_t(y) \leq \frac{1}{\sqrt{2\pi} A^{t-1}} \exp\left( -\frac{y^2}{2\lambda^{2(t-1)}} \right).$$

For any interval $J$, we have

$$\mathbb{E}_{\pi^*}\left[ \mathbb{I}(X_t \in J) \right] = \mathbb{P}(X_t \in J) = \int_J p_t(y) dy \leq |J| \cdot \sup_{y \in J} p_t(y).$$

For $I_{T_1} = [d\varepsilon_q, (d+1)\varepsilon_q]$, $|I_{T_1}| = \varepsilon_q$ and $y \geq d\varepsilon_q$ for all $y \in I_{T_1}$, hence

$$\sup_{y \in I_{T_1}} p_t(y) \leq \frac{1}{\sqrt{2\pi} A^{t-1}} \exp\left( -\frac{d^2 \varepsilon_q^2}{2\lambda^{2(t-1)}} \right),$$

which implies

$$\mathbb{E}_{\pi^*}\left[ \mathbb{I}(X_t \in I_{T_1}) \right] \leq \frac{1}{\sqrt{2\pi} A^{t-1}} \exp\left( -\frac{d^2 \varepsilon_q^2}{2\lambda^{2(t-1)}} \right) \cdot \varepsilon_q.$$

Similarly, for $I_{T_2} = [Ad\varepsilon_q + B,\ A(d+1)\varepsilon_q + B]$, we have

$$\mathbb{E}_{\pi^*}\left[ \mathbb{I}(X_t \in I_{T_2}) \right] \leq \frac{1}{\sqrt{2\pi} A^{t-1}} \exp\left( -\frac{B^2}{2\lambda^{2(t-1)}} \right) \cdot A\varepsilon_q.$$

This proves the claim. $\qquad\square$

We define the quantizer on those intervals:

$$q(\pi^*(x)) := \begin{cases} \frac{(d+\frac{1}{2})}{B} \varepsilon_q, & x \in I_P, \\ 1, & x \in I_{T_1}, \\ \frac{(d+1)(1-A^2)}{B} \varepsilon_q - A, & x \in I_{T_2}, \\ \pi^*(x) + \delta(x), & \text{otherwise}, \end{cases}$$

with $|\delta(x)| < \varepsilon_q$. We let $\delta(x) < 0, x \in \mathbb{R}_{<0} \setminus I_P$. Let $q_0 := \frac{B}{1-\lambda}$ and $\rho_t := \frac{\frac{k}{2} - q_0(1-\lambda^{t-1})}{\lambda^{t-1}}$, and $I_t = [-\rho_t \varepsilon_q, 0]$.

**Claim 2.** $X_1 \in I_t \Rightarrow X_t \in I_P \cup I_{T_1} \cup I_{T_2}$.

*Proof.* First notice that under the quantizer $q$, we have $\forall s$,

$$X_s \in I_P \cup I_{T_1} \cup I_{T_2}, \Rightarrow X_t \in I_{T_1} \cup I_{T_2}, t > s$$
$$X_s < 0, X_s \notin I_P \Rightarrow X_{s+1} < 0,$$

where the second equation is since $A, B > 0, \delta(x) < 0$. Therefore for $X_1 \in [-\rho_t \varepsilon_q, 0)$, if $\exists s < t, X_s \in I_P \cup I_{T_1} \cup I_{T_2}$, then $X_t \in I_P \cup I_{T_1} \cup I_{T_2}$. Otherwise, since $X_s < 0, X_s \notin I_P, \forall s < t$, it holds that,

$$X_{s+1} = Ax_s + B \arctan(X_s) + B\delta(X_s)$$
$$\geq (A + B)X_s - B\varepsilon_q = \lambda X_s - B\varepsilon_q$$
$$\Rightarrow X_t \geq \lambda^{t-1} X_1 - \frac{B}{1-\lambda}(1 - \lambda^{t-1})\varepsilon_q = \lambda^{t-1} X_1 - q_0(1 - \lambda^{t-1})\varepsilon_q \geq \frac{-k}{2}\varepsilon_q.$$

Meanwhile $X_t < 0$, this means $X_t \in I_P$. Thus the claim is proved. $\qquad\square$

From Claim 2, we also know that $X_1 \in I_t \Rightarrow X_{t+1} \in I_{T_1} \cup I_{T_2}$. Therefore we have,

$$\mathbb{E}_{q\#\pi^*}\left[\mathbb{I}(X_{t+1} \in I_{T_1} \cup I_{T_2})\right] = \mathbb{P}^{q\#\pi^*}\left(X_{t+1} \in I_{T_1} \cup I_{T_2}\right) \geq \mathbb{P}(X_1 \in I_t) = \Phi(\rho_t \varepsilon_q) - \tfrac{1}{2}, t \geq 1.$$

Now notice that we have,

$$\frac{1}{H}\sum_{h=1}^{H} \mathbb{E}_{\mathbb{P}^{\pi^*,\tau}}\left[\|u_h^* - q(u_h^*)\|\right] = \frac{1}{H}\sum_{h=1}^{H} \mathbb{E}_{\pi^*}\left[|\delta(X_h^*)|(\mathbb{I}(I_{T_1} \cup I_{T_2}) + \mathbb{I}(I_{T_1}^c \cap I_{T_2}^c))\right]$$

$$\overset{\text{Claim 1}}{\leq} \frac{\Theta(1)}{H}\sum_{h=1}^{H}\left(\varepsilon_q + \frac{1}{\sqrt{2\pi}A^h}\exp(-\frac{d^2\varepsilon_q^2}{2\lambda^{2h}})\cdot \varepsilon_q + \frac{1}{\sqrt{2\pi}A^h}\exp(-\frac{B^2}{2\lambda^{2h}})\cdot A\varepsilon_q\right)$$

$$\lesssim \varepsilon_q + \Theta(\frac{\varepsilon_q |\log \varepsilon_q|}{H}) = O(\varepsilon_q), \text{ when } H = \omega(|\log \varepsilon_q|).$$

Finally, notice that on $I_{T_1} \cup I_{T_2}$, the quantization error is at least $A$, therefore,

$$J(\pi^*) - J(q\#\pi^*) = \sum_{h=1}^{H} \mathbb{E}_{q\#\pi^*}\left[|\delta(x_h)|\right]$$

$$\geq \sum_{h=2}^{H} A\left(\Phi(\rho_{h-1}\varepsilon_q) - \frac{1}{2}\right) \gtrsim (H - \Theta(|\log \varepsilon_q|)) = O(H), \text{ when } H = \omega(|\log \varepsilon_q|).$$

$$\qquad\square$$

**Proof of Theorem 6, stochastic dynamic part.** **Setup** The initial distribution is $T_0(\emptyset) = N(0,1)$. Consider a scaler linear system $f(x, u) = Ax + Bu$, the noisy dynamic is $x_{t+1} = f(x_t, u_t) + \omega_t, \omega_t \sim \mathcal{N}(0, \sigma_\omega^2)$. The policy is $\pi^*(x) = \arctan(x)$. Now we define three intervals as,

$$I_P = [-k\varepsilon_q, k\varepsilon_q], I_{T_1} = [2k\varepsilon_q, (2k + 2Ak)\varepsilon_q], I_{T_2} = [B + 2Ak\varepsilon_q, B + A(2k + 2Ak)\varepsilon_q].$$

We design the quantizer to be,

$$q(\pi^*(x)) = \begin{cases} \frac{(2+A)k\varepsilon_q}{B}, & x \in I_P, \\ 1, & x \in I_{T_1}, \\ -\frac{(1+2A^2)k\varepsilon_q}{B} - A, & x \in I_{T_2}, \\ \pi^*(x) + \delta(x), & \text{otherwise.} \end{cases}$$

By constructing such a quantizer , we can always maintain the loop $I_P \to I_{T_1} \to I_{T_2} \to I_P$ under noiseless dynamic. For the rest of the value, we set a binning quantizer such that $|\delta(x)| < c_\delta \varepsilon_q$.

We will denote $\lambda = A + B, I_T = I_{T_1} \cup I_{T_2}, I_A = I_T \cup I_P$. For a constant $R > 0$, define $C_R = [-R, R]$. Under the quantized expert, the state sequences will form a homogeneous Markov chain on a continuous space, we denote the chain with $(X_t)_{t \geq 0}$. The quantized policy induced transition is denoted with $T^{q\#\pi^*}$. Now we will first show that the stable distribution for this Markov chain exists, when deploying the quantized policy in the noisy dynamic.

**Claim 3.** *There exists a stable distribution on $\mathcal{X}$ denoted as $\nu(\cdot)$. It also holds the geometric convergence to the stable distribution*

$$\|d_t^{q\#\pi^*} - \nu\|_{\mathrm{TV}} \leq M\rho^t$$

*for a constant $M > 0$ and $\rho \in (0, 1)$, where $d_t^{q\#\pi^*}$ is the state distribution at step $t$ under policy $q\#\pi^*$.*

*Proof.* Define the Lyapunov function

$$V(x) := 1 + x^2.$$

Conditioning on $X_t = x$, using $\mathbb{E}[\omega_t] = 0$ and $\mathrm{Var}(\omega_t) = \sigma_\omega^2$,

$$T^{q\#\pi^*}V(x) := \mathbb{E}\big[V(X_{t+1}) \mid X_t = x\big] = 1 + \big(Ax + Bq(\pi^*(x))\big)^2 + \sigma_\omega^2.$$

Choose $R$ large enough so that $I_A \subset \mathrm{int}(C_R)$. For $x \notin C_R$, the binning quantizer satisfies $q(\pi^*(x)) = \pi^*(x) + \delta(x)$ with $|\delta(x)| < c_\delta \varepsilon_q$, hence

$$\big(Ax + Bq(\pi^*(x))\big)^2 \leq \big(Ax + B\pi^*(x)\big)^2 + 2Bc_\delta \varepsilon_q \big|Ax + B\pi^*(x)\big| + B^2 c_\delta^2 \varepsilon_q^2.$$

Using $|\arctan(x)| \leq \pi/2$ and $|Ax + B\pi^*(x)| \leq \lambda|x|$,

$$T^{q\#\pi^*}V(x) \leq 1 + A^2x^2 + \pi AB|x| + \tfrac{\pi^2}{4}B^2 + 2Bc_\delta \lambda \varepsilon_q |x| + O(\varepsilon_q^2) + \sigma_\omega^2.$$

Since $A^2 < \lambda$ (as $A^2 - A < 0 < B$ gives $A^2 < A + B = \lambda$), the right-hand side minus $\lambda V(x)$ equals

$$(A^2 - \lambda)x^2 + \big(\pi AB + 2Bc_\delta \lambda \varepsilon_q\big)|x| + \big(1 + \tfrac{\pi^2}{4}B^2 + \sigma_\omega^2 - \lambda\big) + O(\varepsilon_q^2),$$

which is a quadratic in $|x|$ with strictly negative leading coefficient $A^2 - \lambda < 0$, hence negative for all $|x|$ large enough. Thus, enlarging $R$ if necessary, we have $T^{q\#\pi^*}V(x) \leq \lambda V(x)$ for all $|x| > R$. Set

$$c := \sup_{x \in C_R} \big\{T^{q\#\pi^*}V(x) - \lambda V(x)\big\} < \infty,$$

where finiteness holds because $T^{q\#\pi^*}V$ is continuous and $C_R$ is compact. Then, for all $x \in \mathbb{R}$,

$$T^{q\#\pi^*}V(x) \leq \lambda V(x) + c\,\mathbf{1}_{C_R}(x), \tag{11}$$

which is a geometric Foster–Lyapunov drift condition towards the bounded set $C_R$.

Because the noise $\omega_t$ has a strictly positive continuous Gaussian density, the Markov chain admits a strictly positive continuous transition density. It follows by standard arguments that the chain is $\psi$-irreducible and aperiodic, and that $C_R$ is a small (petite) set. Combining these properties with the drift condition (11), by Theorem 15.0.1 of Meyn and Tweedie (2012) yields the result. $\square$

Another thing to notice is we take expectation w.r.t. $\nu(x)$ for Eq. (11), since $\nu T^{q\#\pi^*} = \nu$, then we have,

$$\nu(V) := \mathbb{E}_\nu[V(x)] \leq \lambda \nu(V) + c\nu(C_R)$$
$$\Rightarrow \nu(C_R) \geq \frac{(1 - \lambda)\nu(V)}{c} \geq \frac{1 - \lambda}{c}. \tag{12}$$

Now our goal is to lower bound the probability on $I_T$ under the stable distribution. To do so, we evaluate how many steps it take to move from outside of those intervals back. Define

$$\tau_{I_A}^X = \inf\{t \geq 0 : X_t \in I_A\}.$$

Define $\mathbb{P}_x$ as the probability measure on the chain $(X_t)_{t \geq 0}$ starting at $x$. We now evaluate $\sup_{x \in C_R \setminus I_A} \mathbb{P}_x(\tau_{I_A}^X \geq m)$. Our goal is to find an approriate $m$, such that this quantity can be upper bounded by a constant.

**Claim 4.** *Define* $K_\delta := \frac{Bc_\delta}{1-\lambda}$, $K_\omega := \frac{\sqrt{2/\pi}\,\sigma_\omega}{\varepsilon_q(1-\lambda)}$, *consider a constant* $u \in (0, k - K_\delta - K_\omega)$, *for* $T_{mix} :=$ $\lfloor \frac{\log \frac{R}{(k-u-(K_\delta+K_\omega))\varepsilon_q}}{|\log \lambda|} \rfloor_+ = \Theta\left(\log \frac{1}{\varepsilon_q}\right)$, *it holds that,*

$$\sup_{x \in C_R \setminus I_A} \mathbb{P}_x(\tau_{I_A}^X \geq T_{\text{mix}}) \leq 2\exp\left(-\frac{u^2\varepsilon_q^2(1-\lambda^2)}{2\sigma_\omega^2}\right) := \eta \tag{13}$$

*Proof.* Let us define,

$$R_t := \lambda^t R + (K_\delta + K_\omega)\varepsilon_q + S_t := r_t + S_t, \, S_t := \sum_{j=0}^{t-1} \lambda^j (|\omega_{t-1-j}|) - \mathbb{E}|\omega_0|).$$

Notice that under $\{\tau_{I_A}^x > T_{\text{mix}}\}$, it holds $|X_t| \leq R_t, t \leq T_{\text{mix}}$. And since $r_{T_{\text{mix}}} < (k-u)\varepsilon_q$, this means $S_{T_{\text{mix}}} \geq u\varepsilon_q$.

$$\forall x, P(\tau_{I_A}^x > T_{\text{mix}}) \leq P(r_{T_{\text{mix}}} + S_{\text{mix}} \geq k\varepsilon_q) \leq P(S_{T_{\text{mix}}} > u\varepsilon_q).$$

Since $|\omega_s|$ is a 1-Lipschitz function of $\omega_s \sim \mathcal{N}(0, \sigma_\omega^2)$, by Gaussian concentration, $|\omega_s| - \mathbb{E}|\omega_0|$ is sub-Gaussian with parameter $\sigma_\omega$. Hence $S_t = \sum_{j=0}^{t-1} \lambda^j(|\omega_{t-1-j}| - \mathbb{E}|\omega_0|)$ is sub-Gaussian with variance proxy $\sigma_\omega^2 \sum_{j=0}^{t-1} \lambda^{2j} \leq \frac{\sigma_\omega^2}{1-\lambda^2}$, and for any $t > 0$,

$$P(|S_t| \geq u\varepsilon_q) \leq 2\exp\left(-\frac{u^2\varepsilon_q^2}{2\sigma_\omega^2 \sum_{j=0}^{t-1} \lambda^{2j}}\right) \leq 2\exp\left(-\frac{u^2\varepsilon_q^2(1-\lambda^2)}{2\sigma_\omega^2}\right). \qquad \square$$

To make sure $\eta < 1$, we will require $\sigma_\omega = o(\varepsilon_q)$, so that so that $u$ appropriately selected can be positive, and the ratio $\frac{\varepsilon_q}{\sigma_\omega}$ in $\eta$ will not make it near 1. Since when $x \in I_A$, $\mathbb{P}_x(\tau_{I_A}^x \geq T_{\text{mix}}) = 0$, we have $\sup_{x \in C_R} \mathbb{P}_x(\tau_{I_A}^X \geq T_{\text{mix}}) \leq \eta$. Now we know that with at least a $O(1)$ probability, before $\Theta\left(\log(\frac{1}{\varepsilon_q})\right)$ steps, the state will travels back into $I_A$, entering the loop again. However this only holds for a bounded subset $C_R$. To further proceed, we will try to restrict the Markov chain to $C_R$ and use the new Markov chain to derive the result.

Now denote

$$\sigma_0 := \inf\{t \geq 0 : X_t \in C_R\}, \sigma_{n+1} = \inf\{t > \sigma_n : X_t \in C_R\}, Y_n := X_{\sigma_n}.$$

We know that $Y_n$ is a Markov Chain on $C_R$.

**Claim 5.** *$Y_n$ is positive recurrence with unique stable distribution,*

$$\mu(\cdot) = \frac{\nu(\cdot \cap C_R)}{\nu(C_R)}.$$

*Proof.* This is lemma 2 of Farago (2020) $\qquad \square$

Let $\mathbb{Q}_x$ denote the law of the censored chain $(Y_n)_{n\geq 0}$ with $Y_0 = x$.

**Claim 6.** *For any $x \in C_R$ and $m \geq 1$, it holds that*

$$\mathbb{Q}_x(\tau_{I_A}^Y \geq m) \leq \mathbb{P}_x(\tau_{I_A}^X \geq m).$$

*Proof.* If $x \in I_A$, then the inequality holds. If $x \notin I_A$, suppose $\tau_{I_A}^X < m$. Then there exists $0 \leq t < m$ such that $X_t \in I_A$. By the construction of the censored chain, there exists $l \geq 0$ such that the $l$-th visit of $(Y_t)$ corresponds to time $t$ of $(X_t)$. Hence,

$$\tau_{I_A}^Y \leq l \leq t < m.$$

Therefore,

$$\{\tau_{I_A}^X < m\} \subset \{\tau_{I_A}^Y < m\},$$

which implies the claim. $\qquad \square$

Taking $m = T_{\text{mix}}$, we obtain

$$\sup_{x \in C_R} \mathbb{Q}_x(\tau_{I_A}^Y \geq T_{\text{mix}}) \leq \eta.$$

Next, we bound $\mathbb{E}_x^{\mathbb{Q}}[\tau_{I_A}^Y]$ for $x \in C_R$. By the Markov property of $(Y_n)$, we have

$$\mathbb{Q}_x(\tau_{I_A}^Y \geq (m+1)T_{\text{mix}}) = \mathbb{E}_x^{\mathbb{Q}}\left[\mathbf{1}\{\tau_{I_A}^Y \geq mT_{\text{mix}}\}\mathbb{Q}_{Y_{mT_{\text{mix}}}}(\tau_{I_A}^Y \geq T_{\text{mix}})\right]$$
$$\leq \eta\mathbb{Q}_x(\tau_{I_A}^Y \geq mT_{\text{mix}}).$$

Iterating yields

$$\mathbb{Q}_x(\tau_{I_A}^Y \geq mT_{\text{mix}}) \leq \eta^m, m \geq 1.$$

Consequently, for $x \in C_R$

$$\mathbb{E}_x^{\mathbb{Q}}[\tau_{I_A}^Y] = \sum_{n \geq 0} \mathbb{Q}_x(\tau_{I_A}^Y \geq n) \leq T_{\text{mix}} \sum_{k \geq 0} \eta^k = \frac{T_{\text{mix}}}{1 - \eta}.$$

Finally, define the return time

$$\tau_{I_A}^+ := \inf\{n \geq 1 : Y_n \in I_A\}.$$

For $Y_0 \sim \mu(\cdot \mid I_A)$, it holds that

$$\mathbb{E}^{\mathbb{Q}}[\tau_{I_A}^+ \mid Y_0 \in I_A] \leq 1 + \sup_{x \in C_R} \mathbb{E}_x^{\mathbb{Q}}[\tau_{I_A}^Y] \leq 1 + \frac{T_{\text{mix}}}{1 - \eta}.$$

Then we use Theorem 10.4.9 of (Meyn and Tweedie, 2012) and get,

$$\mu(I_A) \geq \frac{1 - \eta}{1 - \eta + T_{\text{mix}}}.$$

Now we analyze the probabilistic structure inside $I_A$. We use $P(x, B)$ to denote the one-step transition probability starting at $x$ to $B$ for chain $(X_t)_{t \geq 0}$ and $Q(x, B)$ for chain $(Y_t)_{t \geq 0}$.

Let us define,

$$a := \sup_{x \in I_P} P(x, I_{T_1}^c) = P(k\varepsilon_q, I_{T_1}^c) = \frac{3}{2} - \Phi\left(\frac{2kA\varepsilon_q}{\sigma_\omega}\right), b := \sup_{x \in I_{T_1}} P(x, I_{T_2}^c) = \frac{3}{2} - \Phi\left(\frac{2kA^2\varepsilon_q}{\sigma_\omega}\right).$$

Since $\sigma_\omega \in o(\varepsilon_q)$, the ratio $\varepsilon_q/\sigma_\omega \to \infty$, so $\Phi\left(\frac{2kA\varepsilon_q}{\sigma_\omega}\right) \to 1$ and hence $1 - a, 1 - b \to \frac{1}{2}$. In particular, $a, b, 1 - a, 1 - b$ are all $\Theta(1)$, which is essential for the lower bound below not to degenerate.

Now we do a more fine-grained analysis to decompose $\mu(I_A)$ into $\mu(I_P)$ and $\mu(I_T)$ by constant $a, b$. First let us define,

$$a_Y := \sup_{x \in I_P} Q(x, I_{T_1}^c), b_Y := \sup_{x \in I_{T_1}} Q(x, I_{T_2}^c).$$

**Claim 7.**
$$a_Y \leq a, b_Y \leq b$$

*Proof.* This is proved as long as noticing that, $X_1 \in I_{T_1} \subset C_R \Rightarrow \sigma_1 = 1, Y_1 = X_1 \in I_{T_1}$. $\qquad \square$

Now let $p_0 := \mu(I_P), p_1 := \mu(I_{T_1}), p_2 := \mu(I_{T_2})$. Since $\mu$ is invariance to $Q$, we have,

$$p_1 = \int_{C_R} \mu(y)Q(y, I_{T_1})dy \geq \int_{I_P} \mu(y)Q(y, I_{T_1})dy \geq (1 - a_Y)p_0$$
$$p_2 \geq (1 - b_Y)p_1, p_0 + p_1 + p_2 = \mu(I_A).$$

We can solve this linear inequalities and have,

$$p_1 + p_2 \geq (1 - a_Y)p_0 + (1 - a_Y)(1 - b_Y)p_0$$
$$\Rightarrow p_1 + p_2 \geq (1 - a_Y)(2 - b_Y)[\mu(I_A) - (p_1 + p_2)]$$
$$\Rightarrow p_1 + p_2 \geq \mu(I_A)\frac{(1 - a_Y)(2 - b_Y)}{1 + (1 - a_Y)(2 - b_Y)} \geq \mu(I_A)\frac{(1 - a)(2 - b)}{1 + (1 - a)(2 - b)}.$$

Finally we plug in the lower bound of $\mu(I_A)$, then we get,

$$\nu(I_T|C_R) = \mu(I_T) \geq \frac{1 - \eta}{1 - \eta + T_{\text{mix}}}\frac{(1 - a)(2 - b)}{1 + (1 - a)(2 - b)}.$$

Finally, by the geometric ergodic property and utilizing Eq. (12) we have,

$$d_t^{q\#\pi^*}(I_T) \geq \nu(I_T) - M\rho^t = \Theta\left(\frac{1}{|\log(1/\varepsilon_q)|}\right) - M\rho^t$$

$$\frac{1}{H}\sum_{t=1}^{H}d_t^{q\#\pi^*}(I_T) \geq \Theta\left(\frac{1}{|\log(1/\varepsilon_q)|}\right) - M\frac{1}{H(1 - \rho)}.$$

Now we calculate the quantization error on the expert distribution (unquantized). We have the following claim.

**Claim 8.** *For all $t \geq 1$,*

$$\mathbb{P}\{X_t \in I_{T_1}\} \leq \exp\left(-\frac{2k^2\varepsilon_q^2}{\tau_t^2}\right), \tag{14}$$

*where*

$$\tau_t^2 := \lambda^{2(t-1)} + \sigma_\omega^2\frac{1 - \lambda^{2(t-1)}}{1 - \lambda^2}, t \geq 1.$$

*Proof.* Write $X_t = g_t(X_1, \omega_1, \ldots, \omega_{t-1})$ as a deterministic function of the independent Gaussians $X_1 \sim \mathcal{N}(0, 1)$ and $\omega_i \sim \mathcal{N}(0, \sigma_\omega^2)$. Since $f^{\pi^*}$ is $\lambda$-Lipschitz, the chain rule gives $|\partial X_t/\partial X_1| \leq \lambda^{t-1}$ and $|\partial X_t/\partial\omega_i| \leq \lambda^{t-1-i}$. Introduce the standardized variables $\tilde{X}_1 = X_1, \tilde{\omega}_i = \omega_i/\sigma_\omega$, all i.i.d. $\mathcal{N}(0, 1)$. The Lipschitz constant of the map $(\tilde{X}_1, \tilde{\omega}_1, \ldots, \tilde{\omega}_{t-1}) \mapsto X_t$ satisfies

$$L_t^2 = \lambda^{2(t-1)} + \sum_{i=1}^{t-1}\lambda^{2(t-1-i)}\sigma_\omega^2 = \tau_t^2.$$

By Gaussian concentration , $X_t - \mathbb{E}[X_t]$ is sub-Gaussian with proxy variance $\tau_t^2$. Since $f^{\pi^*}$ is an odd function and the noise is symmetric, $\mathbb{E}[X_t] = 0$ by induction. Therefore $\mathbb{P}\{X_t \geq r\} \leq \exp(-r^2/(2\tau_t^2))$ for every $r \geq 0$. Since $I_{T_1} \subset [2k\varepsilon_q, \infty)$, setting $r = 2k\varepsilon_q$ gives the claim. $\square$

We now show the in-distribution quantization error is $O(\varepsilon_q)$. First we will require $\sigma_\omega = O(\frac{\varepsilon_q}{|\log\varepsilon_q|^\alpha})$ with $\alpha > \frac{1}{2}$. Define

$$t_* := \min\left\{t \geq 1 : \exp\left(-\frac{2k^2\varepsilon_q^2}{\tau_t^2}\right) \leq \varepsilon_q\right\}.$$

Since $\tau_t^2$ is dominated by $\lambda^{2(t-1)}$ until the noise floor $\sigma_\omega^2/(1 - \lambda^2)$ takes over, we have $t_* = O(|\log\varepsilon_q|)$. For all $t \geq t_*$, $\tau_t^2 \leq \tau_{t_*}^2$, so the tail bound is at most $\varepsilon_q$.

Since $I_{T_2}$ is farther from the origin than $I_{T_1}$ and $X_t$ is symmetric around zero, $\mathbb{P}\{X_t \in I_{T_2}\} \leq \mathbb{P}\{X_t \in I_{T_1}\}$ for all $t$. Therefore,

$$\frac{1}{H}\sum_{t=1}^{H}\left[d_t^{\pi^*}(I_{T_1}) + d_t^{\pi^*}(I_{T_2})\right] \leq \frac{2t_*}{H} + \frac{1}{H}\sum_{t=t_*+1}^{H}2\exp\left(-\frac{2k^2\varepsilon_q^2}{\tau_t^2}\right)$$

$$\leq \frac{O(|\log\varepsilon_q|)}{H} + \varepsilon_q.$$

Taking $H = \Omega(|\log \varepsilon_q|/\varepsilon_q)$, the first term is also $O(\varepsilon_q)$, giving

$$\frac{1}{H} \sum_{t=1}^{H} \left[ d_t^{\pi^*}(I_{T_1}) + d_t^{\pi^*}(I_{T_2}) \right] = O(\varepsilon_q),$$

which is also the order of quantization error since the quantizer only induces constant error on $I_{T_1}$ and $I_{T_2}$, and the error on quantized expert is,

$$\frac{1}{H} \mathbb{E}_{q \# \pi^*} \left[ \sum_{h=1}^{H} \| \arctan(x_h) - q(u_h) \| \right] \geq \Theta(\frac{1}{|\log(1/\varepsilon_q)|}) - M \frac{1}{H(1-\rho)} = \Omega(\frac{1}{|\log \varepsilon_q|}).$$

□

### D.3. Proof of Theorem 7

*Proof.* We state again our notation here. The expert trajectory is upper indexed by $^*$. The auxiliary trajectory is upper indexed by $^a$. The rolled-out trajectory is upper indexed by $^L$.

We construct the coupling in this way. First let us denote $\bar{\mathbb{P}}^{\pi^*, T, q}$ as the joint distribution of the expert trajectory, explicitly including the noise $\{\omega_{0:H}^*, u_{1:H}^*, \tilde{u}_{1:H}^*, x_{1:H+1}^*\}$, where we remind the readers again that $x_{h+1}^* = f_h(x_h^*, u_h^*, \omega_h^*), \tilde{u}_h^* = q(u_h^*)$. Notice that the marginal of $\{x_{1:H+1}^*, \tilde{u}_{1:H}^*\} \sim \mathbb{P}^{q \# \pi^*, T \circ \rho}$. For a learned policy and model $\hat{\pi}, \widehat{(T \circ \rho)}$, denote the trajectory as $\{x_{1:H+1}^a, u_{1:H}^a\} \sim \mathbb{P}^{\hat{\pi}, \widehat{(T \circ \rho)}}$. Now we define $\mu$ as the maximal coupling between $\mathbb{P}^{\hat{\pi}, \widehat{(T \circ \rho)}}$ and $\mathbb{P}^{q \# \pi^*, T \circ \rho}$. Following the same spirit as in Lemma 21, we know that we can make sure under $\mu$, $x_1^* = x_1^a$ $\mu$ a.s. Then we use the glueing lemma to glue $\bar{\mathbb{P}}^{\pi^*, T, q}$ and $\mu$ to become $\mu'$. Finally, under $\mu'$, we define the rolled-out trajectory as $x_h^L = x_1^*, u_h^L = u_h^a, \omega_h^L = \omega_h^*, x_{h+1}^L = f_h(x_h^L, u_h^L, \omega_h^L)$. Now notice that the marginal of $\{x_{1:H+1}^L, u_{1:H}^L\}$, and $\{x_{1:H+1}^a, u_{1:H}^a\}$ is exactly the measure generated by the rolled-out algorithm. This means that,

$$J(\pi^*) - J(\text{alg}) = \mathbb{E}_{\mu'} \left[ \sum_{h=1}^{H} r_h(x_h^*, u_h^*) - r_h(x_h^L, u_h^L) \right].$$

We denote the exponential summation modulus as $\gamma$ first, now notice that

$$\mathbb{E}_{\mu'} \left[ \left( \sum_{h=1}^{H} r_h(x_h^*, u_h^*) - r_h(x_h^L, u_h^L) \right) \mathbb{I} \left( \bigcap_{h=1}^{H} \{\tilde{u}_h^* = u_h^L\} \right) \right]$$

$$\leq \mathbb{E}_{\mu'} \left[ \left( \sum_{h=1}^{H} L_r(\|x_h^* - x_h^L\| + \|u_h^* - u_h^a\|) \right) \mathbb{I} \left( \bigcap_{h=1}^{H} \{\tilde{u}_h^* = u_h^a\} \right) \right]$$

$$\leq \mathbb{E}_{\mu'} \left[ \sum_{h=1}^{H} L_r(\gamma(\|u_t^* - \tilde{u}_t^*\|)_{t=1}^{h-1} + \|u_h^* - \tilde{u}_h^*\|) \right],$$

where the second inequality is because by the construction of $\mu'$, the coupling between expert trajectory and rolled-out trajectory is a shared-noise coupling, and we assume global P-EIISS. Then, on the complement event, we have

$$\mathbb{E}_{\mu'} \left[ \left( \sum_{h=1}^{H} r_h(x_h^*, u_h^*) - r_h(x_h^L, u_h^L) \right) \mathbb{I} \left( \bigcup_{h=1}^{H} \{\tilde{u}_h^* \neq u_h^a\} \right) \right]$$

$$\leq H \cdot \mu'(\bigcup_{h=1}^{H} \{\tilde{u}_h^* \neq u_h^a\}) \leq H \cdot \mu'(\tau^* \neq \tau^a)$$

$$\leq H \cdot \text{TV}(\mathbb{P}^{q \# \pi^*, (T \circ \rho)}, \mathbb{P}^{\hat{\pi}, (\hat{T} \circ \rho)}).$$

Now by Proposition 11, for any $\delta > 0$, it holds with probability at least $1 - \delta$,

$$\text{TV}(\mathbb{P}^{q \# \pi^*, (T \circ \rho)}, \mathbb{P}^{\hat{\pi}, (\hat{T} \circ \rho)}) \leq \sqrt{D_H^2 \left( \mathbb{P}^{q \# \pi^*, (T \circ \rho)}, \mathbb{P}^{\hat{\pi}, \widehat{T \circ \rho}} \right)} \lesssim \sqrt{\frac{\log(|\Pi|\delta^{-1}) + \log(|\mathcal{T}|\delta^{-1})}{n}}.$$

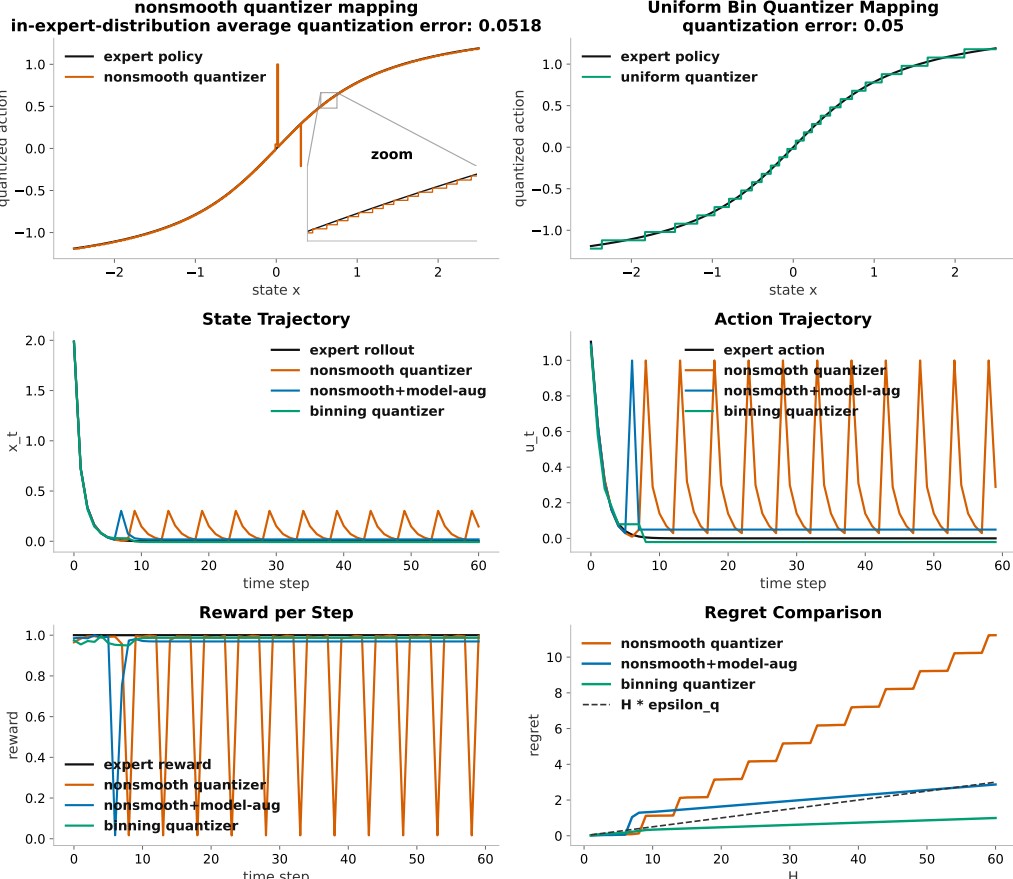

*Figure 1.* Simulation Results for binning, nonsmooth policy and model-based augmentation. The first row show the quantized policy $q \circ \pi^*(x)$ induced by a non-smooth quantizer and a binning-based quantizer, respectively. Specifically, the non-smooth quantizer is the learning-based quantizer described above, which achieves small quantization error on the expert distribution yet lacks smoothness. The second row and the first panel of the third row show the trajectories obtained by deploying the models imitating these quantized policies, as well as the trajectory obtained wia model-based augmentation. The last panel reports the regret, demonstrating that model-based augmentation significantly reduces it.

Then we plug in $\gamma$, we have,

$$\mathbb{E}_{\mu'} \left[ \sum_{h=1}^{H} L_r (\gamma(\|u_t^* - \tilde{u}_t^*\|)_{t=1}^{h-1} + \|u_h^* - \tilde{u}_h^*\| \right] \lesssim \mathbb{E}_{\mathbb{P}^{\pi^*}, \tau} \left[ \sum_{h=1}^{H} \|u_t^* - \tilde{u}_t^*\| \right] = H\varepsilon_q.$$

Combining the bounds on two events, we have proved the desired bound. $\qquad \square$

### D.4. Simulation Results

We conduct a simulation study to validate our main arguments in Section 4. The results are shown in Figure 1.

**Simulation Setting.** We use the 1-d system $x_{t+1} = 0.2x_t + 0.3u_t$ and the expert $\pi^*(x) = \arctan(x)$, following the construction in the proof of Theorem 6, deterministic part. The learning-based quantizer is also constructed following the same proof; specifically, it is designed so that deploying $q \circ \pi^*(x)$ traps the state into two sets, and is thus different from the expert trajectory. The quantizer's average quantization error on expert distribution is $\varepsilon_q \approx 0.5$ (estimated via Monte Carlo, following the definition in Eq. (4)). The binning-based quantizer uses uniform quantization on the action space with bin width $0.5$.

In the simulation, the learner's policy is parameterized by a transformer. Given a quantizer, we map the representative actions

$\tilde{\mathcal{U}}$ to base-6 token sequences and train the transformer on these tokens. The transition model is a conditional Gaussian model in which only the mean is parameterized by a neural network, while the variance is fixed. We train the transformer policy and transition model on 5000 trajectories until converge.

**Key takeaways.** We summarize the key takeaways from the results as follows.

1. With a learning-based quantizer with small quantization error but a non-smooth induced quantized policy, the log-loss BC with action quantization suffers from severe distribution shift. This issue is not resolved by increasing the sample size (we use 5000 trajectories with horizon $H = 60$) or scaling up the model (we use a 4-layer, 3-head transformer on scaler state and action space).

2. Under exactly the same setting, adding model-based augmentation significantly reduces the regret and resolves this issue.

3. Under the same dynamic, expert, and reward function, a binning-based quantizer (which is smooth) does not exhibit this issue.

# E. Proofs from Section 5

## E.1. Proof of Theorem 8

*Proof.* Consider $\mathcal{X} = \mathcal{U} = [0, 1]$. $T_0 = (1 - \Delta)\delta_0 + \Delta\delta_1$. Denote $\pi_1^a(\cdot|x) = \delta_0$, $\pi_1^b(\cdot|x) = \delta_x$. We consider set of rewards $r_1^a(x, u) = 1 - u$, $r_1^b(x, u) = xu + (1 - x)(1 - u)$, $r_h^a(x, u) = r_h^b(x, u) = x, h \geq 2$. Let the two dynamic be: $f^a = r^a$, $f^b = r^b$. One can easily verify that $\mathbb{P}^a$ and $\mathbb{P}^b$ is $(\delta, \eta)$-stable with any $\delta > 0$ and $\eta\left((r_k)_{k=1}^H\right) = r_1$.

Consider two instances $(T_0, f^a, \pi^a, r_a)$ and $(T_0, f^b, \pi^b, r_b)$. For any algorithm that seeing the quantized data, denote $\hat{\theta}(S_n^q) := \int u d\hat{\pi}(S_n^q)(du|1)$

$$J_a(\pi^a) - J_a(\hat{\pi}) = H\Delta[1 - (1 - \hat{\theta}(S_n^q))] + H(1 - \Delta)\int_0^1 1 - (1 - u)d\hat{\pi}_1(u|0)$$

$$\geq H\Delta|\hat{\theta}(S_n^q) - \theta^*(\pi^a)|, \theta^*(\pi^a) := \mathbb{E}_{u_1 \sim \pi_1^a(\cdot|1)}[u_1],$$

$$J_b(\pi^b) - J_b(\hat{\pi}) = H\Delta[1 - \hat{\theta}(S_n^q))] + H(1 - \Delta)\int_0^1 u d\hat{\pi}_1(u|0)$$

$$\geq H\Delta|\theta^*(\pi^b) - \hat{\theta}(S_n^q)|, \theta^*(\pi^b) := \mathbb{E}_{u_1 \sim \pi_1^b(\cdot|1)}[u_1].$$

Therefore, we can define the metric as $\rho(\pi, \pi') := |\mathbb{E}_{u_1 \sim \pi_1(\cdot|1)}[u_1] - \mathbb{E}_{a_1' \sim \pi_1(\cdot|1)}[u_1']|$. Using the standard Le Cam two-point argument, the algorithm must have,

$$\max\{\mathbb{E}[J_{r_a}(\pi_a) - J_{r_a}(\hat{\pi})], \mathbb{E}[J_{r_b}(\pi_b) - J_{r_b}(\hat{\pi})]\} \geq \frac{\Delta H}{4}(1 - D_{\text{TV}}(\mathbb{P}^{a \otimes n}, \mathbb{P}^{b \otimes n})).$$

Notice that it holds that,

$$\text{TV}(\mathbb{P}^{a \otimes n}, \mathbb{P}^{b \otimes n}) \leq \sqrt{D_H^2(\mathbb{P}^{a \otimes n}, \mathbb{P}^{b \otimes n})} = \sqrt{2}\sqrt{1 - (1 - \frac{1}{2}D_H^2(\mathbb{P}^a, \mathbb{P}^b))^n}.$$

Through computation, it holds that $1 - \frac{1}{2}D_H^2(\mathbb{P}^a, \mathbb{P}^b)) = 1 - \Delta$. Set $\Delta = \frac{1}{3n}$. By the limiting behavior of $(1 - \frac{1}{3n})^n$, we know for large enough $n$, it holds that $\text{TV}(\mathbb{P}^{a \otimes n}, \mathbb{P}^{b \otimes n}) \leq 0.8$ (via numerical computation), then we have $\frac{H}{n}$ lower bound.

Now we show how to achieve the additional $H\varepsilon_q$ for the lower bound. This is trivial. First we specify the quantization scheme. Consider a uniform binning quantizer $q : [0, 1] \to \mathbb{R}$. Partition $[0, 1]$ into $K$ consecutive subintervals of equal length $1/K$, denoted $\{I_k\}_{k=1}^K$, with $I_1 = [0, \frac{1}{K}]$ and $I_K = [\frac{K-1}{K}, 1]$. For $k = 2, \ldots, K - 1$, the endpoints may be chosen arbitrarily as open or closed, provided that the $I_k$ are pairwise disjoint and $\bigcup_{k=1}^K I_k = [0, 1]$. Let $m_k = \frac{2k-1}{2K}$ be the midpoint of $I_k$. Then

$$q(x) = m_k \quad \text{for } x \in I_k, k = 1, \ldots, K.$$

Notice that under such a quantizer,

$$\max_{i \in [K]} \text{diam}(q^{-1}(\bar{u}_i)) = \varepsilon_q, \frac{1}{H} \mathbb{E}^{\pi^*}[\sum_{h=1}^{H} \|u_h^* - q(u_h^*)\|] \leq \varepsilon_q.$$

If we change $\pi^a$ into $\pi^{\tilde{a}}$ and $\pi^b$ into $\pi^{\tilde{b}}$ as

$$\pi_1^{\tilde{a}}(u|x) = \delta_{\varepsilon_q}, \pi_1^{\tilde{b}}(u|x) = x\delta_{1-\varepsilon_q} + (1-x)\delta_{\varepsilon_q},$$

and let,

$$r_u^0(u) = u\mathbb{I}(u \leq 1 - \varepsilon_q) + (2 - 2\varepsilon_q - u)\mathbb{I}(u \geq 1 - \varepsilon_q)$$
$$r_u^1(u) = (1 - 2\varepsilon_q + u)\mathbb{I}(u \leq \varepsilon_q) + (1 - u)\mathbb{I}(u > \varepsilon_q)$$
$$r_1^{\tilde{a}}(x, u) = r_u^1(u), r_1^{\tilde{b}}(x, u) = xr_u^0(u) + (1-x)r_u^1(u),$$
$$f_1^{\tilde{a}} = r_1^{\tilde{a}}, f^{\tilde{b}} = r_1^{\tilde{b}}, f_h^{\tilde{a}}(x, u) = f_b^{\tilde{b}}(x, u) = x.$$

Notice that we have,

$$J_a(\pi^a) - J_a(\hat{\pi}) \geq H\Delta \left[ \int_0^{\varepsilon_q} u d\hat{\pi}_1(u|1) + \int_{\varepsilon_q}^1 \varepsilon_q d\hat{\pi}(u|1) \right] + H(1-\Delta) \left[ \int_0^{\varepsilon_q} u d\hat{\pi}_1(u|0) + \int_{\varepsilon_q}^1 \varepsilon_q d\hat{\pi}(u|0) \right]$$

$$J_{\tilde{a}}(\pi^{\tilde{a}}) - J_{\tilde{a}}(\hat{\pi}) \geq H\Delta \int_0^{\varepsilon_q} (\varepsilon_q - u) d\hat{\pi}_1(u|1) + H(1-\Delta) \int_0^{\varepsilon_q} (\varepsilon_q - u) d\hat{\pi}_1(u|0)$$

$$\Rightarrow J_a(\pi^a) - J_{r^a}(\hat{\pi}) + J_{r^{\tilde{a}}}(\pi^{\tilde{a}}) - J_{r^{\tilde{a}}}(\hat{\pi}) \geq H\Delta\varepsilon_q + H(1-\Delta)\varepsilon_q = H\varepsilon_q.$$

This is because when $\pi^a$ changes to $\pi^{\tilde{a}}$, $\hat{\pi}$ will remain unchanged since it can only access the quantized data, and the distribution of the quantized data under $\pi^a$ and $\pi^{\tilde{a}}$ are the same for the given quantizer. One can also verify that it holds,

$$J_b(\pi^b) - J_b(\hat{\pi}) \geq H\Delta \left[ \int_0^{1-\varepsilon_q} \varepsilon_q d\hat{\pi}_1(u|1) + \int_{1-\varepsilon_q}^1 (1-u) d\hat{\pi}(u|1) \right] + H(1-\Delta) \left[ \int_0^{\varepsilon_q} u d\hat{\pi}_1(u|0) + \int_{\varepsilon_q}^1 \varepsilon_q d\hat{\pi}(u|0) \right]$$

$$J_{\tilde{b}}(\pi^{\tilde{b}}) - J_{\tilde{b}}(\hat{\pi}) \geq H\Delta \int_{1-\varepsilon_q}^1 [(1-\varepsilon_q) - (2 - 2\varepsilon_q - u)] d\hat{\pi}_1(u|1) + H(1-\Delta) \int_0^{\varepsilon_q} (\varepsilon_q - u) d\hat{\pi}_1(u|0)$$

$$\Rightarrow J_b(\pi^b) - J_b(\hat{\pi}) + J_{\tilde{b}}(\pi^{\tilde{b}}) - J_{\tilde{b}}(\hat{\pi}) \geq H\varepsilon_q.$$

Then we have,

$$J_a(\pi^a) - J_a(\hat{\pi}) + J_{\tilde{a}}(\pi^{\tilde{a}}) - J_{\tilde{a}}(\hat{\pi}) \geq \frac{1}{2} J_a(\pi^a) - J_a(\hat{\pi}) + \frac{1}{2} H\varepsilon_q,$$

$$\max\{J_a(\pi^a) - J_a(\hat{\pi}), J_{\tilde{a}}(\pi^{\tilde{a}}) - J_{\tilde{a}}(\hat{\pi})\} \geq \frac{1}{4} J_a(\pi^a) - J_a(\hat{\pi}) + \frac{1}{4} H\varepsilon_q,$$

$$\max\{\mathbb{E}[J_a(\pi_a) - J_a(\hat{\pi})], \mathbb{E}[J_{\tilde{a}}(\pi^{\tilde{a}}) - J_{\tilde{a}}(\hat{\pi})], \mathbb{E}[J_b(\pi_b) - J_b(\hat{\pi})], \mathbb{E}[J_{\tilde{b}}(\pi^{\tilde{b}}) - J_{\tilde{b}}(\hat{\pi})]\}$$

$$\geq \max\{\frac{1}{4}\mathbb{E}[J_a(\pi_a) - J_a(\hat{\pi})] + \frac{1}{4} H\varepsilon_q, \frac{1}{4}\mathbb{E}[J_b(\pi_b) - J_b(\hat{\pi})] + \frac{1}{4} H\varepsilon_q\}$$

$$\geq c(\frac{H}{n} + H\varepsilon_q). \qquad \square$$

### E.2. Proof of Theorem 9

*Proof.* We will show that it suffices to prove the lower bound for the simple distribution estimation problem for the initial action distribution. Consider $\mathcal{X} = \mathcal{A} = [-1, 2]$. We construct the dynamic as $f_1(x, u) = u, f_h(x, u) = x, h \geq 2$. And a policy class $\Pi = \{\pi^a, \pi^b\}$ with

$$\pi_1^a(u|x) = (\frac{1}{2} + \Delta)\delta_1(u) + (\frac{1}{2} - \Delta)\delta_0(u), \pi_1^b(u|x) = (\frac{1}{2} - \Delta)\delta_1(u) + (\frac{1}{2} + \Delta)\delta_0(u),$$

and $\pi_h^a = \pi_h^b = \delta_1, h \geq 2$. The policy after $h \geq 2$ does not matter. Let $\mathbb{P}^a$ and $\mathbb{P}^b$ denote the distribution of a single trajectory $x_1, a_1, \ldots, x_H$ under $(f, \pi^a)$ and $(f, \pi^b)$. Consider reward as

$$r_1^a(x_1, u_1) = u_1, r_h^a(x_h, u_h) = x_h, h \geq 2$$
$$r_1^b(x_1, u_1) = 1 - u_1, r_h^a(x_h, u_h) = 1 - x_h, h \geq 2.$$

Notice that under different combinations of reward and policy, denote $\hat{\theta}(S_n^q) = \int_{-1}^2 u d\hat{\pi}_1(u)(S_n^q)$, it has

$$J_{r^a}(\pi^a) - J_{r^a}(\hat{\pi}) = (\frac{1}{2} + \Delta)H - H\hat{\theta}(S_n^q),$$

$$J_{r^b}(\pi^a) - J_{r^b}(\hat{\pi}) = (\frac{1}{2} - \Delta)H - H[1 - \hat{\theta}(S_n^q)],$$

$$J_{r^a}(\pi^b) - J_{r^a}(\hat{\pi}) = (\frac{1}{2} - \Delta)H - H\hat{\theta}(S_n^q),$$

$$J_{r^b}(\pi^b) - J_{r^b}(\hat{\pi}) = (\frac{1}{2} + \Delta)H - H[1 - \hat{\theta}(S_n^q)].$$

Notice that we have,

$$\max\left\{\mathbb{P}^a\big[J_{r^a}(\pi^a) - J_{r^a}(\hat{\pi}) \geq \Delta H\big], \mathbb{P}^a\big[J_{r^b}(\pi^a) - J_{r^b}(\hat{\pi}) \geq \Delta H\big],\right.$$
$$\left.\mathbb{P}^b\big[J_{r^b}(\pi^b) - J_{r^b}(\hat{\pi}) \geq \Delta H\big], \mathbb{P}^b\big[J_{r^a}(\pi^b) - J_{r^a}(\hat{\pi}) \geq \Delta H\big]\right\}$$

$$\geq \max\left\{\mathbb{P}^a\big[(\frac{1}{2} + \Delta)H - \hat{\theta}(S_n^q)H \geq \Delta H\big], \mathbb{P}^a\big[\hat{\theta}(S_n^q)H - (\frac{1}{2} + \Delta)H \geq \Delta H\big],\right.$$
$$\left.\mathbb{P}^b\big[\hat{\theta}(S_n^q)H - (\frac{1}{2} - \Delta)H \geq \Delta H\big], \mathbb{P}^b\big[(\frac{1}{2} - \Delta)H - \hat{\theta}(S_n^q)H \geq \Delta H\big]\right\}$$

$$\geq \frac{1}{2}\max\left\{\mathbb{P}^a\big[|(\frac{1}{2} + \Delta) - \hat{\theta}(S_n^q)|H \geq \Delta H\big], \mathbb{P}^b\big[|\hat{\theta}(S_n^q) - (\frac{1}{2} - \Delta)|H \geq \Delta H\big]\right\}$$

$$= \frac{1}{2}\max\left\{\mathbb{P}^a\big[|(\frac{1}{2} + \Delta) - \hat{\theta}(S_n^q)| \geq \Delta\big], \mathbb{P}^b\big[|\hat{\theta}(S_n^q) - (\frac{1}{2} - \Delta)| \geq \Delta\big]\right\}$$

$$\geq \frac{1}{4}\left(\mathbb{P}^a\big[|(\frac{1}{2} + \Delta) - \hat{\theta}(S_n^q)| \geq \Delta\big] + \mathbb{P}^b\big[|\hat{\theta}(S_n^q) - (\frac{1}{2} - \Delta)| \geq \Delta\big]\right).$$

Now consider the hypothesis test task that distinguishing between $\pi^a$ and $\pi^b$. The decision rule is accepting $\pi^a$ if $|\hat{\theta} - (\frac{1}{2} + \Delta)| \leq \Delta$ and accepting $\pi^b$ otherwise, then by Bretagnolle-Huber inequality, we have,

$$\mathbb{P}^a\big[|(\frac{1}{2}+\Delta)-\hat{\theta}(S_n^q)| \geq \Delta\big] + \mathbb{P}^b\big[|\hat{\theta}(S_n^q)-(\frac{1}{2}-\Delta)| \geq \Delta\big] \geq \mathbb{P}^a[\text{rejecting } \pi^a] + \mathbb{P}^b[\text{rejecting } \pi^b] \geq 1 - D_{\text{TV}}(\mathbb{P}^{a \otimes n}, \mathbb{P}^{b \otimes n}).$$

And it holds that,

$$D_{\text{TV}}(\mathbb{P}^{a \otimes n}, \mathbb{P}^{b \otimes n}) \leq \sqrt{1 - (1 - 4\Delta^2)^n}.$$

Take $\Delta = \frac{3}{5}\frac{1}{\sqrt{n}}$, then for large $n$, it holds that $D_{\text{TV}}(\mathbb{P}^{a \otimes n}, \mathbb{P}^{b \otimes n}) \leq \frac{7}{8}$. Then it holds that,

$$\max\left\{\mathbb{P}^a\big[J_{r^a}(\pi^a) - J_{r^a}(\hat{\pi}) \geq \frac{3H}{5\sqrt{n}}\big], \mathbb{P}^a\big[J_{r^b}(\pi^a) - J_{r^b}(\hat{\pi}) \geq \frac{3H}{5\sqrt{n}}\big],\right.$$
$$\left.\mathbb{P}^b\big[J_{r^b}(\pi^b) - J_{r^b}(\hat{\pi}) \geq \frac{3H}{5\sqrt{n}}\big], \mathbb{P}^b\big[J_{r^a}(\pi^b) - J_{r^a}(\hat{\pi}) \geq \frac{3H}{5\sqrt{n}}\big]\right\} \geq \frac{1}{8}.$$

Finally, we introduce the extra $H\varepsilon_q$ term. Let $q$ be defined the same way as in the proof of Theorem 8. Let,

$$\pi_1^{\tilde{a}}(u|x) = (\frac{1}{2} + \Delta)\delta_{1+\varepsilon_q}(u) + (\frac{1}{2} - \Delta)\delta_{\varepsilon_q}(u), \pi_1^{\hat{a}}(u|x) = (\frac{1}{2} + \Delta)\delta_{1-\varepsilon_q}(u) + (\frac{1}{2} - \Delta)\delta_{-\varepsilon_q}(u)$$

$$\pi_1^{\tilde{b}}(u|x) = (\frac{1}{2} - \Delta)\delta_{(1-\varepsilon_q)}(u) + (\frac{1}{2} + \Delta)\delta_{-\varepsilon_q}(u), \pi_1^{\hat{b}}(u|x) = (\frac{1}{2} - \Delta)\delta_{1+\varepsilon_q}(u) + (\frac{1}{2} + \Delta)\delta_{\varepsilon_q}(u).$$

Now notice that, we should have,

$$\mathbb{P}^{\tilde{a}}[J_{r^a}(\pi^{\tilde{a}}) - J_{r^a}(\hat{\pi}) \geq \frac{3H}{5\sqrt{n}} + H\varepsilon_q] = \mathbb{P}^a[J_{r^a}(\pi^a) - J_{r^a}(\hat{\pi}) \geq \frac{3H}{5\sqrt{n}}]$$

$$\mathbb{P}^{\hat{a}}[J_{r^b}(\pi^{\hat{a}}) - J_{r^b}(\hat{\pi}) \geq \frac{3H}{5\sqrt{n}} + H\varepsilon_q] = \mathbb{P}^a[J_{r^b}(\pi^a) - J_{r^b}(\hat{\pi}) \geq \frac{3H}{5\sqrt{n}}]$$

$$\mathbb{P}^{\tilde{b}}[J_{r^a}(\pi^{\tilde{b}}) - J_{r^a}(\hat{\pi}) \geq \frac{3H}{5\sqrt{n}} + H\varepsilon_q] = \mathbb{P}^b[J_{r^a}(\pi^a) - J_{r^a}(\hat{\pi}) \geq \frac{3H}{5\sqrt{n}}]$$

$$\mathbb{P}^{\hat{b}}[J_{r^b}(\pi^{\hat{b}}) - J_{r^b}(\hat{\pi}) \geq \frac{3H}{5\sqrt{n}} + H\varepsilon_q] = \mathbb{P}^b[J_{r^b}(\pi^b) - J_{r^b}(\hat{\pi}) \geq \frac{3H}{5\sqrt{n}}].$$

This is because $\hat{\pi}$ should be the same under $\mathbb{P}^{\tilde{a}}$ and $\mathbb{P}^a$. Therefore we have,

$$\max\left\{\mathbb{P}^a[J_{r^a}(\pi^{\tilde{a}}) - J_{r^a}(\hat{\pi}) \geq \frac{3H}{5\sqrt{n}} + \frac{1}{2}H\varepsilon_q], \mathbb{P}^a[J_{r^b}(\pi^{\hat{a}}) - J_{r^b}(\hat{\pi}) \geq \frac{3H}{5\sqrt{n}} + \frac{1}{2}H\varepsilon_q],\right.$$

$$\left.\mathbb{P}^b[J_{r^b}(\pi^{\tilde{b}}) - J_{r^b}(\hat{\pi}) \geq \frac{3H}{5\sqrt{n}} + \frac{1}{2}H\varepsilon_q], \mathbb{P}^b[J_{r^a}(\pi^{\hat{b}}) - J_{r^a}(\hat{\pi}) \geq \frac{3H}{5\sqrt{n}} + \frac{1}{2}H\varepsilon_q]\right\} \geq \frac{1}{8}. \qquad \square$$

