# OpenReview forum: "Understanding Behavior Cloning with Action Quantization"
_ICML.cc/2026/Conference — ICML 2026 regular_

### Official Review · Reviewer_d7pw · 2026-02-18

**Soundness:** 3
**Presentation:** 3
**Significance:** 4
**Originality:** 2
**Overall Recommendation:** 5
**Confidence:** 4

**Summary:**

This paper studies the important question of imitation learning in continuous action spaces. The approach studied here is natural and often used in practice: action quantization. The authors proposed two notions of quantization error ( learning based and binning-like) and prove benign (Theorem 2 and 3) quantization error propagations under the assumption that the expert trajectory is globally P-EIISS, that the quantized expert is realizable and that the learner hypothesis class is smooth (RTVC or TVC).

The authors then argue that these assumptions ( which imply that the expert needs to be smooth) are very limiting for the deterministic expert case, but they show that for log-loss BC having a smooth class is absolutely necessary to prove benign quantization error propagation results(Theorem 6).

This motivates new algorithmic choices. In particular, the authors suggest to learn a model of the transitions via log-loss minimization and use it to simulate a trajectory according to the learner policy. The actions are then played in the environment in an open loop action chunking style. The paper is completed by information-theoretic lower bounds (Theorems 8 and 9) which match the dependence in H and number of expert samples for the last presented algorithm.

**Compliance With Llm Reviewing Policy:**

Affirmed.

**Final Justification:**

After the rebuttal, I am happy to keep my positive evaluation.

**Key Questions For Authors:**

1) In Proposition 1, why do you need to write $\pi_2$ instead of simply $\tilde{\pi}_1$ ? Moreover, I would suggest to the authors to highlight which terms in this proposition can be made smaller via more expert data and which one can be made small by a more-fine grained quantizer. Also, in the second line of the equation in Proposition 1, I think that $\tilde{u}$ should be $\tilde{u}^0$.

2) In Theorems 2 and 3 the assumptions are slightly different binning based vs learning based quantizer, RTVC vs TVC classes but both results are proven via Proposition 1. Are the different assumptions only for the sake of expositions ? Or is it that for deterministic experts one could not prove a result for learning-based quantizers ? Moreover, why proposition1 requires only local P-IISS while in Theorem 2 and 3 you require the stronger assumption of global P-EIISS for the stochastic case and P-IISS for a certain gamma  for the deterministic case.  Is this again only for clarity ? Is it important that for deterministic experts you choose P-IISS while for stochastic you require P-EIISS ? Please clarifies this points also in your next revision.

3) I liked the comparison with Simchovitz et al. 2025 work. In their setting any source of error in the regression target (no matter if due to learning or quantization would be exponentially increased). This is avoided in your analysis thanks to the TVC assumption while the learner class in Simchovitz et al. 2025 is not TVC ?

4) The model augmented algorithm is very interesting and cleverly designed but it introduces a dependence on $log T$. Do the authors think this dependence is necessary for the improved horizon dependence ?

5) In the conclusive lower bounds, only the dependence on the horizon and number of expert data is captured. Can you also capture the dependence on log policy class factor ?

**Limitations:**

Computational complexity is not discussed throughout the paper but I think that it is an important point and potential drawback of quantization.

In particular, the authors say that a bin-based quantizer should be preferred but computation-wise this requires $e^d$ points  e.g. for a $d$  dimensional $\ell_\infty$ norm action space. On the other hand, my intuition says that learning based quantizer should be way more attractive computationally because they require a fine discretization only around the actions that can be taken.

I think the authors should discuss these points. In addition, would you know any "trick" that one could use to implement BC with bin-quantized actions in a computationally efficient manner in the same particular cases ?

**Strengths And Weaknesses:**

The paper is well written and the results are sounds.

The problem studied here is of practical impact.

---

> ### Author Rebuttal · Authors · 2026-03-31
>
> We sincerely thank the reviewer for the very positive evaluation and the many thoughtful, constructive questions, which are very helpful for improving the paper. Below we respond to each point as clearly and concisely as possible.
>
> > Why do you write $\pi^2$ instead of $\tilde \pi^1$?
>
> We write $\pi^2$ mainly because, in the proof, we construct and couple three trajectory measures, namely $\tau^0$, $\tau^1$, and $\tau^2$.
>
> > I would suggest ... highlight which term...
>
> We will add this in the revision.
>
> > I think that $\tilde u$ should be $\tilde u^0$.
>
> We will correct it in the revision.
>
> > Key question 2
>
> We agree that the presentation can be improved. The overall idea is Proposition 1 gives the most general upper bound, while Theorems 2 and 3 impose additional structural assumptions on $\gamma$ and $\kappa$ in order to obtain cleaner bounds.
>
> > Proposition 1 requires only local P-IISS, while Theorems 2 and 3 require the stronger assumption of global P-EIISS.
>
> Yes. This stronger assumption is mainly adopted for simplicity, so that we can remove the first and third terms on the right-hand side of the upper bound in Proposition 1.
>
> > In Theorems 2 and 3 the assumptions are slightly different ... Are the different assumptions only for the sake of exposition? Or ... for deterministic experts one could not prove a result for learning-based quantizers?
>
> These are not only for the sake of exposition.
>
> For Theorem 2, we consider stochastic experts. As discussed in Section 4, both learning-based quantizers and binning quantizers can satisfy TVC in this setting, so we adopt the stronger smoothness notion there.
>
> For Theorem 3, we consider deterministic experts. As discussed in Section 4, learning-based quantizers essentially cannot make the quantized expert satisfy RTVC, let alone TVC, so we restrict attention to binning quantizers.
>
> > Is it important that for deterministic experts you choose P-IISS while for stochastic you require P-EIISS?
>
> This is mainly for technical convenience rather than a fundamental distinction. For stochastic experts, we also allow learning-based quantizers and represent the quantization error as in Equation (4). Under EIISS and a linear modulus $\kappa$, this yields
> $$
> \mathbb E_{\bar{\mathbb P}^{\pi^0,T,\tilde \pi^0}}\left[\sum_{h=1}^H \kappa\left(\gamma((\|u_k^0-\tilde u_k^0\|)_{k=1}^{h-1})\right)\right]\lesssim \varepsilon_q.
> $$
> For deterministic experts, we use binning quantizers and directly plug in $\|u_h^0-\tilde u_h^0\|\le \varepsilon_q$, which gives the simpler bound in Theorem 3.
>
> > ... in Simchovitz et al. 2025 ... any source of error ... would be exponentially increased. This is avoided thanks to the TVC assumption ... ?
>
> Yes, this is our understanding. Our upper bound relies on TVC, whereas [1] does not assume TVC. A subtlety, however, is that [1] studies deterministic experts, which naturally violate TVC as discussed in Section 4. Therefore, although our Proposition 4 remains valid, its right-hand side may not be small when a TVC learner is used to imitate a non-TVC expert. This issue remains open from our perspective. For stochastic experts, by contrast, TVC can hold, so Proposition 4 is still meaningful.
>
> > The model augmented algorithm is very interesting ... Do the authors think this dependence is necessary for the improved horizon dependence?
>
> Yes, we believe this dependence is necessary, as it reflects the complexity of the transition model class.
>
> > In the conclusive lower bounds ... can you also capture the dependence on log policy class factor?
>
> Since the lower bound is information-theoretic, we mainly focus on the informational cost required for learning. For this reason, the policy class term—which is more algorithmic in nature—is not emphasized there.
>
> > ... learning based quantizer should be way more attractive computationally because they require a fine discretization only around the actions that can be taken.
>
> We agree. A learning-based quantizer may compress the action space much more efficiently, reducing the effective statistical learning complexity. The downside is that its behavior may be poorly controlled off the expert distribution, especially without the smoothness needed to correct OOD states. So there is indeed a tradeoff. An interesting direction is whether one can apply a post-hoc smoothing procedure to a learning-based quantizer and obtain the advantages of both approaches.
>
> > Computational complexity is not discussed throughout the paper ... In addition, would you know any "trick" ... to implement BC with bin-quantized actions in a computationally efficient manner ... ?
>
>  We agree that computational complexity is important and should be discussed more explicitly. As far as we know, one can discretize each action dimension independently and train BC as token prediction with cross-entropy. This is adopted in RT-1, RT-2 and Open-VLA.
>
> [1] Simchowitz et al. (2025), *The pitfalls of imitation learning when actions are continuous*.

---

> > ### Author Rebuttal · Reviewer_d7pw · 2026-04-01
> >
> > Thank you for your rebuttal. All my points have been clarified, and I am happy to keep my very positive evaluation.
> >
> > I have a last question out of curiosity. Do you think that learning the model could be beneficial even in the real continuous action setting? i.e., without using a quantizer and learning the transition model by regressing onto the expert next states?
> >
> > I am wondering if this algorithmic modification could break the algorithm-dependent lower bound in [1] while not introducing dependence on the quantizer resolution $\varepsilon_q$.
> >
> > Thank you in advance, and congratulations on this nice work.

---

> > > ### Author Response · Authors · 2026-04-05
> > >
> > > Thank you for your thoughtful response — it is truly pleasant to discuss these questions with you. For your question, our answer is that it depends. Let us first discuss the claim in [1]. In the truly continuous-action setting, the main issue is not deviation in the metric space (which would require TVC or RTVC to correct), but rather that log-loss BC itself fails for a deterministic expert in continuous space. This issue is discussed in Appendix B.2 (and Section 2.4) of [1]. The reason is that, in the deterministic continuous-action case, the log-loss can become infinite. Notably, this pathology arises in the truly continuous setting; once action quantization is introduced, log-loss can still be well defined and effective.
> > >
> > > Then we understand your question as follows: if, instead of imitating the continuous expert with log-loss, we use a metric-based loss such as the $L_1/L_2$ loss in [1], can model-based augmentation correct the deviation in the metric space without requiring TVC/RTVC? Our current belief is yes. In particular, see [2] , where Section 4 discusses action chunking under open-loop incremental stability. At a high level, their formulation of action chunking can be viewed as implicitly fitting a transition model (see Definition 3.2, where their chunking policy is induced by a Markovian policy and a dynamic model), which breaks the algorithmic lower bound in [1].
> > >
> > > That said, we would also like to point out an additional subtlety. If one first trains the policy as in [1], then fits a transition model with log-loss, and finally applies the sampling procedure in our paper, this may not work directly. The reason is that our proof relies on the standard trajectory-level concentration argument for MLE, whereas training the policy with a metric-based loss means the policy learner is no longer exactly performing MLE. One possible remedy would be to fit the transition model using a corresponding metric-based loss as well, and do not relies on MLE guarantee. At present, we are not fully sure about this point, but we would be very happy to discuss it further.
> > >
> > > [1] Simchowitz, Max, Daniel Pfrommer, and Ali Jadbabaie. "The pitfalls of imitation learning when actions are continuous." *arXiv preprint arXiv:2503.09722* (2025).
> > >
> > > [2] Zhang, Thomas T., et al. "Action Chunking and Exploratory Data Collection Yield Exponential Improvements in Behavior Cloning for Continuous Control." *arXiv preprint arXiv:2507.09061* (2025).

---

### Official Review · Reviewer_f82u · 2026-03-04

**Soundness:** 2
**Presentation:** 2
**Significance:** 2
**Originality:** 3
**Overall Recommendation:** 5
**Confidence:** 3

**Summary:**

This paper studies the effect of action quantization in behavior cloning for continuous control, motivated by autoregressive policies that discretize actions and optimize log loss. The paper shows that quantization introduces an additional regret term beyond standard estimation error, and develops a theoretical analysis that separates these two effects. It provides upper bounds under stability and smoothness assumptions, shows that binning-based quantizers are better behaved than general learned quantizers, proposes a model-based augmentation with improved horizon dependence, and gives lower bounds establishing that the quantization penalty is unavoidable in general. Overall, the paper provides a useful theoretical perspective on when action tokenization is safe and when it becomes a bottleneck.

**Compliance With Llm Reviewing Policy:**

Affirmed.

**Final Justification:**

The authors’ rebuttal and follow-up substantially improved my assessment of the paper. The clarifications on the assumptions and the correction to Algorithm 1 addressed important presentation and interpretation issues. More importantly, the added simulation provides the minimum practical validation that I felt was missing: it directly demonstrates the failure mode predicted by the theory, and shows that model-based augmentation meaningfully alleviates it under the same setting. This considerably strengthens the connection between the theoretical results and the paper’s motivating applications. Although some limitations remain and should be stated clearly in the final version, I now view the contribution as technically solid and practically better supported. Overall, I assess the paper more positively after the rebuttal.

**Key Questions For Authors:**

(1) The main guarantees rely on global P-IISS / P-EIISS and TVC / RTVC assumptions. How realistic are these assumptions in the robotic regimes motivating the paper?

(2) Theorem 7 improves the horizon dependence, but at the price of learning an additional induced transition model and paying a complexity term; the paper also notes that learning an H-step model may itself be challenging. When should this augmentation actually be preferred over plain quantized BC, once model-learning cost is taken into account?

(3) In Algorithm 1, the transition MLE objective appears to use
$
\log (T\circ \rho)h(x_{h+1}^{(i)} \mid q(u_h^{(i)})),
$
but shouldn’t this also be conditioned on the current state $(x_h^{(i)})$? Please clarify.

**Limitations:**

A stronger limitations section should briefly discuss as mentioned in weaknesses and questions.

**Strengths And Weaknesses:**

Strengths.
The paper addresses a timely and important problem with a clean theoretical framing. It studies the effect of action quantization in behavior cloning, a design choice that is increasingly common in autoregressive and VLAs. And the comparison between quantizer classes leads to a genuinely useful insight.


Weaknesses.
(a) Theorem 2 and Theorem 3 require global P-IISS / P-EIISS stability and TVC / RTVC-type smoothness assumptions (Definition 3–4, Theorems 2–3), but the paper does not sufficiently discuss when these conditions are expected to hold in the robotic settings motivating the work.
(b) The model-based augmentation improves the theory, but introduce additional burden to learning a transition model. Theorem 7 is technically appealing, but the paper itself notes that learning an H step transition model may be challenging. A discussion about the induced cost is suggested to be added.
(c) In Algorithm 1, the transition MLE objective appears to model without conditioning on the current state (x_h). If taken literally, this is not a Markov transition kernel (it learns a marginal next-state distribution given the token), which would materially change the algorithm and weaken the interpretation/guarantees that rely on learning a proper state-conditioned dynamics model.

Overall, while the paper offers a strong theorem-driven perspective, it leaves a noticeable gap between the theory and the robotics/VLA motivation. The main guarantees rely on global stability and smoothness assumptions whose applicability to realistic robotic systems is not discussed, and the proposed model-based augmentation improves the bounds but adds a potentially significant burden by requiring learning an (H)-step transition model. Moreover, Algorithm 1’s transition objective appears to omit conditioning on the current state, which (if not a notation slip) would break the Markov modeling interpretation that the guarantees depend on. Given these gaps, the paper would benefit from (i) a clearer discussion of when the assumptions and augmentation costs are expected to be reasonable in practice, and (ii) at least minimal empirical validation (even toy control experiments) to support the key qualitative claims and make the practical implications more credible.

---

> ### Author Rebuttal · Authors · 2026-03-31
>
> We thank the reviewer for the constructive feedback.
> > Weaknesses (a) and Key questions (1)
>
> **System stability (P-IISS/P-EIISS).**
>
> Firstly, our P-IISS is strictly weaker than the standard $\delta$-ISS notion (Definition 4.1 of [1]), which requires global stability and is the basic assumption in recent line of imitation learning works for continuous control. We only require stability on a local subset $S$ where the expert trajectory remains with high probability; see Remark 3 in Appendix B. The $\delta$-ISS framework is one of the most widely used stability paradigm in nonlinear control, see [6] for a tutorial. Hence, P-IISS applies more broadly.
>
> Secondly, Proposition 16 gives an explicit sufficient condition: if the dynamics satisfy a local contraction $\|f(x,u)-f(x',u')\| \le \eta\|x-x'\| + C\|u-u'\|$ with $\eta<1$ on a subset $S$, then P-EIISS holds. Although strong, this condition is only needed locally along the expert trajectory. More broadly, one can also stabilize the system through a controller and then work with the resulting closed-loop dynamics. In that case, the learner imitates deviations in the controller input, while stability is inherited from the stabilized dynamics; [2] gives an example.
>
> Third, our assumption is especially natural in systems with low-level controllers, where the learner should imitate the controller rather than the raw action. The resulting controller-induced closed-loop dynamics are already stabilized. Recent systems such as [3] and [4] follow this architecture. Moreover, following [5], when the low-level controller is a stabilizing affine mapping, their Definition 5.2 is provably satisfied (Lemma H.3 of [5]), and Definition 5.2 directly implies our P-IISS. Hence, systems under [5]'s framework satisfy our assumption with a quantitative guarantee.
>
> **Smoothness of policies (TVC/RTVC).**
>
> We note that some smoothness condition is **necessary**: [7] proves that for smooth deterministic policies, imitation error grows exponentially with horizon even under stable dynamics. TVC/RTVC is among the weakest conditions that provably avoids this exponential blowup.
>
> Our theory requires: (1) the policy class is TVC/RTVC, so the learned $\hat\pi$ satisfies TVC/RTVC; (2) the expert is realizable by this class, so it inherits the same property. For an arbitrary learner, Proposition 18 shows that after Gaussian smoothing, the new learner can always satisfy TVC. For the expert, Appendix B.2 shows Gaussian policies are already TVC, and by Lemma 15 the quantized policy remains TVC for any measurable quantizer. For deterministic experts that are Lipschitz continuous with a suitable Lipschitz constant, Proposition 5 shows a binning quantizer yields RTVC.
>
> > Weaknesses(b) and key question (2)
>
> The benefit of model-based augmentation is more fundamental than improved horizon dependence. For non-smooth (non-RTVC) deterministic experts, Theorem 6 shows that standard BC fails catastrophically even with infinite data: the deployment error remains $\Omega(1)$, because OOD rollout states unavoidably compound errors. In this regime, model-based augmentation is not merely better; it is the only approach that provably yields a meaningful guarantee. Theorem 7 makes an otherwise infeasible case feasible. The horizon improvement is secondary.
>
> As noted in Section 4, action chunking shares this spirit: it commits to actions generated from an earlier observation, where the state is closer to the expert distribution and less affected by OOD shift. Our model-based augmentation formalizes this idea.
>
> > Weaknesses (c) and key question (3)
>
> We apologize for this typo. The learner transition dynamic should depend on both the quantized action and the current state. It should be $(T_h \circ \rho)(x_{h+1}^{(i)} \mid q(u_h^{(i)}), x_h^{(i)})$, which is still a Markovian transition kernel.
>
> > Limitations
>
> We will expand the limitations section to address the points raised above. We will discuss more carefully prior theoretical literature with similar stability and smoothness assumptions. We will also provide a more balanced discussion of our model-based algorithm, emphasizing that its role is mainly conceptual.
>
> **References**
>
> [1] Angeli. "A Lyapunov approach to incremental stability properties." *IEEE TAC* 47.3 (2002).
> [2] Majid et al. "Backstepping design for incremental stability." *IEEE Transactions on Automatic Control* 56.9 (2011).
> [3] He et al. "Flying hand: End-effector-centric framework..." arXiv:2504.10334 (2025).
> [4] Chi et al. "Diffusion policy: Visuomotor policy learning via action diffusion." *IJRR* 44.10-11 (2025).
> [5] Block et al. "Provable Guarantees for Generative Behavior Cloning ..." (2023).
> [6] Tsukamoto et al. "Contraction theory for nonlinear stability analysis and learning-based control." (2021).
> [7] Simchowitz et al. "The Pitfalls of Imitation Learning ..." (2025).

---

> > ### Author Rebuttal · Reviewer_f82u · 2026-04-04
> >
> > Thank you for the rebuttal. I appreciate the clarifications, especially regarding the stability assumptions and the correction of the typo in Algorithm 1. That said, because the paper is motivated by practical problems, I still think a minimum level of simulation-based validation should be included. Without such evidence, and given that there are presentation issues even in the main algorithm, the practical impact of the paper seems limited at this stage. I will keep my score at this stage.

---

> > > ### Author Response · Authors · 2026-04-05
> > >
> > > We are glad to see that our previous rebuttal addressed some of your concerns and helped reduce ambiguity. We are also very happy to address your further questions. In particular, we conducted a simulation study to validate our main arguments. The results are shown [here](https://anonymous.4open.science/r/BC_Quantization_rebuttal-B223/quantizer_regret_comparison_base6.pdf), and we summarize the main takeaways below.
> > >
> > > **Key takeaways**
> > >
> > > 1. With a learning-based quantizer with small quantization error but a non-smooth induced quantized policy, the log-loss BC with action quantization suffers from severe distribution shift. This issue is not resolved by increasing the sample size (we use 5000 trajectories with horizon $H=60$) or scaling up the model (we use a 4-layer, 3-head transformer on scaler state and action space).
> > > 2. Under exactly the same setting, adding model-based augmentation significantly reduces the regret and resolves this issue.
> > > 3. Under the same dynamic, expert, and reward function, a binning-based quantizer (which is smooth) does not exhibit this issue.
> > >
> > > **Simulation setting**
> > >
> > > We use the 1-d system $x_{t+1}=0.2x_t+0.3u_t$ and the expert $\pi^{\star}(x)=\arctan(x)$, following the construction in the proof of Theorem 6, deterministic part. The learning-based quantizer is also constructed following the same proof; specifically, it is designed so that deploying  $q\circ\pi^{\star}(x)$ traps the state into two sets, and is thus different from the expert trajectory. The quantizer's average quantization error on expert distribution is $\varepsilon_q \approx 0.5$ (estimated via Monte Carlo, following the definition in Equation (4)). The binning-based quantizer uses uniform quantization on the action space with bin width 0.5.
> > >
> > > In the simulation, the learner's policy is parameterized by a transformer. Given a quantizer, we map the representative actions $\tilde{\mathcal U}$ to base-6 token sequences and train the transformer on these tokens. The transition model is a conditional Gaussian model in which only the mean is parameterized by a neural network, while the variance is fixed. We train the transformer policy and transition model on 5000 trajectories until converge.
> > >
> > > **Explanation of the figures**
> > >
> > > To help interpret the figures, the first row show the quantized policy $q\circ\pi^{\star}(x)$ induced by a non-smooth quantizer and a binning-based quantizer, respectively. Specifically, the non-smooth quantizer is the learning-based quantizer described above, which achieves small quantization error on the expert distribution yet lacks smoothness. The second row and the first panel of the third row show the trajectories obtained by deploying the models imitating these quantized policies, as well as the trajectory obtained via model-based augmentation. The last panel reports the regret, demonstrating that model-based augmentation significantly reduces it.
> > >
> > > **Clarification of our contribution for model-based augmentation**
> > >
> > > Here we clarify our contribution of proposing the model-based augmentation. As shown by the simulation, when a non-smooth quantizer is used, log-loss BC can fail completely. Moreover, this issue cannot be resolved simply by using a larger model or more data, because, as illustrated in Theorem 6, the oracle quantized expert that the learner tries to imitate is itself problematic. Therefore, model-based augmentation is, to the best of our knowledge, the most natural and only provably correct remedy for this issue. Our paper does not present this as a major algorithmic contribution; rather, we argue that the principled remedy for non-smoothness is to avoid deploying the learned policy on out-of-distribution states. We believe that, in practice, action chunking has a similar spirit and is already widely adopted.

---

### Official Review · Reviewer_56te · 2026-03-13

**Soundness:** 2
**Presentation:** 3
**Significance:** 3
**Originality:** 3
**Overall Recommendation:** 4
**Confidence:** 3

**Summary:**

The paper studies behavior cloning with autoregressive models when continuous actions are discretized through quantization, with the goal of understanding when action quantization works and when it fails. Its main theoretical focus is to characterize how quantization error and statistical estimation error jointly propagate over the horizon in finite-horizon MDPs, showing sample-complexity bounds with only polynomial dependence on horizon under suitable conditions, while also distinguishing between different quantization schemes. The paper further proposes a model-based augmentation intended to avoid the smoothness requirement on the quantized policy.

**Compliance With Llm Reviewing Policy:**

Affirmed.

**Final Justification:**

The rebuttal clarified my main concerns, and I will raise my score accordingly.
I think that the work provides valuable theory-to-practice insights once the presentation concerns are addressed.

**Key Questions For Authors:**

Questions:
- Is my understanding correct that the theory suggests simple binning-style quantizers may be more robust, or easier to analyze, than generic learned quantizers in deterministic BC?

**Limitations:**

yes

**Strengths And Weaknesses:**

Strengths:
- The authors clearly motivates the problem, and action quantization is highly relevant in current practice.
- The paper provides substantial preliminaries, which improve readability.
- The paper shows that small in-distribution quantization error alone is insufficient, and identifies policy smoothness after quantization as a key condition controlling deployment regret; this is a clear conceptual contribution.
- The analysis distinguishes stochastic vs deterministic experts and binning-based vs generic learned quantizers.


Weaknesses:
- The main guarantees rely on P-IISS/P-EIISS stability and TVC/RTVC-type smoothness assumptions whose relevance or verifiability in practical settings remains unclear. Can the authors comment on that point?
- The paper would benefit from empirical evidence showing either that these assumptions meaningfully diagnose real quantizers or that the proposed model-based augmentation is useful in practice, especially given the paper’s motivation around VLAs and autoregressive models.
- Although the model-based augmentation improves the stated horizon dependence of the quantization term, this benefit may be partly offset in practice, since the added transition-model complexity can itself conceal horizon-dependent factors.


Minor:
- Assumption 1: missing space after closing parenthesis.

---

> ### Author Rebuttal · Authors · 2026-03-31
>
> We thank the reviewer for the constructive feedback.
>
> > Weakness 1
>
> **Stability (P-IISS/P-EIISS).**
>
> Firstly, our P-IISS is strictly weaker than the standard $\delta$-ISS notion (Definition 4.1 of [1]), which requires global stability and is the basic assumption in recent line of imitation learning works for continuous control. We only require stability on a local subset $S$ where the expert trajectory remains with high probability; see Remark 3 in Appendix B. The $\delta$-ISS framework is one of the most widely used stability paradigm in nonlinear control, see [6] for a tutorial. Hence, P-IISS applies more broadly.
>
> Secondly, Proposition 16 gives an explicit sufficient condition: if the dynamics satisfy a local contraction $\|f(x,u)-f(x',u')\| \le \eta\|x-x'\| + C\|u-u'\|$ with $\eta<1$ on a subset $S$, then P-EIISS holds. Although strong, this condition is only needed locally along the expert trajectory. More broadly, one can also stabilize the system through a controller and then work with the resulting closed-loop dynamics. In that case, the learner imitates deviations in the controller input, while stability is inherited from the stabilized dynamics; [2] gives an example.
>
> Third, our assumption is especially natural in systems with low-level controllers, where the learner should imitate the controller rather than the raw action. The resulting controller-induced closed-loop dynamics are already stabilized. Recent systems such as [3] and [4] follow this architecture. Moreover, following [5], when the low-level controller is a stabilizing affine mapping, their Definition 5.2 is provably satisfied (Lemma H.3 of [5]), and Definition 5.2 directly implies our P-IISS. Hence, systems under [5]'s framework satisfy our assumption with a quantitative guarantee.
>
> **Smoothness(TVC/RTVC).**
>
> We note that some smoothness condition is **necessary**: [7] proves that for smooth deterministic policies, imitation error grows exponentially with horizon even under stable dynamics. TVC/RTVC is among the weakest conditions that provably avoids this exponential blowup.
>
> Our theory requires: (1) the policy class is TVC/RTVC, so the learned $\hat\pi$ satisfies TVC/RTVC; (2) the expert is realizable by this class, so it inherits the same property. For an arbitrary learner, Proposition 18 shows that after Gaussian smoothing, the new learner can always satisfy TVC. For the expert, Appendix B.2 shows Gaussian policies are already TVC, and by Lemma 15 the quantized policy remains TVC for any measurable quantizer. For deterministic experts that are Lipschitz continuous with a suitable Lipschitz constant, Proposition 5 shows a binning quantizer yields RTVC.
>
> > Weakness 2
>
> **Smoothness matters.** Theorem 6 shows a learned quantizer can have small in-distribution error yet cause large deployment error by breaking RTVC, while binning preserves it (Proposition 5). This matches practice: RT-1 [8], RT-2 [9], and OpenVLA [10] use uniform binning with no post-hoc correction. In contrast, VQ-BeT [11] uses a learned vector quantizer and clearly states that they requires a continuous correction step, from our perspective, this correction compensates for the smoothness violation that learned quantizers introduce.
>
> **Stability matters.** Diffusion Policy [4] finds end-effector position control much better than velocity control. This is consistent with our theory, since position control induces a more contractive closed-loop system.
>
> > Weakness 3
>
> The benefit of model-based augmentation is more fundamental than improved horizon dependence. For non-smooth (non-RTVC) deterministic experts, Theorem 6 shows that standard BC fails catastrophically even with infinite data: the deployment error remains $\Omega(1)$, because OOD rollout states unavoidably compound errors. In this regime, model-based augmentation is not merely better; it is the only approach that provably yields a meaningful guarantee. Theorem 7 makes an otherwise infeasible case feasible. The horizon improvement is secondary.
>
> As noted in Section 4, action chunking shares this spirit: it commits to actions generated from an earlier observation, where the state is closer to the expert distribution and less affected by OOD. Our model-based augmentation formalizes the idea.
>
> > Key question
>
> Yes. Binning preserves RTVC (Proposition 5), while learned quantizers can break it and induces huge regret (Theorem 6).
>
> [1] Angeli. “Lyapunov approach to incremental stability ...” 2002.
> [2] Majid et al. “Backstepping design for incremental stability.” 2011.
> [3] He et al. “Flying Hand.” 2025.
> [4] Chi et al. “Diffusion Policy.” 2025.
> [5] Block et al. "Provable Guarantees for Generative Behavior Cloning ..." (2023).
> [6] Tsukamoto et al. “Contraction theory ...” 2021.
> [7] Simchowitz et al. “Pitfalls of Imitation Learning ...” 2025.
> [8] Brohan et al. “RT-1.” 2022.
> [9] Brohan et al. “RT-2.” 2023.
> [10] Kim et al. “OpenVLA.” 2024.
> [11] Lee et al. “Behavior Generation with Latent Actions.” 2024.

---

> > ### Author Rebuttal · Reviewer_56te · 2026-04-03
> >
> > The rebuttal was helpful, and I appreciated the added discussion connecting the theory to design choices used in practice. I think these connections strengthen the paper and would benefit from being made more visible in the revision.
> >
> > At the same time, my original concern is only partially resolved. The paper is strongly motivated by practical usage, but I still would have liked to see at least some empirical evidence, even in simple toy settings, illustrating whether the assumptions are diagnostically meaningful for real quantizers and whether the proposed model-based augmentation is useful and computationally reasonable in practice.
> >
> > Taken together, the rebuttal clarified several points for me, but it does not materially change my original evaluation.

---

> > > ### Author Response · Authors · 2026-04-05
> > >
> > > We are glad to see that our previous rebuttal addressed some of your concerns and helped reduce ambiguity. We are also very happy to address your further questions. In particular, we conducted a simulation study to validate our main arguments. The results are shown [here](https://anonymous.4open.science/r/BC_Quantization_rebuttal-B223/quantizer_regret_comparison_base6.pdf), and we summarize the main takeaways below.
> > >
> > > **Key takeaways**
> > >
> > > 1. With a learning-based quantizer with small quantization error but a non-smooth induced quantized policy, the log-loss BC with action quantization suffers from severe distribution shift. This issue is not resolved by increasing the sample size (we use 5000 trajectories with horizon $H=60$) or scaling up the model (we use a 4-layer, 3-head transformer on scaler state and action space).
> > > 2. Under exactly the same setting, adding model-based augmentation significantly reduces the regret and resolves this issue.
> > > 3. Under the same dynamic, expert, and reward function, a binning-based quantizer (which is smooth) does not exhibit this issue.
> > >
> > > **Simulation setting**
> > >
> > > We use the 1-d system $x_{t+1}=0.2x_t+0.3u_t$ and the expert $\pi^{\star}(x)=\arctan(x)$, following the construction in the proof of Theorem 6, deterministic part. The learning-based quantizer is also constructed following the same proof; specifically, it is designed so that deploying  $q\circ\pi^{\star}(x)$ traps the state into two sets, and is thus different from the expert trajectory. The quantizer's average quantization error on expert distribution is $\varepsilon_q \approx 0.5$ (estimated via Monte Carlo, following the definition in Equation (4)). The binning-based quantizer uses uniform quantization on the action space with bin width 0.5.
> > >
> > > In the simulation, the learner's policy is parameterized by a transformer. Given a quantizer, we map the representative actions $\tilde{\mathcal U}$ to base-6 token sequences and train the transformer on these tokens. The transition model is a conditional Gaussian model in which only the mean is parameterized by a neural network, while the variance is fixed. We train the transformer policy and transition model on 5000 trajectories until converge.
> > >
> > > **Explanation of the figures**
> > >
> > > To help interpret the figures, the first row show the quantized policy $q\circ\pi^{\star}(x)$ induced by a non-smooth quantizer and a binning-based quantizer, respectively. Specifically, the non-smooth quantizer is the learning-based quantizer described above, which achieves small quantization error on the expert distribution yet lacks smoothness. The second row and the first panel of the third row show the trajectories obtained by deploying the models imitating these quantized policies, as well as the trajectory obtained via model-based augmentation. The last panel reports the regret, demonstrating that model-based augmentation significantly reduces it.
> > >
> > > **Clarification of our contribution for model-based augmentation**
> > >
> > > Here we clarify our contribution of proposing the model-based augmentation. As shown by the simulation, when a non-smooth quantizer is used, log-loss BC can fail completely. Moreover, this issue cannot be resolved simply by using a larger model or more data, because, as illustrated in Theorem 6, the oracle quantized expert that the learner tries to imitate is itself problematic. Therefore, model-based augmentation is, to the best of our knowledge, the most natural and only provably correct remedy for this issue. Our paper does not present this as a major algorithmic contribution; rather, we argue that the principled remedy for non-smoothness is to avoid deploying the learned policy on out-of-distribution states. We believe that, in practice, action chunking has a similar spirit and is already widely adopted.

---

### Official Review · Reviewer_D4Sx · 2026-03-23

**Soundness:** 3
**Presentation:** 3
**Significance:** 3
**Originality:** 3
**Overall Recommendation:** 4
**Confidence:** 2

**Summary:**

The paper provides theoretical foundations of action quantization for behavior cloning in continuous control tasks. The authors show that quantized actions with log loss provides optimal sample complexity for behavior cloning that matches existing lower bounds and shows that the quantization error is polynomial in the horizon length provided that the dynamics are stable and the policy satisfies . The authors then presents different quantization schemes and present a model based augmentation that improves the error bound bypassing the policy smoothness requirement.

**Compliance With Llm Reviewing Policy:**

Affirmed.

**Key Questions For Authors:**

The authors mention that their results indicate that under log loss behavior cloning binning based quantizers are preferable compared to general learning based ones, can you elaborate on this? Is based on the worst case behavior?

**Limitations:**

The authors motivate the work, on the practical setting of using action quantization  using the transformer as a policy function. While the theoretical understanding is useful, presenting an experiment with transformers will reduce the friction of moving from theory to satisfying different conditions and assumptions in practice and presenting some simulation results.

**Strengths And Weaknesses:**

The paper is easy to follow and the authors did a good job explaining the central idea. Regarding the pressentation, I liked the fact that the authors made a clear distinction between statistical error and quantization induced rollout missmatch. However, in the beginning of the paper, the author clearly motivates the work around auto regressive transformers however, almost all the theoretical work, presented here, relates to policies with finite classes and smoothness assumptions. It will be great if the authors can make it more explicit on how these translates to the initial motivation of using autoregressive tnasformers as a policy. Given the entire theoretical nature of this paper, I believe some empirical evaluation with transformers as a policy that learns via BC, will be useful for the motivation. Overall I find this paper studying an interesting problem albeit some notational inconsistencies for instance in section 2.1, the authors use \mathcal U to define action space but then writes the transition function in terms of \mathcal A.

---

> ### Author Rebuttal · Authors · 2026-03-31
>
> Thank you for your time and effort in reviewing our paper. Below, we address your concerns and provide further clarification.
>
> > In the beginning of the paper, the author clearly motivates the work around auto regressive transformers however, almost all the theoretical work, presented here, relates to policies with finite classes and smoothness assumptions. It will be great if the authors can make it more explicit on how these translates to the initial motivation of using autoregressive tnasformers as a policy.
>
> Firstly, it is quite common for theoretical work to assume a finite policy class and realizability even when motivated by autoregressive models like transformers. For example, both [1] and [2] all adopt such simplified assumptions and focus on general function classes. The transition from finite-class realizability to specific function classes typically only requires standard coverage and misspecification arguments, which are not the main focus of our paper.
>
> As for whether transformers fit our smoothness assumptions, one possible simplification is to consider autoregressive linear models, as discussed in [2] and [3]. Under such a simplification, one can show that the TVC assumption holds if appropriate boundedness conditions are imposed on the parameter space ( for example under similar conditions of Assumption 2.2 of [3] ). However, overall we feel that this line of analysis is still somewhat toy-like, and for this reason we chose not to include it in the current version. Our main message regarding the smoothness assumptions is that small in-distribution quantization error alone is not sufficient; a probabilistic smoothness condition is needed to ensure that trajectories can recover from OOD states and return toward the expert trajectory.
>
> > Given the entire theoretical nature of this paper, I believe some empirical evaluation with transformers as a policy that learns via BC, will be useful for the motivation.
> >
> > presenting an experiment with transformers will reduce the friction of moving from theory to satisfying different conditions and assumptions in practice and presenting some simulation results.
>
> We agree. At present, we would refer readers to [4], [5], and [6], where binning-based quantizers are used together with log-loss behavior cloning using transformers, and they already attain desirable empirical results. In [7], the authors use a vector quantization scheme to train the quantizer and then train the policy with GPT-style behavior cloning using transformers. They report that continuous corrections are important during policy training, which, in our interpretation, suggests that some form of smoothness adjustment is needed and that pure vector quantization may be risky. We will discuss these connections more carefully in the revised version.
>
> > The authors mention that their results indicate that under log loss behavior cloning binning based quantizers are preferable compared to general learning based ones, can you elaborate on this? Is based on the worst case behavior?
>
> Yes. The reason is that learning-based quantizers typically only minimize quantization error on the expert distribution and often do not provide any smoothness guarantee. As a result, when such quantized policies are deployed, the initial quantization error may push the system into OOD states, and without smoothness, the policy may fail to recover and instead induce severe distribution shift. Theorem 6 gives exactly such an example: we construct quantizers with small quantization error but no smoothness, and deploying the resulting policies can still induce very large regret.
>
> >  notational inconsistencies for instance in section 2.1
>
> We agree. These inconsistencies will be corrected in the revised version, and we apologize for the confusion they caused.
>
> [1] Foster, Dylan J., Adam Block, and Dipendra Misra. "Is behavior cloning all you need? understanding horizon in imitation learning." *Advances in Neural Information Processing Systems* 37 (2024).
>
> [2] Rohatgi, Dhruv, et al. "Computational-Statistical Tradeoffs at the Next-Token Prediction Barrier: Autoregressive and Imitation Learning under Misspecification." *The Thirty Eighth Annual Conference on Learning Theory*. PMLR, 2025.
>
> [3] Chen, Fan, et al. "The Coverage Principle: How Pre-Training Enables Post-Training." *arXiv preprint arXiv:2510.15020* (2025).
>
> [4] Brohan, Anthony, et al. "Rt-1: Robotics transformer for real-world control at scale." *arXiv preprint arXiv:2212.06817* (2022).
>
> [5] Zitkovich, Brianna, et al. "Rt-2: Vision-language-action models transfer web knowledge to robotic control." *Conference on Robot Learning*. PMLR, 2023.
>
> [6] Kim, Moo Jin, et al. "Openvla: An open-source vision-language-action model." *arXiv:2406.09246* (2024).
>
> [7] Lee, Seungjae, et al. "Behavior generation with latent actions." *arXiv:2403.03181* (2024).

---

### Decision · Program_Chairs · 2026-04-30

**Decision:**

Accept (regular)

**Comment:**

The paper provides a theoretical analysis of action quantization in BC. It successfully establishes upper bounds on regret by separating statistical estimation error from quantization-induced mismatch. It advances the theoretical understanding of a widely used but poorly understood design choice in modern robotics and generative AI. The combination of upper/lower bounds and the practical insights regarding binning vs. learned quantization makes it a solid contribution. After the rebuttal, most of the concerns from the reviewers have been addressed, and the final scores are two Accept and two Weak Accept, so the reviewers reached a consensus of acceptance. AC has checked the submission, the scores, the rebuttal, and the discussion, and agreed with reviewers' consensus, thus acceptance is recommended.